# Immune-privileged tissues formed from immunologically cloaked mouse embryonic stem cells survive long term in allogeneic hosts

Jeffrey Harding[1,9], Kristina Vintersten-Nagy[1,2,9], Huijuan Yang[1,2,9], Jean Kit Tang[1,2], Maria Shutova[1], Eric D. Jong [1,3], Ju Hee Lee[4,5], Mohammad Massumi[1], Tatiana Oussenko[1], Zohreh Izadifar[1], Puzheng Zhang[1], Ian M. Rogers [1,2,6], Michael B. Wheeler[2,7], Stephen J. Lye[1,2,6], Hoon-Ki Sung[4,5], ChengJin Li[1], Mohammad Izadifar[1] & Andras Nagy [1,3,6,8] ✉

The immunogenicity of transplanted allogeneic cells and tissues is a major hurdle to the advancement of cell therapies. Here we show that the overexpression of eight immunomodulatory transgenes (*Pdl1*, *Cd200*, *Cd47*, *H2-M3*, *Fasl*, *Serpinb9*, *Ccl21* and *Mfge8*) in mouse embryonic stem cells (mESCs) is sufficient to immunologically 'cloak' the cells as well as tissues derived from them, allowing their survival for months in outbred and allogeneic inbred recipients. Overexpression of the human orthologues of these genes in human ESCs abolished the activation of allogeneic human peripheral blood mononuclear cells and their inflammatory responses. Moreover, by using the previously reported FailSafe transgene system, which transcriptionally links a gene essential for cell division with an inducible and cell-proliferation-dependent kill switch, we generated cloaked tissues from mESCs that served as immune-privileged subcutaneous sites that protected uncloaked allogeneic and xenogeneic cells from rejection in immune-competent hosts. The combination of cloaking and FailSafe technologies may allow for the generation of safe and allogeneically accepted cell lines and off-the-shelf cell products.

A barrier to getting cell-based therapies into the clinic is that cell products derived from a foreign donor can be immune rejected. Immunosuppressive drugs are one tool to mitigate these risks[1]. Alternatively, there are also ongoing and promising programmes to develop autologous cell therapies derived from induced pluripotent stem cells (iPSCs). However, there will be many clinical situations where the costs and time needed to derive and differentiate autologous cells are prohibitive; for instance, in patients needing immediate treatment for a given disease or injury. As a response, some countries are developing iPSC banks containing cell lines that are homozygous

[1]Lunenfeld-Tanenbaum Research Institute, Sinai Health System, Toronto, Ontario, Canada. [2]Department of Physiology, University of Toronto, Toronto, Ontario, Canada. [3]Institute of Medical Science, University of Toronto, Toronto, Ontario, Canada. [4]Translational Medicine Program, The Hospital for Sick Children, Toronto, Ontario, Canada. [5]Department of Laboratory Medicine and Pathobiology, University of Toronto, Toronto, Ontario, Canada. [6]Department of Obstetrics and Gynaecology, University of Toronto, Toronto, Ontario, Canada. [7]Toronto General Hospital Research Institute, Toronto, Ontario, Canada. [8]Australian Regenerative Medicine Institute, Monash University, Melbourne, Victoria, Australia. [9]These authors contributed equally: Jeffrey Harding, Kristina Vintersten-Nagy, Huijuan Yang. ✉e-mail: nagy@lunenfeld.ca

**Fig. 1 | Generating cloaked mESCs. a**, Candidate immunomodulatory factors and their roles in the innate and adaptive immune pathways involved in rejection of non-self. The underlined factors were selected to generate transgene-containing vectors. **b**, Schema showing the generation of clonal B6 mESCs that express the selected eight immunomodulatory transgenes via insertion with piggyBac (PB) and Sleeping Beauty (SB) transposon expression vectors. **c**, Expression level of each inserted transgene by RT–qPCR. Values are shown relative to the corresponding expressions in splenocytes (thick black line on each

radial graph) from B6 mice stimulated ex vivo with αCD3ε and αCD28 antibodies. Concentric circles show $\log_{10}$ scale. Untransfected (FS) mESCs (parental line) are shown in upper left corner. Clones outlined in red and blue boxes were later screened for acceptance in allogeneic hosts. Red boxes, clones accepted in allogeneic hosts (clones 1 and 43, renamed Klg-1 and Klg-2, respectively). Blue boxes, clones rejected. **d**, In vitro antibody staining of all transgene-encoded factors in Klg-1 mESCs.

for the most common human leucocyte antigen (HLA) alleles among a given population[2,3].

In parallel to these programmes, several groups have worked to immune engineer or encapsulate pluripotent stem cells (PSCs) and PSC-derived cells to resist or escape immune rejection[4]. Many of these approaches involve knockout of the highly polymorphic class I and II genes in the major histocompatibility complex (MHC) to remove the major source of antigen mismatch between donor and

recipients[5–9]. Some also rely on overexpression of immunomodulatory factors[10–15] to protect donor cells from the immune rejection caused by minor antigens[16–18].

We first sought to engineer PSCs that could escape immune rejection solely by overexpression of immunomodulatory transgenes and without deletion of MHC genes. We were encouraged by evolved immune-evading strategies in nature[19–21] and especially those where MHC expression is retained, such as the type 2 facial tumours in

Tasmanian Devils[22–25]. On the basis of these examples and the complexity of mammalian immunity, we collected candidate transgenes and overexpressed many of them to simultaneously interfere with antigen-presenting cells, lymphocytes and monocytes across the innate and adaptive arms of the immune system (Fig. 1a). We selected eight factors, including (1) CCL21 to disrupt dendritic cell migration[26], (2) PDL1, FASL, H2-M3 and SERPINB9 to target and/or protect against T-cell and NK-cell attack[27–29] and (3) CD47, CD200 and MFGE8 to target monocytes and macrophages[30–32]. A goal in our selection criteria was to minimize effects on systemic immunity. Five of the eight factors are membrane bound and one is cytoplasmic. Two are secreted, with CCL21 having a heparin-binding domain for retention[33,34] and MFGE8 binding to tissue collagen[35]. We aimed to test the long-term resistance to rejection of these engineered ESCs in fully immunocompetent hosts, which many short-term or human models of allotransplantation do not recapitulate.

When paired with clonal screening, this approach allowed for the generation and isolation of 'cloaked' mESC lines that could survive and form long-term accepted, solid tissues in several MHC-mismatched immunocompetent strains. Extending these findings to humans, we also show that overexpression of the same human functional orthologues in human PSCs allowed their differentiated derivatives to block the activation and inflammatory response of allogeneic peripheral blood mononuclear cells (PBMCs).

Building on this work and leveraging our genetically integrated FailSafe (FS) suicide system[36,37], we also show that the dormant subcutaneous tissues generated from these engineered mouse embryonic stem cells (mESCs) maintain transgene overexpression and protect unmodified cells or tissue injected into the modified tissue. In doing so, we show that tissues formed from these engineered cells act as immune-privileged sites, conferring their protection to embedded uncloaked allogeneic and even xenogeneic cells. Together, our data show an effective approach for engineering cells that are resistant to immune rejection. Ultimately, this will support the development of pluripotent cells that can be used to generate therapeutic cells that persist in many different allogeneic recipients without the need for immune suppression. Our data also describe a method to generate immune-privileged tissue that can be used to host or protect allo- or xenogeneic unmodified cells, which could ultimately expand the scope of cell therapy approaches.

## Results

### Overexpression of 'cloaking' immune modulatory transgenes in mESCs

We cloned the coding sequence (CDS) of mouse *Pdl1*, *Cd200*, *Cd47*, *H2-M3*, *Fasl*, *Serpinb9*, *Ccl21* and *Mfge8* genes, as well as luciferase, into individual piggyBac[38–40] and Sleeping Beauty[41,42] transposon (cloaking) vectors in which transcription was driven by a CAG promoter[43] and linked to a drug or fluorescent selectable marker (Fig. 1b and Extended Data Fig. 1a,b). All eight cloaking vectors were then transfected into a C57BL/6N (B6) mESC line[44] that contained the recently described and genomically integrated FailSafe (FS) system[36], which consists of the herpes virus thymidine kinase (TK) gene, a negatively selectable 'kill-switch', transcriptionally linked to both alleles of the endogenous *Cdk1* gene essential for cell division[36]. Any dividing cell that expresses *Cdk1* will also express TK and can therefore be killed by treatment with the pro-drug ganciclovir (GCV)[45].

After transfection of FS mESCs with the cloaking vectors, we used stringent drug selection and/or fluorescence-activated cell sorting (FACS) enrichment (Extended Data Fig. 1b) to generate polyclonal pools of cells with high levels of each cloaking factor. From these pools, we then established 47 single-cell-derived clonal lines ('cloaked FS mESC lines') and quantified the expression level of each transgene by reverse transcription–quantitative polymerase chain reaction (RT–qPCR) (Fig. 1b,c). As expected, based on the random integration of transposon vectors,

there was a wide distribution of expression levels among the clonal lines. Expression of the protein encoded by each transgene was confirmed in vitro by immunohistochemistry on selected clones (Fig. 1d). These factors were undetectable in parental FS mESCs, except for MFGE8 and SERPINB9, which is consistent with our RT–qPCR results and published data[46,47].

### B6 cloaked mESCs survive long term and form teratomas in allogeneic recipients

From the 47 cloaked FS mESC lines generated, we picked 13 candidates with the highest overall transgene expression levels as measured by RT–qPCR and tested their ability to survive, compared to non-transgenic cells, relatively short term in immunocompetent allogeneic recipients. As a readout, we relied on the inserted luciferase gene and bioluminescent imaging (BLI) after subcutaneous injection into the neck or flank. As expected, wild-type (WT) FS B6 (MHC H-2k$^b$) mESCs without any of the cloaking transgenes survived in isogeneic B6 recipients (Fig. 2a top row) but were cleared from allogeneic inbred FVB (H-2k$^q$), C3H (H-2k$^k$), BALB/c (H-2k$^d$) and outbred CD-1 mice within 5–10 days (Fig. 2a bottom rows). One representative recipient is shown in Fig. 2a, but 10–20 total recipients for each strain were followed after transplantation using BLI to show clearance. However, at least two of the cloaked clones ('Clone 1' and 'Clone 43' in Fig. 1c, denoted as Klg-1 and Klg-2 in further studies, respectively) survived for the 17-day test period (Fig. 2b for Klg-1). Interestingly, these clones, especially Klg-1, were among the highest overall expressors of the cloaking transgenes compared with all 47 lines generated and measured by qPCR (Fig. 1c).

We then tested whether Klg-1 and Klg-2 mESCs could form a palpable and differentiated tissue (a teratoma) in both isogeneic and allogeneic recipients (Fig. 2c and Extended Data Fig. 2), teratoma being expected to develop ~2–6 weeks after subcutaneous injection. Indeed, both Klg-1 and Klg-2 formed teratomas in all allogeneic backgrounds tested (Fig. 2d,e and Extended Data Fig. 2b,c). Within 2 months after transplantation, mESC-derived teratomas usually reach a size that requires euthanasia of the recipients (Fig. 2d top graph), which limits the length of these graft survival studies. Therefore, we activated the FS system by regularly dosing recipients with GCV for several weeks (orange interval in Fig. 2d) to keep the tissue at an approximate volume of 0.5 cm³. Figure 2d shows three representative recipients for each condition. As already described[36], GCV treatment ablated the proliferating cells and stabilized the tissue volume long term (Fig. 2d lower graph and Extended Data Fig. 2b). Doing this allowed us to generate dormant Klg-1-derived transplants in more than 100 allogeneic recipients among all genetic backgrounds tested (Fig. 2d,e). The total number of transplant recipients for reach strain tested is indicated in the horizontal bar of Fig. 2e (which includes recipients shown in Fig. 2d). Notably, we did not observe a single instance of clearance once these transplants formed, including in mice monitored for 9 months after cell injection. We also showed that Klg-2 could form long-term-accepted dormant transplants in allogeneic recipients, albeit with different efficiencies compared with Klg-1 in the two recipient strains (FVB and C3H) that were tested (Fig. 2e and Extended Data Fig. 2c, number of transplant recipients shown in the horizontal bars). A complete list of all tested mice and the observation period in these in vivo experiments is shown in Supplementary Table 3.

Klg-1 behaved like typical mESCs in vitro, with characteristic morphology and ability to form embryoid bodies (EBs), positive alkaline phosphatase (AP) staining, and expressed both Oct4 and SSEA1 (Extended Data Fig. 3a–d). Klg-1 teratomas that formed in both isogeneic and allogeneic recipients were well vascularized and contained cells from all three embryonic germ layers (Extended Data Fig. 3e,f), showing that overexpression of the eight cloaking factors did not alter the developmental potential of the cell line, which remained pluripotent. Teratomas from the Klg-1 line expressed high levels of MHC class I even after long-term acceptance in allogeneic recipients,

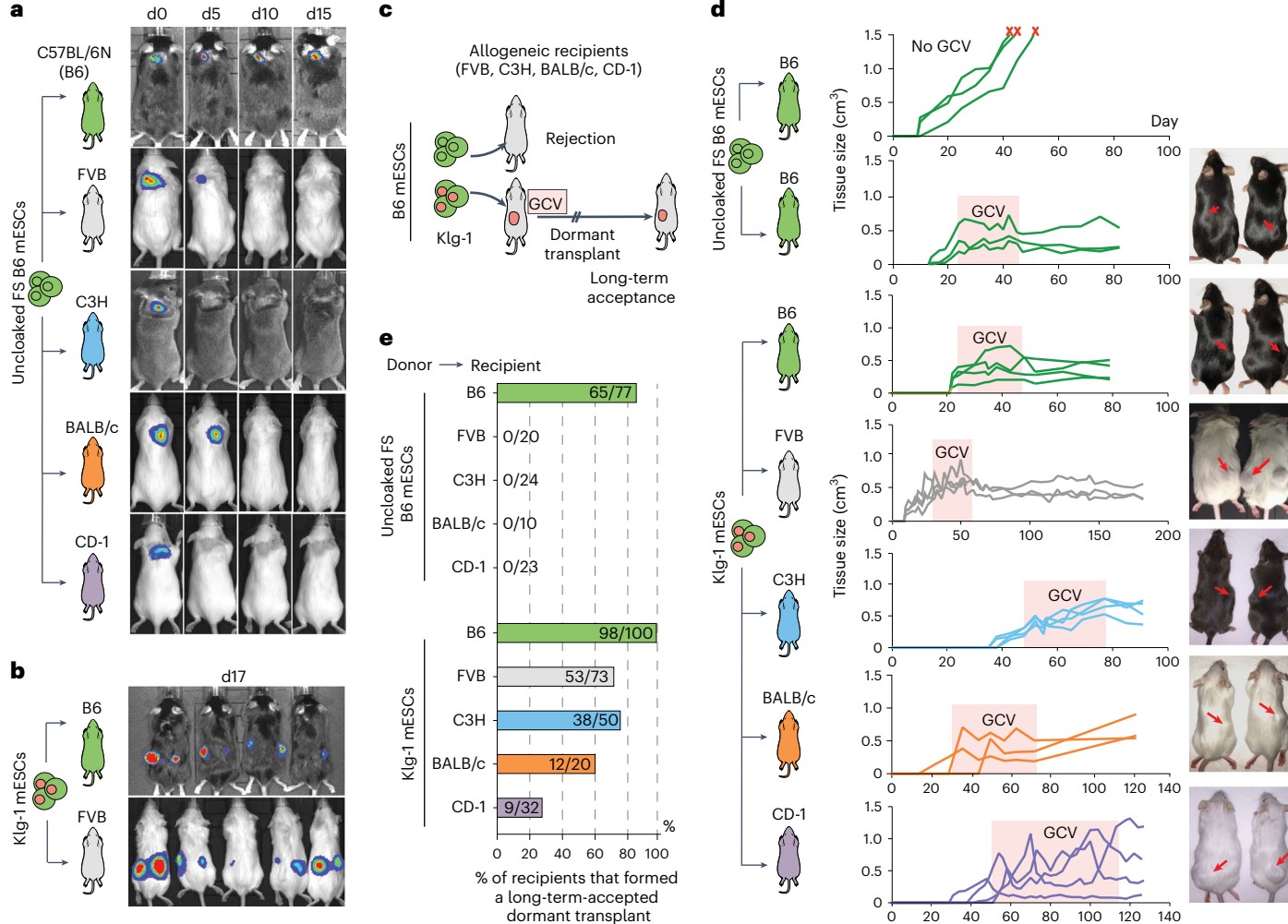

**Fig. 2 | Cloaked mESCs escape immune rejection and survive long term in various immunocompetent and MHC-mismatched recipient strains.**
**a**, Uncloaked FS B6 mESCs were injected subcutaneously into isogeneic (B6) or allogeneic (FVB, C3H, BALB/c or CD-1) recipients and followed for 15 days with BLI. One representative mouse is shown for each condition, but 10–20 were transplanted and followed for each condition. **b**, Klg-1 mESCs injected into FVB recipients; BLI showed the presence of donor cells 17 days after injection in one representative mouse. **c**, Assay to test rejection or long-term acceptance of cloaked FS mESCs. Rejected cells are cleared and no teratoma forms; acceptance results in formation and expansion of the teratoma. The growth of teratomas is halted by activation of the FS system with GCV to ablate dividing cells and allow for testing of long-term persistence/survival of injected allogeneic cells. **d**, Growth and survival curves of untreated and GCV-stabilized

teratomas derived from uncloaked FS and Klg-1 FS B6 mESCs in 3 representative isogeneic and allogeneic (FVB, C3H, BALB/c and CD-1) recipients. Teratoma-growth curves in 3–4 recipients are shown per group. Pictures of representative teratomas (red arrows) shown on the right were taken at the endpoint. Red 'x' indicates recipients were euthanized. **e**, Percent of isogeneic and allogeneic recipients that formed long-term and GCV-stabilized teratomas from uncloaked FS and Klg-1 FS mESCs. Horizontal bar shows the total number of recipients that formed teratomas divided by the total number of mice injected. Observation period ranged from 2 to 9 months depending on euthanasia date. No instances of Klg-1 teratoma clearance occurred in any recipient once formed and stabilized. Data for all recipients including observation period are shown in Supplementary Table 1.

comparable to the tissues derived from uncloaked FS parental line formed in isogeneic recipients (Extended Data Fig. 3g). This shows that acceptance of Klg-1 cells in allogeneic recipients was not the result of loss of surface expression of allogeneic MHC class I protein. We then tested whether Klg-1 could differentiate into specific lineages and cell types in vitro using various spontaneous and directed protocols. Indeed, the Klg-1 line could form a variety of cell types including neurofilament positive ectoderm cells, smooth muscle actin expressing mesoderm cells, tube-forming CD31 positive endothelial-like cells, Fox2A/Sox17 positive definitive endoderm, troponin-positive cardiomyocytes and Oil Red O positive adipocytes (Extended Data Fig. 4). Therefore, the overexpression of the cloaking factors did not interfere with the developmental potential of the line into numerous cell types with therapeutic value.

To quantify the expression level of each cloaking transgene in the context of the entire transcriptome of the cells, we did next-generation sequencing (NGS) on total RNA isolated from uncloaked FS and Klg-1 mESCs, as well as an excised, long-term- accepted teratoma formed by Klg-1 cells in an allogeneic recipient (Fig. 3a). As expected from the initial factor-specific RT–qPCR analysis, the cloaking transgenes were all expressed at very high levels in Klg-1 compared with uncloaked parental cells (Fig. 3a, red and green lines), and these high transgene levels were maintained in the long-term-accepted dormant transplant formed in allogeneic hosts (Fig. 3a, blue line). Since the teratomas are composed of many spontaneously differentiated cell types, these data show the robustness of transposon transgene expression achieved with stringent selection and screening. In fact, placed against all the endogenous genes of the genome, the cloaking transgenes were among the 5% highest of

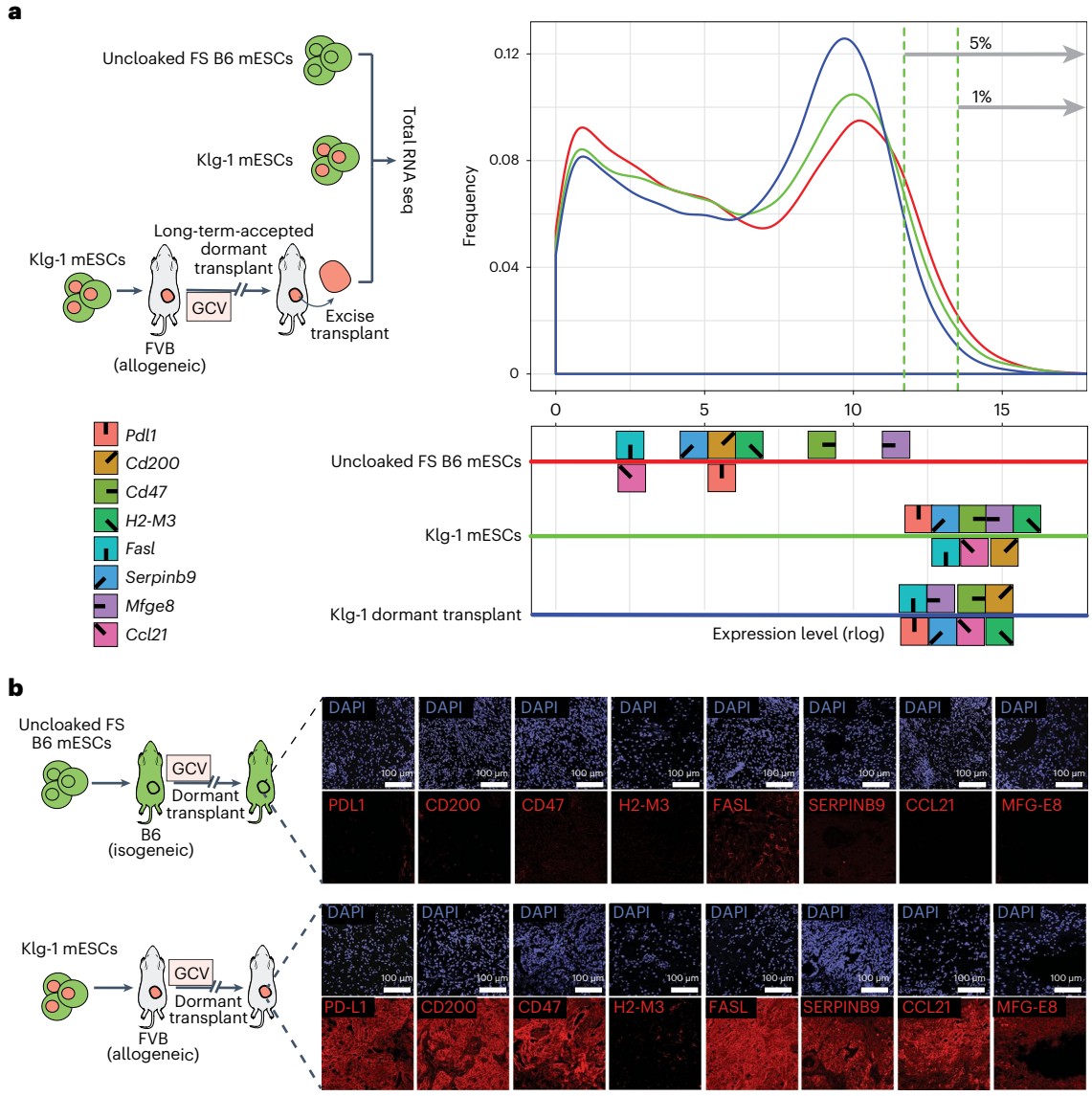

**Fig. 3 | Klg-1 cloaking transgenes within the gene expression profile. a**, Left: experimental schema showing collection of RNA for whole-genome RNA-seq expression analysis, from uncloaked FS and Klg-1 mESCs, as well as cells isolated from a Klg-1-derived dormant teratoma in allogeneic FVB recipients. Right: gene expression level distribution in uncloaked FS (red), Klg-1 (green) mESCs and a Klg-1 teratoma (blue), with plot showing the proportion of genes (y axis) at a given expression level (x axis). The symbols with corresponding colours show the cloaking gene expression levels in the distributions. Vertical green dotted lines show 95–99% highest expression range. **b**, Antibody staining of immune factors in long-term-accepted teratomas derived in isogeneic (top) and allogeneic (bottom) settings.

all genes expressed (Fig. 3a). As with the RNA, immunohistochemistry showed robust protein levels for all cloaking factors, except for H2-M3 (which has a complex pathway for transport to the cell surface[48]), in long-term-accepted, dormant Klg-1-derived tissues (Fig. 3b).

We next did splinkerette PCR (spPCR) on genomic DNA from Klg-1 mESCs using primers against the transposon inverted terminal repeats, which flank each cloaking transgene, and then sequenced the product by NGS. By comparing the amplified sequence of the products to a reference mouse genome[49], we identified 56 genomic sites that had a piggyBac transposon insertion in Klg-1 cells (Supplementary Table 4). Of these insertions, one was in an exon (*Hells*), one in the 3′UTR (*Fasl*) and one in the 5′UTR (*Anapc11*) of a gene. We found eight intronic insertions and the rest (45) were intergenic.

**Generating cloaked human ESCs (hESCs)**

We next sought to engineer cloaked human pluripotent cells by transgenic overexpression of the functional human orthologues of the mouse immunomodulatory factors that were expressed in the Klg cell lines. Seven of the eight immune genes have direct orthologues (*PDL1*, *CD200*, *CD47*, *FASL, SERPINB9*, *CCL21*, *MFGE8*), while mouse *H2-M3* was replaced with human *HLA-G*, which is known to similarly suppress NK and T cells as a non-classical HLA molecule and is normally upregulated at the maternal–fetal interface[50–53]. The coding sequences of these human factors were cloned into individual transposon expression vectors, including both piggyBac and Sleeping Beauty to allow for serial transfection, and contained a CAG promoter and a drug or fluorescent selectable marker. These vectors were then transfected into a FS H1 hESC line[36] in two serial rounds (Extended Data Fig. 5a) and polyclonal pools of cells with high expression levels of all eight factors were generated by drug selection and FACS analysis (Fig. 4a and Extended Data Fig. 5b). We then generated single-cell-derived clonal lines and measured the expression of each factor by RT–qPCR (Fig. 4b and Extended Data Fig. 5c). We selected several clones to verify expression at the protein level by flow cytometry using antibodies

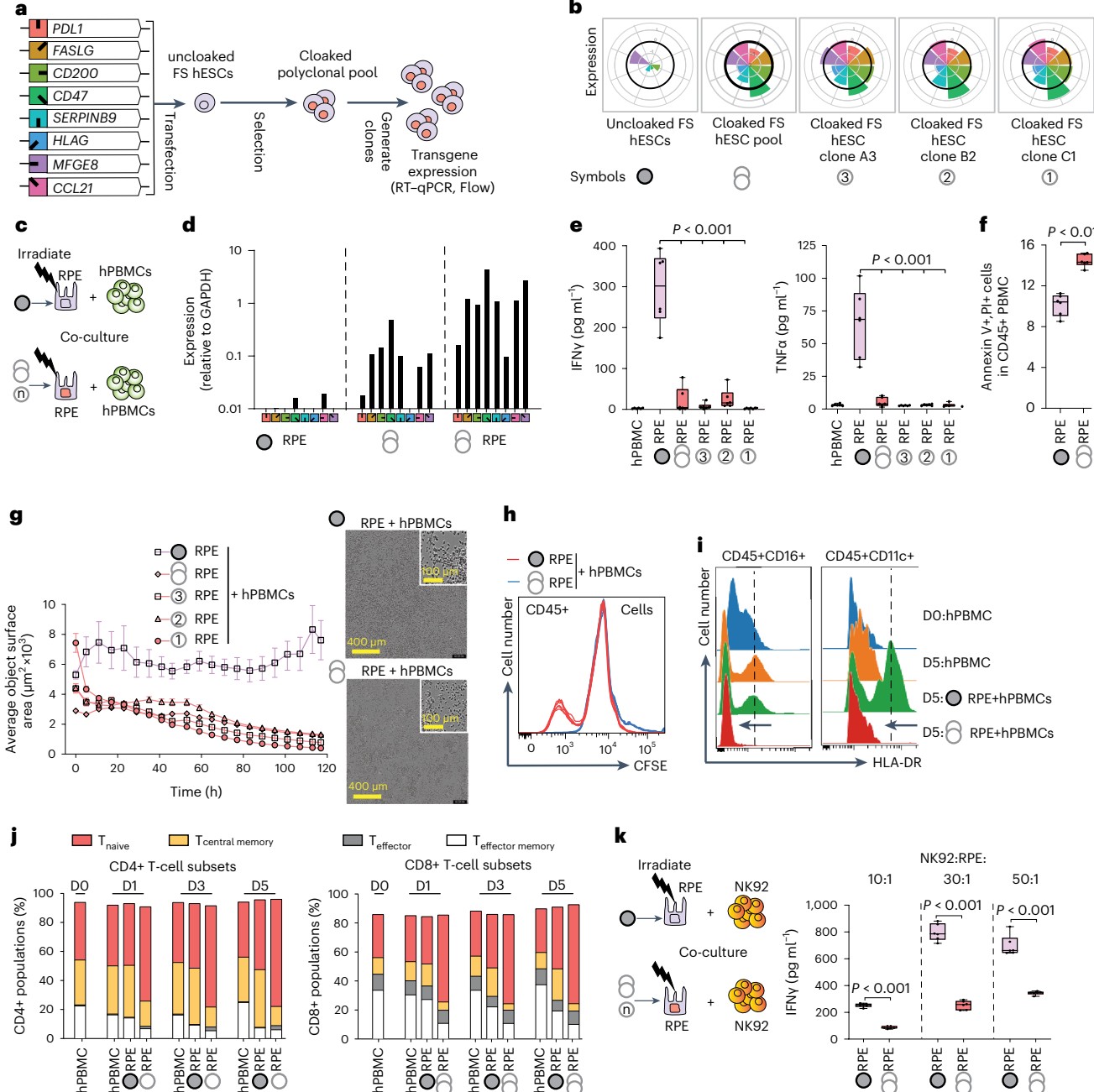

**Fig. 4 | Generation of cloaked hESCs and their properties. a**, Schema showing the generation of hESCs (containing the FS system), which overexpress eight human immunomodulatory transgenes ('cloaked') via insertion of transposon expression vectors. **b**, Expression level of each transgene was determined by RT–qPCR. Values shown are normalized to hPBMCs stimulated in vitro with αCD3ε and αCD28 antibodies (thick black line on each radial graph). The symbols at the bottom are used to designate the corresponding cell lines in further studies. **c**, Experimental design to characterize the interaction between cloaked RPE cells and hPBMCs. Uncloaked or cloaked hESCs were differentiated into RPEs, irradiated and then co-cultured with hPBMCs for use in downstream experiments (Fig. 4d–j). **d**, Expression of immunomodulatory transgenes relative to GAPDH by qPCR before and after RPE differentiation. The symbols correspond to the definitions in **a** and **b. e**, IFNg and TNFa levels in supernatants after 5 days co-culture of RPE cells and hPBMCs (boxes represent the 25th to 75th percentiles with median, whiskers represent the minimum–maximum; $n = 6$ independent wells with 3 independent experiments; two-tailed Student's $t$-test). **f**, Levels of apoptosis in hPBMCs as measured by Annexin and PI after 24 hours co-culture with uncloaked or cloaked RPEs. **g**, Cell surface area in hPBMC + RPE co-cultures measured by phase-contrast microscopy (IncuCyte system) as a correlate of cell

survival or death over 5 days. Images were taken every 6 hours (mean ± s.e.m., $n = 6$ independent wells for each group; average of 5 fields per well). Representative wells with individual phase-contrast field (upper right corner, outlined in white) are shown. **h**, Proliferation of hPBMCs in co-cultures as measured by CFSE. **i**, Two independent experiments showed MHC class II (HLA-DR) expression on monocytes (CD45+CD16+) and dendritic cells (CD45+CD11c+) of hPBMC after 5 days of co-culture with uncloaked versus cloaked RPEs. Dotted line indicates the mean peak intensity of HLA-DR expression at D5 of hPBMC co-culture with uncloaked RPE. **j**, Changes to the immune cell population and phenotyping using CyTOF after co-culture of uncloaked or cloaked RPEs with hPBMCs for 1, 3 and 5 days. Analysis of CD4+CD8+ T-cell (CD45+CD3+) subsets including $T_{naive}$ (CCR7+CD45RA+CD45RO−), $T_{effector}$ (CCR7−CD45RA+CD45RO−), $T_{central memory}$ (CCR7+CD45RA−CD45RO+) and $T_{effector memory}$ (CCR7−CD45RA−CD45RO+) cells. **k**, The experimental design (left) and IFNg in supernatant (right) after 3 days co-culture at varying ratios of uncloaked or cloaked RPEs with the NK92 cell line (boxes indicate the 25th to 75th percentiles with median, whiskers represent the minimum–maximum; $n = 6$ independent wells with 3 independent experiments; two-tailed Student's $t$-test).

against 5 of the 8 factors. We measured the expression of SERPINB9, MFGE8 and CCL21 using the linked fluorescent protein reporters as well as intracellular FACS staining (Extended Data Fig. 5d,e). Seven of the eight factors were robustly expressed, while HLA-G was only modestly increased relative to the endogenous level in untransfected ES cells. All cloaked hESC lines expressed comparable levels of *OCT4* and *NANOG* to that of uncloaked H1 hESCs (Extended Data Fig. 5f). Cloaked hESCs also upregulated HLA class I at comparable levels to uncloaked hESCs upon IFNg stimulation (Extended Data Fig. 5g) and could be differentiated into clinically relevant cell types including retinal pigmented epithelial (RPE) cells, cardiomyocytes, definitive endoderm and endothelial cells (Extended Data Fig. 6a–f).

We then tested how allogeneic human PBMCs (hPBMCs) respond to a clinically relevant cell type such as RPEs that were differentiated from cloaked FS hESCs (Fig. 4c and Extended Data Fig. 6a,b). We differentiated RPEs starting either from a polyclonal pool of cloaked hESCs, or from cloaked clonal hESC lines (A3, B2 and C1). We first verified that these RPEs maintained overexpression of all 8 cloaking transgenes after differentiation (Fig. 4d) and, interestingly, found that the levels of all transgenes in the RPEs were increased compared with the cloaked hESCs. Cloaked RPEs also upregulated similar levels of HLA I antigen after IFNg stimulation compared with uncloaked RPEs (Extended Data Fig. 6c). Next, we irradiated and co-cultured uncloaked or cloaked RPEs with allogeneic hPBMC for 5 days. While uncloaked RPEs induced a robust IFNg and TNFa response, this upregulation was almost completely abrogated when using RPEs differentiated from cloaked hESCs (Fig. 4). Compared with uncloaked RPEs, cloaked RPEs also induced higher levels of cell death detected by Annexin V and propidium iodide (PI) positivity in the CD45+ hPBMC compartment (Fig. 4f and Extended Data Fig. 7a) and resulted in a significant and gradual loss of total cell number in co-cultures as measured by total cell object surface area on the culture dish (Fig. 4g). We did not observe significant intrinsic differences in growth and survival of the different uncloaked and cloaked RPEs cultured without PBMCs (Extended Data Fig. 7b). Together, these findings suggest that cloaked RPEs do not activate allogeneic PBMCs.

Fitting with the lack of an inflammatory response to cloaked RPEs, we did not detect any proliferation of hPBMCs in co-cultures with cloaked RPEs using a cell tracer dye, carboxyfluorescein succinimidyl ester (CFSE), but did find proliferation in co-cultures with FS-only RPEs (Fig. 4h). Furthermore, while co-culture with FS-only RPEs caused an upregulation of the MHC HLA-DR class II protein among antigen-presenting cells including monocytes (CD45+CD16+) and dendritic (CD45+CD11c+) cells, this upregulation was completely abrogated in co-cultures with cloaked RPEs (Fig. 4i). We also measured the proportion of various CD3+CD4+ and CD3+CD8+ T-cell subsets including naïve, central memory, effector and effector memory phenotypes by Cytometry by Time-of-Flight (CyTOF) analysis and found that over time, co-culture with cloaked RPEs led to an enrichment of naive T-cell subsets and a depletion of central T and effector memory T-cell subsets (Fig. 4j). Since memory T cells express FAS receptors[54], it is possible that their depletion results from FAS activation by FASL, which is highly expressed on cloaked RPEs as a transgene (Fig. 4d). Finally, to examine the effect of cloaking on NK cells, we co-cultured uncloaked or cloaked RPEs with the NK92 cell line and saw a major abrogation of IFNg release, even up to very high ratios of 50:1 NK92:RPEs (Fig. 4k). Together, these co-culture experiments show that cells differentiated from human pluripotent cells, which have forced expression of the selected eight immunomodulatory genes, can block the induction of critical inflammatory cytokines, as well as the induction or activity of critical immune cells such as T cells, antigen-presenting cells and NK cells.

We next developed an in vitro system to model interactions between cloaked cells and hPBMCs. For this, FS-only or cloaked FS hESCs were differentiated into endothelial cells (ECs) and loaded into a microfluidic device such that the ECs coated the interior walls of the microchannels of the device to mimic the lumen of a blood vessel

(Extended Data Fig. 8a,b and Supplementary Data Fig. 1a,b). The system was connected to a reservoir of culture media that continually circulates through the device, with the reservoir containing an inlet to deliver tracer-labelled PBMCs and cell viability dyes such as DAPI to monitor the PBMC–EC interaction and the kinetics of cell death by confocal microscopy. The ECs could be visualized by the presence of enhanced green fluorescent protein (eGFP), which is co-expressed with the FS system from the *CDK1* gene and is present in both cloaked and uncloaked lines. In the human cloaked cells, eGFP is also expressed on the integrated, MFGE8-encoding transcript (Extended Data Fig. 5a).

When hPBMCs were loaded into the circulation of the microfluidic device coated with FS-only ECs, there was a rapid and increasing induction of cell death as measured by the uptake of DAPI into coated cells over 10 hours (Extended Data Fig. 8c left two columns and Extended Data Fig. 8d) and a corresponding loss of eGFP signal detected (Extended Data Fig. 8e). However, when the same hPBMCs were loaded into a device coated with cloaked FS ECs, there was almost no DAPI uptake in coated ECs (Extended Data Fig. 8c right two columns and Extended Data Fig. 8d) and only a very modest reduction of eGFP signal detected (Extended Data Fig. 8e). Together, the results support our previous findings and suggest that cells differentiated from cloaked hESCs can resist the killing of circulating allogeneic hPBMCs.

## Tissues formed by cloaked mESCs are immune-privileged and can host and protect both allogeneic and xenogeneic cells

On the basis of the observation that cloaking transgene expression and protein levels remained extremely high in long-term, allo-accepted dormant Klg-1 transplants (Fig. 3), we asked whether these tissues could protect internally transplanted uncloaked cells. To test this, we first formed dormant Klg-1 subcutaneous transplants in FVB recipients (Fig. 5a, red dashed boxes in schema). Then, uncloaked, luciferase transgenic, B6 FS mESCs, which are allogeneic with respect to the recipient, were injected into the dormant cloaked transplants and followed for 40 days by BLI (Fig. 5a, red solid boxes in scheme and BLI images, with one representative transplant recipient shown from 2 separate recipients). The injected uncloaked cells not only survived but continued to proliferate inside the cloaked tissue. Eventually these recipients required additional doses of GCV to induce the injected cells into dormancy. To show that this effect was due to the local, immune-privileged nature of the Klg-1 transplant itself, we then injected the same mESCs subcutaneously into the contralateral flank of the same FVB recipient (Fig. 5a, blue boxes in schema and BLI images). Clearance of these cells started within 5 days and when measured again 30 days after injection, no BLI signal could be detected. However, the uncloaked FS mESC-derived cells that had been injected into the Klg-1 tissue persisted throughout, even while the host cleared the cells on the contralateral flank.

We then asked whether the Klg-1 ectopic transplant could protect xenogeneic cells. For these experiments, we used luciferase-expressing hESCs which also had the FS system (FS hESC[luc+]). These cells survive long term when subcutaneously injected into immune-deficient NOD-scid-gamma (NSG) recipients (Extended Data Fig. 9a) but were rapidly cleared in wild-type B6 recipients (Extended Data Fig. 9b). After establishing long-term-accepted, dormant uncloaked FS or Klg-1 ectopic transplants in isogeneic (B6) recipients, we then injected FS hESC[luc+] into the transplants (*n* = 6 recipients in 2 experiments) and tracked the injected cells by BLI (Fig. 5b). FS hESC[luc+] that were injected into uncloaked FS transplants were all cleared between ~20 and 30 days (Fig. 5b, red solid boxes and lines). However, those injected into dormant Klg-1 transplants were all accepted long term (Fig. 5c, red dashed lines and boxes), with all recipients being monitored for almost 4 months and two monitored for 240 days. These persisting FS hESC[luc+] cells could also be seen in the excised Klg-1 tissue by ex vivo BLI (Fig. 5c, images on right side, one representative teratoma shown). Similar to our allogeneic mouse experiments, we injected FS hESCs[luc+]

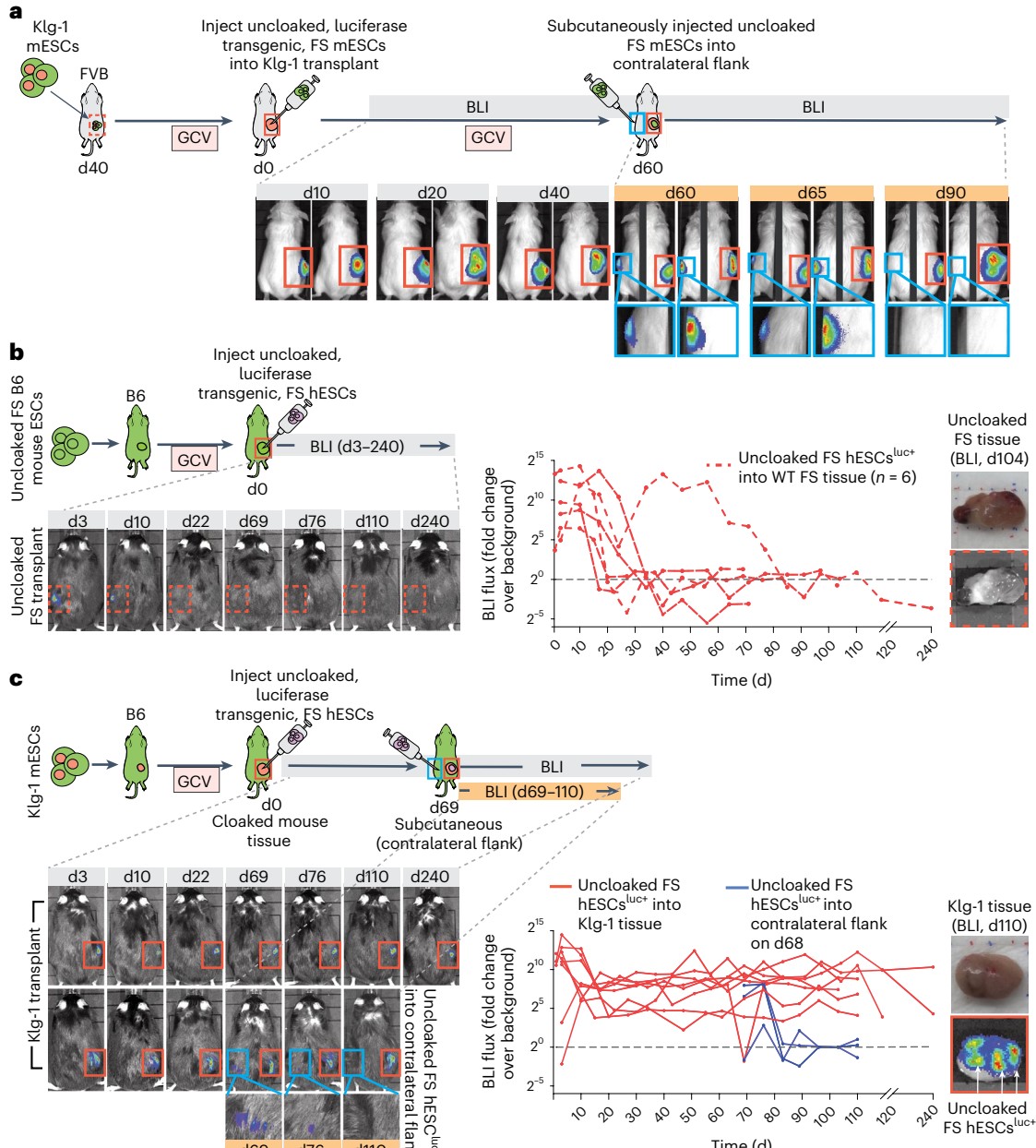

**Fig. 5 | Dormant transplants formed from cloaked mESCs (Klg-1) serve as immune-privileged sites that protect allogeneic and xenogeneic transplants.** **a**, Klg-1 dormant transplants were established in FVB and then, as secondary transplants, $5 \times 10^6$ uncloaked FS luciferase transgenic B6 mESCs were injected into the primary transplant and tracked with BLI for 40 days (red solid boxes, d0–40). On day 60, as a tertiary transplant, $5 \times 10^6$ uncloaked FS mESCs were injected subcutaneously into the contralateral flank (blue boxes) and tracked with BLI for 30 days (d60–90). BLI exposure right side, 10 s; left side, 1 min. $n = 2$ mice. **b**, Left: a dormant uncloaked transplant was established in isogeneic B6 mice and, as a secondary transplant, $10^7$ luciferase transgenic FS hESCs were injected into uncloaked transplant. Right: the human cells were followed with BLI (dashed red boxes and lines, d0–240). **c**, Left: a dormant Klg-1 (cloaked) transplant was established in B6 mice and, as a secondary transplant, $10^7$ luciferase transgenic

FS hESCs were injected into the cloaked transplant. Right: the human cells were followed with BLI (red boxes and lines, d0–240). On day 68, as a tertiary transplant, $10^7$ luciferase transgenic FS hESCs were injected into the contralateral flank (blue boxes and lines) of mice with Klg-1 tissue and followed for 42 days by BLI (d68–110). The hosting Klg-1 tissues also expressed luciferase but at lower levels than FS hESCs cells and cannot be detected using short 10 second exposure times. The BLI images in **b** and **c** show one representative mouse in each group. BLI values of all mice are in p $\text{s}^{-1}$. Black dashed lines, average background BLI. Uncloaked FS group, $n = 6$ mice; Klg-1 group, $n = 7$ mice. In **b** and **c**, images on far right show uncloaked (**b**) or Klg-1 (**c**) transplants excised at day 110 and imaged by BLI immediately after intraperitoneal injection of luciferin into recipients. White arrows in **c** show the presence of transplanted Luc+ hESCs in excised Klg-1 tissue.

subcutaneously into the contralateral flank of 3 recipients that had already accepted FS hESCs$^{\text{luc+}}$ in the Klg-1 transplants 68 days earlier. These cells were rapidly cleared (Fig. 5c, blue solid boxes and lines), even while the hESCs$^{\text{luc+}}$ within the Klg-1 dormant transplants in the same recipients persisted.

In a final extension of these experiments, we sought to show the relevance of an immune-privileged tissue in a disease setting. In a small pilot experiment ($n = 3$ mice per group), we tested whether a Klg-1 transplant could protect allogeneic wild-type C3H islets and reverse hyperglycaemia in a streptozotocin (STZ)-induced model of

type 1 diabetes. In healthy FVB mice, the concentration of glucose in the blood remains stably within 4 to 11 mmol l$^{-1}$, which is considered a normoglycaemic range (Extended Data Fig. 10a). However, intraperitoneal injection with STZ selectively kills insulin-producing cells in the pancreas and leads to rapid hyperglycaemia with blood glucose levels reaching 20–25 mmol l$^{-1}$ within 7 days (Extended Data Fig. 10b, red highlighted area). When islets isolated from C3H recipients were transplanted underneath the kidney capsule of these STZ-treated FVB recipients at the peak of their hyperglycaemia, they quickly returned to normoglycaemia 5–10 days later (Extended Data Fig. 10b). However, these same transplant recipients again developed hyperglycaemia ~30 days after transplantation, as expected due to eventual rejection of the foreign donor islets.

To test whether Klg-1 tissue could protect allogeneic islets, we first generated uncloaked FS or Klg-1 ectopic transplants in B6 mice and then injected the mice with STZ. After the mice developed hyperglycaemia, we grafted islets from CH3 donors directly into the dormant transplants (Extended Data Fig. 10c,d). C3H islets restored normoglycaemia, but the uncloaked group returned to hyperglycaemia 3–4 weeks later (Extended Data Fig. 10c). However, those grafted into Klg-1 transplants maintained normoglycaemia and only reverted to hyperglycaemia when the entire Klg-1 tissue was excised either 60 or 120 days later (Extended Data Fig. 10d). Together, these series of experiments showed that the Klg-1 ectopic tissue is an immune-privileged site that can host and protect allogeneic and xenogeneic cells, as well as functional therapeutic tissues such as islets.

## Discussion

Many strategies to engineer immune-evasive allogeneic cells involve removing genes and/or introducing one or more modulatory factors[10–12,55]. MHC genes encode the most important source of allogeneic antigens and loss of these molecules is often coincident with immune evasion in nature[56]. We show here that overexpression of eight selected immunomodulatory factors in cells is sufficient to escape immune rejection despite the presence of allogeneic MHC molecules. Since our cells retain MHC, they may still be able to present viral and bacterial antigens. Moving forward it will be important to study how these engineered cells, as well as those coming from other editing strategies including MHC knockout, respond to infection. This approach is compatible with inducible transgene expression systems such as TetON[57–59], which could be turned off to clear allogeneic cells if needed.

Our approach was motivated by an appreciation of the highly redundant and adaptive nature of mammalian immunity, which evolved in an arms race with malignancies, viruses, bacteria and helminths and should necessarily be complex. Most examples of immune escape in nature require many loss- or gain-of-function adaptations. These include those where MHC expression is retained, including some transmissible cancers[23,24], as well as the placenta which expresses HLA-C, -G and -E in the extravillous cytotrophoblast[60–62].

The immunomodulatory factors used in our study interfere with many key immune pathways, including antigen presentation and the activation of adaptive immunity, guards against cytotoxic immune cells, NK- and T-cell-mediated lysis, macrophage phagocytosis and general inflammatory responses. These cells and pathways are highly cross-regulated and targeting them simultaneously may support mutually reinforcing effects. This may be an important property of cells that can escape innate, cellular and humoral immunity over time.

The allo-accepted mESCs in this study were derived with modest in vitro and in vivo screening after piggyBac and Sleeping Beauty transposon-mediated transgenesis. While many approaches rely on random integration of immune factors, our data clearly show the huge variation in expression levels among different clones. Our data also show that selecting engineered clones with high transgene expression may be as critical as the function of the factors themselves. Strong promoter elements and suitable transgene delivery methods may

reduce the amount of in vitro and in vivo screening needed to isolate a particular allo-accepted clone.

We used the design guidelines learned from the mouse when generating our cloaked hESCs lines. These human lines were created using functional orthologues of the tested mouse transgene, with seven of the eight having direct orthologues and H2-M3 being replaced by the functional equivalent HLA-G[39]. Given that cloaking in human contexts will probably be more difficult than in rodents, ensuring high transgene expression levels in any engineered product may be important. Under the right conditions, transposon integration can be extremely efficient and robust. We were able to co-transfect multiple vectors, each encoding a single factor, and quickly isolate cells with integration of all transfected factors. Furthermore, the multicopy and multisite integration nature of this approach leads to at least some cells, which can be isolated and propagated, with very high expression of all intended factors. The multiple integrations may mitigate the site-specific silencing and expression variegation that can occur after cell differentiations.

Of course, random transgene integration could disrupt endogenous gene activity or activate oncogenes. It is also true that allo-accepted cells are inherently dangerous because they compromise the immune system's ability to survey and eliminate infected or potentially tumorigenic cells of graft origin. For these reasons, cloaked cells should contain a tightly controlled safety switch such as our FS system[36,37]. Variations on this system are also possible using different suicide genes and/or linking suicide genes to different endogenous genes, such as those that are always expressed by the cell and not dependent on cell division; for instance, genes needed for transcription or translation. The introduction of our FS system is also critical as it allows for a mathematical quantification of risk[36], which will allow patients and clinicians to decide the risk-to-benefit ratio of future cell therapies, especially those coming from universal cell products.

Other approaches to protect unmodified allogeneic donor cells from host immunity involve encapsulation in permeable or semipermeable devices[63,64]. This is the case in the current trials that use encapsulation to protect transplanted allogeneic islets in patients with type 1 diabetes[65,66]. While promising, it is not clear whether these devices will be able to host functional and allogeneic islets long term without immune suppression, given the known complications involving vascularization, fibrosis, foreign body responses and/or immune responses. While coating and/or delivering allogeneic cells with immunosuppressant biomaterials is also a highly active field[67], these techniques are limited by the half-life and non-renewability of the delivered factors.

Here, we also show that our approach for generating allo-accepted pluripotent cells, combined with our FS system, can be used to create a stable, 'artificial' and immune-privileged tissue. To do this, we only need to activate the FS system at an appropriate time after cell injection and while the teratoma, which is naturally well-encapsulated, is growing. This ultimately kills only the proliferating cells in the tissue, while leaving the already formed, terminally differentiated cells intact. Once stabilized, this tissue can protect not only internally transplanted allogeneic uncloaked mouse cells but also xenogeneic human cells in immunocompetent recipients. The tissue's ability to protect xenogeneic cells strongly indicates the potential of immune protection despite high levels of antigenic mismatch between donor and host.

The living immune-privileged tissue we generated is well vascularized and actively prevents the generation of immune responses via genomically encoded immunomodulatory factors. We hypothesize that the action of the overexpressed transgenes confers the immune-privileged status of these tissues. However, beyond an understanding about the function of these cloaking factors in the rejection of non-self cells, we have not yet explored specific tolerogenic mechanisms.

Alternatively to a teratoma, it may be possible to generate an immune-privileged niche from a transplanted, well-defined cell type that is first produced by terminal differentiation in vitro before

engraftment. Although not tested in this work, this approach could allow for the generation of a niche in a more reliable, predictable way. Ultimately, such immune-privileged cells or tissues could host therapeutic cells with paracrine effects to treat hormone or factor deficiencies, make antibodies for passive immunization, or produce other useful biologics in many disease settings. Engraftment of therapeutic cells into the ectopic tissue in our work required only a superficial surgery.

Of course, the safety of these sites must be carefully considered. The FS system that reliably renders the tissue into a dormant state will be necessary. Our mouse experiments show that such pluripotent cell-derived tissues are inert and non-metastatic when formed under the skin. It can be easily excised if, in an unlikely event, the cells escape the suicide system or if the engrafted or hosting cells become aberrant or are no longer needed.

The data presented here demonstrate the long-term survival of a solid tissue allograft in fully immunocompetent hosts without the need for systemic immunosuppression. The same mechanisms that allow our cloaked cells to be allo-accepted across MHC barriers also confer an immune-privileged status to the derived solid tissues. If translated to a human setting, our approach could satisfy the need for a single pluripotent cell source to serve a broad and genetically diverse population, ultimately allowing for cell therapies to use of off-the-shelf cell products without the need for immunosuppressive drugs.

## Methods

### Mice

C57BL/6N (strain 005304), C3H/HeJ (strain 000659), FVB/NJ (strain 001800), BALB/cJ (strain 000651) and NSG mice (stock 005557) were purchased from the Jackson Laboratory. CD-1 (stock 022) mice were purchased from Charles River. Mice (6–20-week-old) of each strain/background were used for teratoma assays. Mice were housed in a pathogen-free facility at the Toronto Centre for Phenogenomics (TCP) in micro-isolator cages (Techniplast) with individual ventilation at a maximum of 5 mice per cage and on a 12 h light/dark cycle. All mouse procedures were performed in compliance with the Animals for Research Act of Ontario and the Guidelines of the Canadian Council on Animal Care. Animal protocols performed in this study were approved by the TCP Animal Care Committee. The number of animals used in experiments was determined in accordance with similar studies in the field; due to the nature of most experiments, blinding was impossible since the results are visible.

### Construction of piggyBac and Sleeping Beauty transposon expression vectors

Plasmids containing the CDS of mouse/human m*Pdl1/hPDL1*, *mFasl/hFASLG*, *mCd47/hCD47*, *H2-M3/hHLA-G*, *mSerbinb9/hSERPINB9*, *mMfge8/hMFGE8*, *mCcl21/hCCL21* and *hCD200* were obtained from the Lunenfeld–Tanenbaum Research Institute Open freezer repository (Toronto, Canada). The plasmid containing the CDS of murine *Cd200* was obtained from GE Dharmacon. The NCBI Refseq protein ID encoded by each CDS is listed in Supplementary Table 1. Each CDS was individually cloned into piggyBac (PB)[38–40] or Sleeping Beauty (SB)[41,42] transposon expression vectors using the Gateway cloning system (ThermoFisher, 12535029) following manufacturer protocol. The CDS of each transgene was amplified using PrimeStar HS mastermix (Takara, R040) using extension PCR that added gateway-compatible attb1/attb2 sites to the 5′ and 3′ ends of the amplicon. These amplicons were then recombined into the gateway pDONR221 vector (ThermoFisher, 12536017), and insertion of the transgenes was verified by Sanger sequencing. Finally, all transgene-containing entry clones where recombined into the gateway cloning sites contained in PB and SB expression vectors to generate the resultant plasmids used for transgene expression in cells. These PB and SB vectors drive expression of the CDS with the constitutively active CAG promoter and express a drug-resistance or fluorophore marker downstream of the immune gene CDS via an internal ribosomal entry site. At each cloning step, plasmids were transformed into chemically competent DH5alpha *E. coli* (ThermoFisher 18625-017) and colonies grown on LB agar plates containing kanamycin (Sigma, K1377) or ampicillin (Sigma, A9518). Colonies were expanded in LB broth (Wisent, 809-060-L), and the plasmid DNA was purified using the Presto Mini Plasmid DNA kit (Geneaid, PDH300). All sequencing was done by Sanger sequencing at The Centre for Applied Genomics, Hospital for Sick Children (Toronto, Ontario).

### Cell culture

mESCs were cultured in high-glucose DMEM (ThermoFischer (Gibco), 11960-044) + 15% fetal bovine serum (FBS) (Wisent, 0901150), leukaemia-inhibiting factor (LIF) produced at the Lunenfeld–Tanenbaum Research institute (Sinai Health Systems), 2 mM Glutamax (ThermoFisher (Gibco), 35050-061), 0.1 mM nonessential amino acids (ThermoFisher (Gibco), 11140-050), 1 mM sodium pyruvate (ThermoFisher (Gibco), 11360-070), 0.1 mM 2-mercaptoethanol (Sigma, M6250) and 50 µg ml$^{-1}$ penicillin/streptomycin (ThermoFisher (Gibco), 15149-122). Cells were maintained on a layer of mouse embryonic fibroblasts, mitotically inactivated by Mitomycin C (Sigma, M4287) and plated at a density of 40,000 cm$^{-2}$. Cultures were kept in incubators with 5% $CO_2$ at 37 °C and 100% humidity. Media were changed every day and cells subcultured when subconfluent every 2–3 days. Human ESCs were cultured on Geltrex (ThermoFischer (Gibco), A14133-02) coated plates in mTesR+ (StemCell Technologies, 85851) with 50 U ml$^{-1}$ penicillin/streptomycin (ThermoFischer (Gibco), 14140-122) and passaged using Accutase (StemCell Technologies, 07922) when 80% confluent. mTesR media were replaced every few days. Rock inhibitor (1 µM; rSelleckchem, S1049) was added for 24 h after a frozen vial of hESCs was thawed and when single-cell sorted. All mESC and hESC cultures, as well as any differentiated progeny in vitro, were routinely tested for *Mycoplasma*. Any cultures testing positive were discarded and not used for any experiment.

### Transfection, selection and cloning of mESCs and hESCs

mESCs or hESCs were transfected with piggyBac or Sleeping Beauty transposon expression vectors using JetPrime (Polyplus, 101000015) or Lipofectamine 3000 (ThermoFisher, L3000008) following manufacturer protocol, on adherent 6-well culture dishes at ~50–80% confluency, along with a plasmid encoding hyperactive piggyBac transposase (The Sanger Center, pCMV-hyPBase) or hyperactive Sleeping Beauty transposase (gift from Dr Zsuzsanna Izsvák, Max Delbrück Center for Molecular Medicine). After transfection, transgene-containing clones were isolated using combinations of drug selection with 150 µg ml$^{-1}$ G418 (ThermoFisher (Gibco), 10131035) or 1 µg ml$^{-1}$ puromycin (ThermoFisher (Gibco), A11138-03), single-cell sorting and/or bulk sorting using antibodies against transgene-encoded factors, and/or in vitro culture plating at clonal densities (200 cells per 100 mm dish). For drug selection, G418 or puromycin was added to the selection media 24 h after transfection and maintained until the establishment of antibiotic-resistant colonies. The transfection sequence, sorting and drug selection that led to transgene-containing clones are shown and described in Extended Data Fig. 1 for mESCs and Extended Data Fig. 5 for hESCs.

### RT–qPCR of immunomodulatory transgenes

RNA was isolated by removing the media and adding Trizol (ThermoFisher, 15596-018) for 5 min at room temperature, followed by 150 µl chloroform (Sigma, C2432). After briefly vortexing and centrifuging at high speed, the nucleic acid phase was collected, precipitated with 100% isopropanol, washed with 70% ethanol and eluted in RNase-free water. Complementary DNA (cDNA) was then made from isolated RNA with the QuantiTect reverse transcription kit (Qiagen, 205313). qPCR was performed using the SensiFast SYBR

No-Rox kit (Frogga Bio, BIO-98020) following manufacturer's protocol and run on a Bio-Rad C1000 CFX3 thermocycler. qPCR primers against all utilized mouse and human immune gene CDS, as well as hOCT4, hSOX17, hFOXA2 and all used housekeeping genes are listed in Supplementary Table 2.

## Activation of splenocytes

The spleen from a C57BL/6N mouse was manually homogenized in a 70 μm cell strainer (Falcon, 352350) in DMEM (ThermoFisher (Gibco), 11960-044). The single-cell suspension was washed with DMEM, lysed with red blood cell lysis buffer (Sigma, R7757-100) and washed again with DMEM. Isolated splenocytes were plated onto adherent culture dishes and activated with anti-mouse antibodies against CD3ε (1:100 dilution; BD Biosciences, 553058) and CD28 (1:100 dilution; BD Biosciences, 553295), as well as Golgi Stop (1:1,000 dilution; BD Biosciences, 554725) for 8 h in high-glucose DMEM (Invitrogen) + 15% FBS (Wisent, 0901150), 2 mM Glutamax (ThermoFisher (Gibco), 35050-061), 0.1 mM nonessential amino acids (ThermoFisher (Gibco), 11140-050), 1 mM sodium pyruvate (ThermoFisher (Gibco), 11360-0700), 0.1 mM 2-mercaptoethanol (Sigma, M6250) and 50 μg ml⁻¹ penicillin/streptomycin (ThermoFisher (Gibco), 15140-122). RNA was then isolated from activated cultures and used as control in RT−qPCR analysis of transgene-containing mESC clones.

## Total RNA-seq

mESC-derived tissues were excised, frozen on dry ice and homogenized with a TissueLyser II (Qiagen, 85300) following manufacturer protocol. RNA was then collected from the tissues, as well as mESCs grown in vitro, using the Qiagen RNeasy kit (Qiagen, 74104). cDNA was made from isolated RNA with the QuantiTect reverse transcription kit (Qiagen, 295313). RNA-seq was performed using an Illumina HiSeq 2500 system (The Center for Applied Genomics, Hospital for Sick Children). Trimmomatic[68] was used to trim residual Illumina adapters from reads, and the FastQC tool (https://www.bioinformatics.babraham.ac.uk/projects/fastqc/) was used for quality control. Trimmed reads were pseudo-aligned with kallisto[69] and further processed using DESeq2 (ref. [70]) in R (http://www.r-project.org/). The DESeq2 tool was used to produce rlog values.

## spPCR and analysis

spPCR was done on genomic DNA from Klg-1 mESCs using Primestar HS mastermix (Takara R040) and primers specific for the inverse terminal repeats of the piggyBac sequences flanking the immunomodulatory transgenes. HMSpAa (top strand primer) CGAAGAGTAACCGTTGCTAGGAGAGAC CGTGGCTGAATGAGACTG-GTGTCGACACTAGTGG; HMSpBb (lower strand primer/hairpin) gatcCCACTAGTGTCGACACCAGTCTCTAATTTTTTTTTTTCAAAAAAA

Primary PCR primers: PB-L-Sp1 (PB 5′TR sequence) GCGT-GCTTGTCAATGCGGTAAGTGTCACTG; PB-R-Sp1 (PB 3′TR sequence) CCTCGATATACAGACCGATAAAACACATGC. Secondary nested PCR primers: PB-L-Sp2 (PB 5′TR sequence) ACGCATGCATTCTTGAAATATT-GCTCTCTC; PB-R-Sp2 (PB 3′TR sequence) ACGCATGATTATCTTTAACG-TACGTCACAA. Amplified DNA was gel-extracted with the Monarch gel extraction kit (NEB T1020L) and sequenced by NGS. Residual Illumina primers, as well as piggyBac and spPCR primers were trimmed from reads using Trimmomatic[68]. Trimmed paired reads were then aligned with the end-to-end mode of STAR tool[71] and visualized in IGV[72,73]. High-quality reads were then filtered for only those beginning or ending with TTAA (the sequences at which piggyBac inserts into the genome[40]) and overlapping with the nearest read by a TTAA sequence. Ninety-three candidate regions of insertion were identified and then filtered to sort only those where paired reads were present on both sides of the TTAA region and without a TTAA in the middle of the read, ultimately resulting in 56 high-quality candidates, listed in Supplementary Table 4.

## BLI

Mice were injected intraperitoneally with 33 μl of 15 mg ml⁻¹ solution of luciferin (Xenolight D-Luciferin; Perkin Elmer, 122799) per 10 g bodyweight. At 15 min following injection, mice were anaesthetized with gaseous isoflurane and imaged with the IVIS Lumina imager (Perkin Elmer) at exposures between 10 s and 3 min. Binning was set to small and F/stop to 1. Images were acquired using Living Images software (Perkin Elmer). Short exposure times of 10 s were used to identify FS mESCs (which are luciferase^high) injected into Klg-1 tissue (which come from luciferase^low Klg-1 mESCs). Long exposure times were used to detect low-level signal in controls.

## In vivo mESCs injections and teratoma assay

Matrigel matrix high concentration (Corning, 354248) was diluted 1:1 in ice-cold DMEM and kept on ice. mESCs were treated with Trypsin-EDTA at 37 °C, resuspended in ESC media, centrifuged and washed once with PBS. Then, $5 \times 10^6$ mESCs were diluted in 100 μl of ice-cold Matrigel:DMEM mixture and injected subcutaneously at the back of the neck or each dorsal flank (two separate injections) for teratoma assays ($5 \times 10^6$ mESCs in 100 μl at each injection site). Mice were anaesthetized during injections with isofluorane to allow for the cell/Matrigel mixture to gel in place. Palpable teratomas developed 10–40 days after injection, depending on the strain. Teratoma size was measured with calipers and the volume calculated using the formula $V = (L \times W \times H)\pi/6$. At endpoint, mice were euthanized and the teratomas excised and prepared for immunohistochemistry and histology.

## Stabilization of teratomas with GCV

All mESCs used in these studies contain the FS system based on a thymidine kinase suicide gene targeted to the *CDK1* gene[36]. For long-term-acceptance studies, mice with mESC-derived teratomas were given ganciclovir (Cytovene) for a period of several weeks once teratomas reached an approximate volume of 400–500 mm³, depending on the strain. Any mice in which teratoma growth resumed at a later date after GCV withdrawal was given GCV again for several days. Ganciclovir administration was given daily or every other day as intraperitoneal injections at a dose of 50 mg kg⁻¹ in 100 μl PBS.

## Single-cell sorting and flow cytometry analysis

**Cell sorting.** To generate the Klg-1 and Klg-2 lines, transfected mESCs were collected at 80% confluency, washed with FACS buffer (PBS + 1% BSA and 0.5% EDTA) and then sorted on the basis of eGFP intensity (Klg-1), or stained in FACS buffer with primary antibodies against mouse PD-L1 (alexa488; Novus, NBP1-43262AF488), CD47 (BV421; BD Biosciences, 740055), CD200 (alexa647; BD Biosciences, 565544) and FASL (Novus, NBP1-97519) to generate Klg-2. Primary antibodies against mFASL were detected with anti-rat Fc Alexa 568 secondary antibody (ThermoFisher, A21112).

To generate cloaked hESCs, transfected cells were collected at 80% confluency, washed and stained in FACS buffer with antibodies against PDL1 (APC; Biolegend, 329708), FALSG (PE; Biolegend, 306406), CD200 (APC/Cy7; Biolegend, 329220) and CD47 (PerCP/Cy5; Biolegend, 323109). In the second round of transfection, hESCs were sorted on the basis of mCherry, eGFP and E2-crimson fluorescence encoded on the vectors containing the cloaking transgenes. For FACS analysis, all cells were washed with FACS buffer before and after antibody staining. Antibodies against human SERPINB9 (ThermoFisher, MA5-17648), CCL21 (R&D, AF366-SP) and MFGE8 (R&D, IC27671A) were used to verify transgene expression in hESCs and detected with secondary antibodies against mouse IgG (a488; ThermoFisher, A10680) or goat IgG (a488; ThermoFisher, A27012). Cell suspensions from hESC-derived teratomas or in vitro co-cultures were stained with antibodies against human CD45 (APC; BD Biosciences, 560973) and CD3 (Pe-Cy5; BD Biosciences, 561006). Antibodies against HLA-A,B,C (APC; Biolegend,311409) were used against hESCs and RPEs. Antibodies against human CD31 (Pe-Cy7; Biolegend, 303118) and CD144

(PE; Biolegend, 138009) were used to detect differentiated endothelial cells. Annexin V (BV421; Biolegend, 640923) and PI (Biolegend, 421301) were used to detect apoptotic and/or necrotic hPBMCs in hPBMC+RPE (retinal pigmented epithelial) co-cultures. All antibody cocktails contained human TruStain FxC Fc blocker, and a viability dye was added before running on a Beckman Gallios flow cytometer in FACS buffer. For both cell sorting and FACS analysis, mouse and human antibody cocktails contained either anti-mouse CD16/CD32 Fc blocker (BD Biosciences, 553141) or Human TruStain FxC Fc blocker (Biolegend, 422301). After antibody staining, both mouse and human cells were washed several times in FACS buffer, then sorted using a MoFlo Astrios EQ cell sorter (Beckman Coulter) in FACs buffer + 25 mM HEPES (Sigma, H3537) at the Lunenfeld–Tanenbaum Research Institute flow cytometry core; data were analysed using Summit v.6.2 (Beckman Coulter) and Kaluza v.1.5 (Beckman Coulter). For cell-sorted samples, the cells were collected in the appropriate mouse or human culture media before plating onto treated culture dishes. Viability dye such as DAPI (ThermoFisher, 62248) was used to discriminate live versus dead, and doublets excluded by forward/side scatter. Uncloaked and untransfected cells stained with the same antibodies, as well as isotype controls, were used as negative controls and for instrument compensation when appropriate. For flow cytometry analysis only, stained cells were washed in FACS buffer alone, run on a Beckman Gallios flow cytometer and analysed with Kaluza v.1.5 (Beckman coulter) or Flowjo v.10.

### Fluorescent immunohistochemistry

mESCs were grown on mitotically inactivated mouse embryonic fibroblasts plated onto coverslips coated with gelatin (StemCell Technologies, 07903). Coverslips were fixed with 4% paraformaldehyde (PFA) (Sigma, 1004960700), washed with PBS and incubated with 100 mM glycine (Sigma, G7126) at 4 °C in PBS with 0.02% sodium azide (Sigma, 71289). Coverslips were then blocked with PBS + 10% donkey serum, 0.01% triton and anti-mouse CD16/CD32 (BD Biosciences, 553141), and stained with primary antibodies in PBS + 10% donkey serum. For immunohistochemistry of mESC-derived tissue, sections were fixed in 4% PFA at room temperature for 24–48 h, dehydrated in 25% sucrose in PBS, embedded in OCT compound and cryo-sectioned. Tissue sections were fixed in cold acetone for 10 min, washed with PBS, blocked in PBS + 3% BSA, 10% goat serum and anti-mouse CD16/CD32, and then stained with primary antibodies overnight at 4 °C in blocking buffer. For both mESCs on coverslips and tissue sections, primary antibodies were used against mouse PD-L1 (Novus, NBP1-43262AF488), CD47 (BD Biosciences, 740055), CD200 (BD Biosciences, 565544), H2-M3 (BD Biosciences, 551769), FASL (Novus, NBP1-97519), SERPINB9 (HycultBiotech, HP8035), CCL21 (R&D, AF457) and MFGE8 (Biolegend, 518603). Additional primary antibodies used on in vitro-cultured cells included those against mouse CD31 (Biolegend, 102417), neurofilament (Fisher Scientific, MS359R7), SSEA1 (Sigma, MAB4301), OCT4 (Santa Cruz, sc-8628), smooth muscle actin (Abcam, ab5694), FOXA2 (Abcam, ab108422), SOX17 (R&D, AF1924), troponin (cTNT; abcam, ab8295) and actinin. Other primary antibodies used on tissue sections included anti-mouse H-2Kb (Biolegend, 116511) and H-2Kq (Biolegend, 115106). Primary antibodies against H2-M3, FASL, SERPINB9, CCl21, MFGE8, neurofilament, OCT4, SSEA1, smooth muscle actin, FOXA2 and SOX17 were detected with secondary antibodies conjugated to Dylight 488 (Biolegend, 405503), DyLight 549 (Novus, NBP1-72975) or Alexa647 (ThermoFisher, 21244) by staining in PBS + blocking buffer at room temperature. mESCs on coverslips and tissue sections were stained with DAPI (ThermoFisher, 62248) and mounted with Immunomount (ThermoFisher, 9990402). Images were visualized with a Zeiss LSM780 confocal microscope.

### Histology

Paraffin embedding, tissue sectioning, and hematoxylin and eosin (H&E) staining were done by staff at the Centre for Phenogenomics Histology laboratory (Toronto).

### Generation of EBs and spontaneous differentiation

To generate EBs, mESCs were collected and plated in suspension on non-adherent tissue cultures plates (Corning, ULC 3471) in mESC media without LIF and FBS (Wisent, 0901150), with concentration reduced to 10%. To induce spontaneous differentiation, EBs were then collected and plated onto adherent tissue culture plates that were coated with Gelatin (StemCell Technologies, 07903). The presence of cardiomyocytes, as well as cells containing neurofilament and muscle actin was then confirmed within the cells that proliferated and differentiated out from attached EBs.

### Differentiation of mESCs into cardiomyocytes

mESCs were maintained in serum-free (SF) mouse ESC culture medium: 1:1 neurobasal media (ThermoFisher, 21103049) and DMEM/F12 (ThermoFisher, A4192001) supplemented with 0.05% bovine serum albumin (ThermoFisher (Gibco), 15260037), 0.5X B27 (ThermoFisher, 17504044), 0.5X N2 (ThermoFisher, 17502001), penicillin/streptomycin, LIF, 200 µg ml⁻¹ BMP4 (R&D, 314-BP-050/CF) and 0.004% v/v Monothioglycerol (Sigma, M6145), and maintained on mitotically inactivated mouse embryonic fibroblasts. On day 0, mESCs were dissociated with TrypLE (ThermoFisher, 12604013) and aggregated in SF medium with LIF. After 48 h, the EBs were dissociated with TrypLE and cultured on Matrigel-coated 24-well cell culture plate in SF medium with LIF for 8 h, followed by media replacement with SF differentiation medium without LIF for 24 h: 1:3 IMDM medium (ThermoFisher, 12440053) and HAM's F12 (ThermoFisher, 21127022) supplemented with glutamine (ThermoFisher, A1286001), 0.05% bovine serum albumin (Sigma, A9418), 0.5X B27 (ThermoFisher, A1895601), 0.5X N2 (ThermoFisher, 17502001), penicillin/streptomycin, 0.004% v/v MTG (Sigma, M6145) and 30 µg ml⁻¹ ascorbic acid (Sigma, A4403).

On day 3, 5 ng ml⁻¹ of vascular endothelial growth factor (VEGF) (PeproTech, 450-32), 0.25 ng ml⁻¹ BMP4 (R&D, 314-BP-050/CF) and 5 ng ml⁻¹ Activin A (R&D, 338-AC-050/CF) were added to the SF differentiation medium. On day 4, the differentiation medium was replaced with StemPro 34 medium (ThermoFisher, 10639011), 5 ng ml⁻¹ VEGF (PeproTech, 450-32), 10 ng ml⁻¹ bFGF (PeproTech, 100-18B) and 50 ng ml⁻¹ FGF10 (PeproTech, 100-26). The medium was changed every 60 h. On day 15, the contractile mESC-derived cardiomyocytes were examined for contractility and calcium transients. A custom MatLab code was developed to analyse the images of the contractile cardiomyocytes. Immunohistochemistry was performed to stain cells with the primary anti-cardiac troponin-T antibody (Abcam, ab8295) and stained cells were detected with secondary fluorophore-conjugated antibody against goat IgG (Abcam, ab205719).

### Differentiation of mESCs into definitive endoderm

mESCs were aggregated in suspension in mESC cell media supplemented with 3% Knockout Serum Replacement (SR) (ThermoFisher, 10828028) without LIF to form EBs. The next day (day 1), 100 ng ml⁻¹ Activin A (R&D, 338-AC-050/CF) and 75 ng ml⁻¹ Wnt3a (PeproTech, 7315-200) were added to the EB culture media. On day 3, the EB media were replaced with mESC (without LIF) + 3% SR and 100 ng ml⁻¹ Activin A. On Day 5, cells were stained with primary antibodies against SOX17 (R&D, AF1924) and FOXA2 (Abcam 108422), and detected with secondary fluorophore-conjugated antibodies against goat IgG (ThermoFisher, 11055) and rabbit IgG (ThermoFisher, 31537), respectively.

### Differentiation of mESCs into CD31+ cells

mESCs were disassociated with Accutase (StemCell Technologies, 07920) and washed in DMEM + 10% FBS (Wisent, 0901150). Then, cells were plated at 1 million cells per well in 2 ml mESC media (without LIF) + 25 µg l⁻¹ Activin A (R&D, 338-AC-050/CF), 5 µg l⁻¹ BMP4 (R&D, 314-BP-050/CF), 1 µM CHIR99021 (PeproTech, 2520691) and 10 µM Rock inhibitor (StemCell Technologies, Y-27632) onto 6-well plates coated with 1 ml per well of 1 µg cm⁻² Vitronectin XF (StemCell Technologies,

07180) for 1 h at room temperature. The media were then replaced each day for the next 2 days (days 1 and 2) with 4 ml of mESC media (without LIF) + 10% FBS, 25 µg l$^{-1}$ Activin A, 5 µg l$^{-1}$ BMP4 and 1 µM CHIR99021. The media were then replaced daily for the next 4 days (days 3, 4, 5, 6) with 4 ml mESC media (without LIF) + 10% FBS, 50 µg l$^{-1}$ BMP4, 50 µg l$^{-1}$ VEGF-A (PeptroTech, 450-32) and 5 µM SB431542 (Sigma, S4317). The next day (day 7), cells were disassociated with Accutase and CD31+ cells were isolated using flow cytometry sorting after staining cells with anti-mouse CD31 antibody (Biolegend, 102417), and plated onto Vitronectin-coated plates (1 ml per well at 1 µg cm$^{-2}$) in mESC media (without LIF) + 10% FBS and 50 µg l$^{-1}$ VEGF-A. This medium was replaced every other day and the cells passaged when confluent at 1:3 until maturation into endothelial cells.

## Differentiation of mESCs into adipocytes
Adipocytes were differentiated from mESC by following a previously established protocol[74]. On the final day of differentiation, cells were fixed in 4% PFA at room temperature for 20 min for Oil Red O staining. After fixing, cells were washed with PBS, incubated in 60% isopropanol for 20 min and incubated in an Oil Red O (0.6% 2/v) solution (prepared with 60% isopropanol) for 2 h at room temperature. Cells were washed with PBS to remove unbound dye.

## Differentiation of hESCs into RPE cells
RPEs were differentiated from hESCs as previously described[75] with minor changes. hESCs were plated on Geltrex (ThermoFisher, A1413201) coated plates at 5,000–10,000 cells per cm$^2$ in mTeSR1 medium (StemCell Technologies, 85850) with 50 U ml$^{-1}$ penicillin/streptomycin (ThermoFisher, 15140122). Once the cells reached 90% confluency, they were cultured for 4 weeks in RPE differentiation media, consisting of basal media supplemented with 13% KO SR (ThermoFisher, 10828028), after which the differentiated, pigmented cell clusters were visible. For further use in vitro, the RPE clusters were picked manually for enrichment and expansion in RPE culture media consisting of the basal media supplemented with 5% FBS, 7% KSR and 10 ng ml$^{-1}$ bFGF (R&D, 234-FSE-025/CF). For all stages, the basal media consisted of Knockout DMEM (ThermoFisher, 10829018) with 500 µg ml$^{-1}$ of penicillin/streptomycin (ThermoFisher (Gibco), 15140-122), 1% NEAA solution (ThermoFisher (Gibco), 11140076), 1% sodium pyruvate solution (ThermoFisher (Gibco), 11360070), 2 mM Glutamax supplement (ThermoFisher, 35050079) and 0.1 mM β-mercaptoethanol (Sigma, M6250).

## Differentiation of hESCs into definitive endoderm
hESCs were plated on Geltrex (ThermoFisher, A1413202) coated plates at a density of 5,000–10,000 cells per cm$^2$. The following day, the cells were washed with PBS and cultured in chemical-defined, serum-free X-VIVO 10 medium (Lonza, 04-380Q) supplemented with 3 µM CHIR99021 (PeproTech, 2520691) and 100 ng ml$^{-1}$ Activin A (R&D, 338-AC-050/CF) for 1 d, followed by media containing 100 ng ml$^{-1}$ Activin A only, for another 4 d. Differentiated cells were assayed by qPCR for downregulation of pluripotency markers *OCT4* and *NANOG*, and upregulation of *SOX17* and *FOXA2*.

## Differentiation of hESCs into cardiomyocytes
hESCs were differentiated using the STEMdif cardiomyocyte differentiation and maintenance kit (StemCell Technologies, 05010) following manufacturer protocol. Differentiated cells were fixed in 4% PFA and then stained with antibodies against a-actinin and troponin.

## Differentiation of hESCs into endothelial cells
Endothelial cells were differentiated as previously described[76] with minor changes. hESCs were plated on Geltrex (ThermoFisher (Gibco), A1413302) coated plates in TeSR-E8TM media (StemCell Technologies, 05990) with 1 µM Rock inhibitor (Y-27632, Selleckchem, S1049). Two days later (d0), cells were collected and reseeded onto non-treated

culture dishes coated with 1 µg cm$^{-1}$ Vitronectin (StemCell Technologies, 07180) at a density of 100,000 cells per cm$^2$ in TeSR-E8 medium supplemented with 5 ng ml$^{-1}$ BMP4 (R&D, 314-BP-050/CF), 25 ng ml$^{-1}$ Activin A (R&D, 338-AC-050/CF) and 1 µM CHIR99021 (PeproTech, 2520691) for 2 d. At d2, the media were switched to TeSRTM-E7TM media (StemCell Technologies, 05914) supplemented with 50 ng ml$^{-1}$ VEGF-A (Peprotech, 100-20), 50 ng ml$^{-1}$ BMP4 (R&D, 314-BP-050/CF) and 5 µM SB431542 (ReproCell, 04-0010-05) for another 4–5 d. Final differentiated cells were stained with antibodies against CD31 (Pe-Cy7; Biolegend,303118) and CD144 (PE; Biolegend, 138009).

## Co-culture of RPEs with hPBMCs or NK92 wells
Human blood samples were ordered from StemCell Technologies or the Canadian Blood Service (Vancouver), and the PBMCs were isolated using Ficoll-Paque Plus media (GE Healthcare) and rapid centrifugation. Isolated PBMCs were frozen in media with 10% DMSO + 90% FBS for future use. Human NK92 cancer cell line (HANK) was kindly provided by Dr Armand Keating (Krembil Research Institute, UHN, Toronto) and cultured in flasks using X-vivo 10 (Lonza, 04-380Q) media supplemented with 5% human serum AB (Sigma, A7030). RPEs were irradiated with a dose of 10 Gy before co-culture experiments with PBMCs (20:1 ratio of PBMCs:RPE) or NK92 (various ratios shown in Fig. 4k) cells in 96-well plates for 5 d. IFNg and TNFa were measured from culture supernatants on day 5 by ELISA (ELISA MAX; Biolegend, 430204 and 430104). An IncuCyte live-cell analysis system (Sartorius), which takes images every 1–2 h, was used for cell area measurements over the course of 5 days in RPE + hPBMC co-cultures. Separate co-culture plates were used for downstream analysis by CyTOF, which involved collection of cells at days 0, 1, 3 and 5. The intracellular permeabilization/fixation, staining and CyTOF instrumental run were performed as a service by the SickKids Hospital (Toronto) and the full panel of antibodies is listed in Supplementary Table 5. Cisplatin was used for cell viability. The samples were acquired on a Helios mass cytometer and normalized by technicians at The Flow Cytometry Center at the SickKids Hospital. The data were analysed using the Cytobank software.

## Loading of endothelial cells into the microfluidic device
ECs differentiated from uncloaked or cloaked human iPSCs were maintained in EC culture medium (Lonza, CC-3162) supplemented with 1 µl ml$^{-1}$ VEGF (1 µl ml$^{-1}$ Peprotech, 100-20). The microchannels of the device were washed with 70% ethanol, penicillin/streptomycin (1%) solution, distilled water and PBS, and then treated with CellAdher solution and Vitronectin (StemCell Technologies, 07180) for 1 h at room temperature. ECs were collected from culture plates using Accutase (StemCell Technologies, 07920), resuspended in EC culture medium at a density of $2 \times 10^6$ cells per ml and then gently injected into the microchannels through the inlet. The microfluidic device was then incubated at 37 °C for 15 min to allow for cell adherence to the microchannels. Confocal microscopy was used to confirm cell seeding across the microchannels. The inlet and outlet of the microfluidic device were connected to a closed-loop recirculatory flow system and supplied with continuous (20 µl min$^{-1}$) culture media. The entire setup was placed in an incubator at 37 °C and 5% CO$_2$ for 24–48 h to allow for cell adaptation to the system. Cell morphology and adherence to the microchannels were confirmed using confocal microscopy. The microfluidic system was integrated with a two-photon confocal microscope (LSM750, Zeiss) equipped with an incubation chamber at 37 °C and 5% CO$_2$, and an automated stage positioning system.

## Measurement of EC–PBMC interaction in the microfluidic device
Human PBMCs were stained with CellTrace Far-Red (ThermoFisher, C34564) and resuspended in EC media in the reservoir at a cell density of $1.8 \times 10^6$. PBMCs were circulated with the culture medium through the endothelialized microchannels while the time-lapse confocal

fluorescence live-cell imaging was carried out over 12 h by confocal microscopy (LSM750, Zeiss). DAPI (Sigma-Aldrich) was added into the culture media at a dilution of 1:15,000 and used as a marker of viability. The live-cell fluorescence imaging was performed over at least different positions across the microfluidic device. The images from the fluorescence video microscopy were processed in ImageJ and the spatiotemporal changes in the occupancy areas ($\mu m^2$) of DAPI, GFP and far-red signals were calculated for each frame and normalized by the area of the microchannel ($\mu m^2$) in the field. The calculated normalized values of DAPI (dead cells) and GFP (ESC-ECs) occupancy areas were evaluated with respect to the elapsed time and compared between the cloaked and non-cloaked systems.

### Induction of hyperglycaemia with STZ
Mice were fasted for 5 h and intraperitoneally injected with 200 mg kg$^{-1}$ STZ (Sigma, C1030).

### Islet isolation and transplantation
Islets were collected from 12-wk-old C3H donor mice as previously described[77]. Islets were then cultured overnight at 37 °C in RPMI-1640 medium (ThermoFisher (Gibco), 11875-119) supplemented with 15% FBS (Wisent, 0901150), 1 mM nonessential amino acids, 1 mM sodium pyruvate, 2 mM Glutamax, 50 U ml$^{-1}$ penicillin, 50 mg ml$^{-1}$ streptomycin and 0.55 mM 2-mercaptoethanol before transplantation. Islets were then grafted into pre-existing0 dormant tissues of STZ-treated mice when recipients' blood glucose levels reached 20 mmol l$^{-1}$ or more. To engraft the islets, a 1–2 cm incision was made through the skin on the dorsal side of the dormant tissue and the tissue was exteriorized. Then, 350–400 islets were loaded on top of a 1–2 mm$^3$ piece of GELOFOAM sponge and stabilized by covering the islets with undiluted Matrigel HC. A small incision ~5 mm in length was then made on the dorsal side of the dormant tissue and the GELOFOAM-containing islets transferred into the space created by the incision. The incision was then sutured and the skin closed with staples. To transplant islets underneath the kidney capsule, 350–500 islets isolated from 12-week-old CH3 donor mice were loaded into a syringe and transferred underneath the kidney capsule via a small tube attached to the syringe, as previously described[49].

### Blood glucose measurement
Blood glucose was measured with a standard glucometer from blood taken from the tail tip and recorded as mmol l$^{-1}$.

### Reporting summary
Further information on research design is available in the Nature Portfolio Reporting Summary linked to this article.

## Data availability
The data for the teratoma-growth experiments (Figs. 2 and 4, and Extended Data Fig. 9) and for the blood-glucose experiments (Extended Data Fig. 10) are provided as source data. The raw and analysed datasets generated during the study are available for research purposes from the corresponding author on reasonable request. Source data are provided with this paper.

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

## Acknowledgements

We thank C. Monetti and Q. Liang for providing the FailSafe mouse and human ESCs; A. Bang for support with flow analysis; E. Langlille for contributed artwork; and the staff at the TCP animal facility, in particular C. Dalrymple, for support. This work was supported by CIHR (Foundation Scheme 143231 JUN2022), Canadian Research Chair (CRCP 950-230422 UT SEP2021) and Medicine by Design (University of Toronto) to A.N., as well as funding by the Ontario Research Fund (MRI ORF RE08-064 SEPT2021).

## Author contributions

J.H. wrote the paper, contributed to conceptual and experimental design, generated data and analysed data. K.V.-N. and H.Y. contributed to conceptual and experimental design, generated data and analysed data. M.S. generated and analysed NGS and RT–qPCR data. J.K.T., E.D.J., Z.I., M.M., C.L., P.Z. and T.O. generated data. H.-K.S. and J.H.L. assisted with the CyTOF assay and analysis. M.I. generated data, characterized cardiomyocyte differentiation, established the microfluidic assay and analysed data. I.M.R., M.B.W. and S.J.L. contributed to conceptual and experimental design. A.N. contributed to conceptual and experimental design and wrote the paper.

## Competing interests

A.N. is an inventor on the patent covering the FailSafe technology (PCT/CA2016/050256) and is a co-founder and shareholder of panCELLa Inc. A.N., J.H. and K.N. are inventors on a patent (PCT/CA2019/051808) covering induction of allograft tolerance using immunomodulatory transgenes. The other authors declare no competing interests.

## Additional information

**Extended data** is available for this paper at https://doi.org/10.1038/s41551-023-01133-y.

**Correspondence and requests for materials** should be addressed to Andras Nagy.

# a

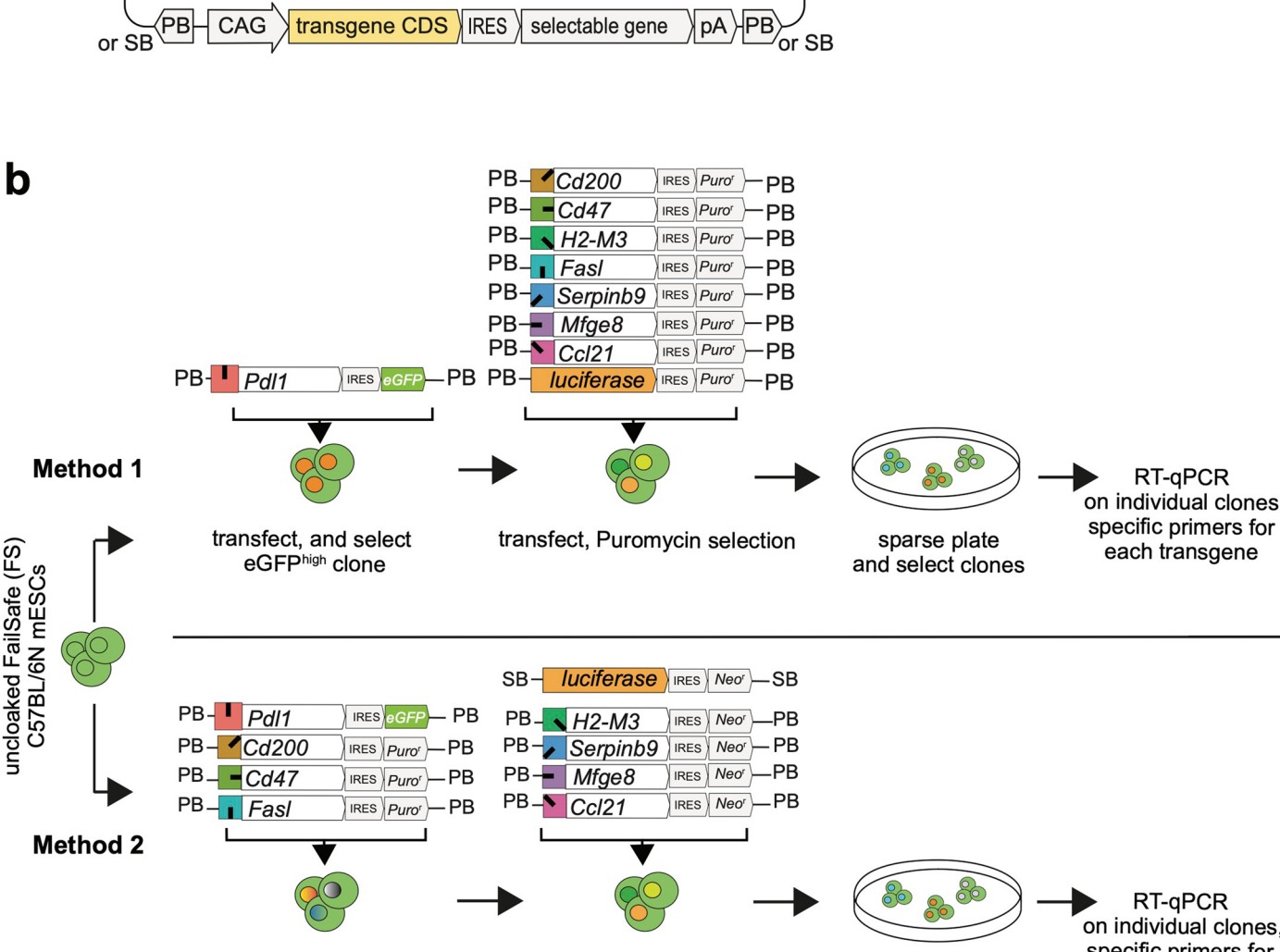

# b

**Extended Data Fig. 1 | Approach to insert immunomodulatory transgenes into mESCs. a**. Structure of representative piggyBac (PB) or Sleeping Beauty (SB) plasmids used to express the CDS of eight immunomodulatory genes (*Ccl21, Pdl1, Fasl, Serpinb9, H2-M3, Cd47, Cd200*, and *Mfge8*) and luciferase in FailSafe B6 mESCs. Gene expression driven by a CAG promoter and linked by an IRES to an eGFP fluorophore, or Puromycin or Neomycin resistance genes. The entire expression cassette is flanked by inverted terminal repeat of piggyBac or Sleeping Beauty transposons. The NCBI Refseq protein ID encoded by each gene CDS is listed in Supplementary Table 1. **b**. Method 1) Failsafe B6 mESCs were transfected with a PB plasmid encoding an eGFP-linked *Pdl1*, then a clone with high eGFP^high expression was isolated. This was followed by co-transfection of PB transposon plasmids encoding *Ccl21, Fasl, Serpinb9, H2-M3, Cd47, Cd200, Mfge8* and *luciferase* and then drug selection with Puromycin. Method 2) FailSafe B6 mESCs were simultaneously transfected with PB plasmids encoding *Pdl1, Fasl, Cd47, and Cd200*, followed by drug selection with Puromycin and then a FACS sort on clones with high expression of each factor based on antibody staining and and eGFP for *Pdl1*. Bulk sorted cells then transfected with SB transposons encoding *Ccl21, Mfge8, SerpbinB9, H2-M3* and luciferase, followed by drug selection with G418. Both methods resulted in a polyclonal pool that were used to establish clonal lines and analyzed by RT-qPCR for transgene expression levels.

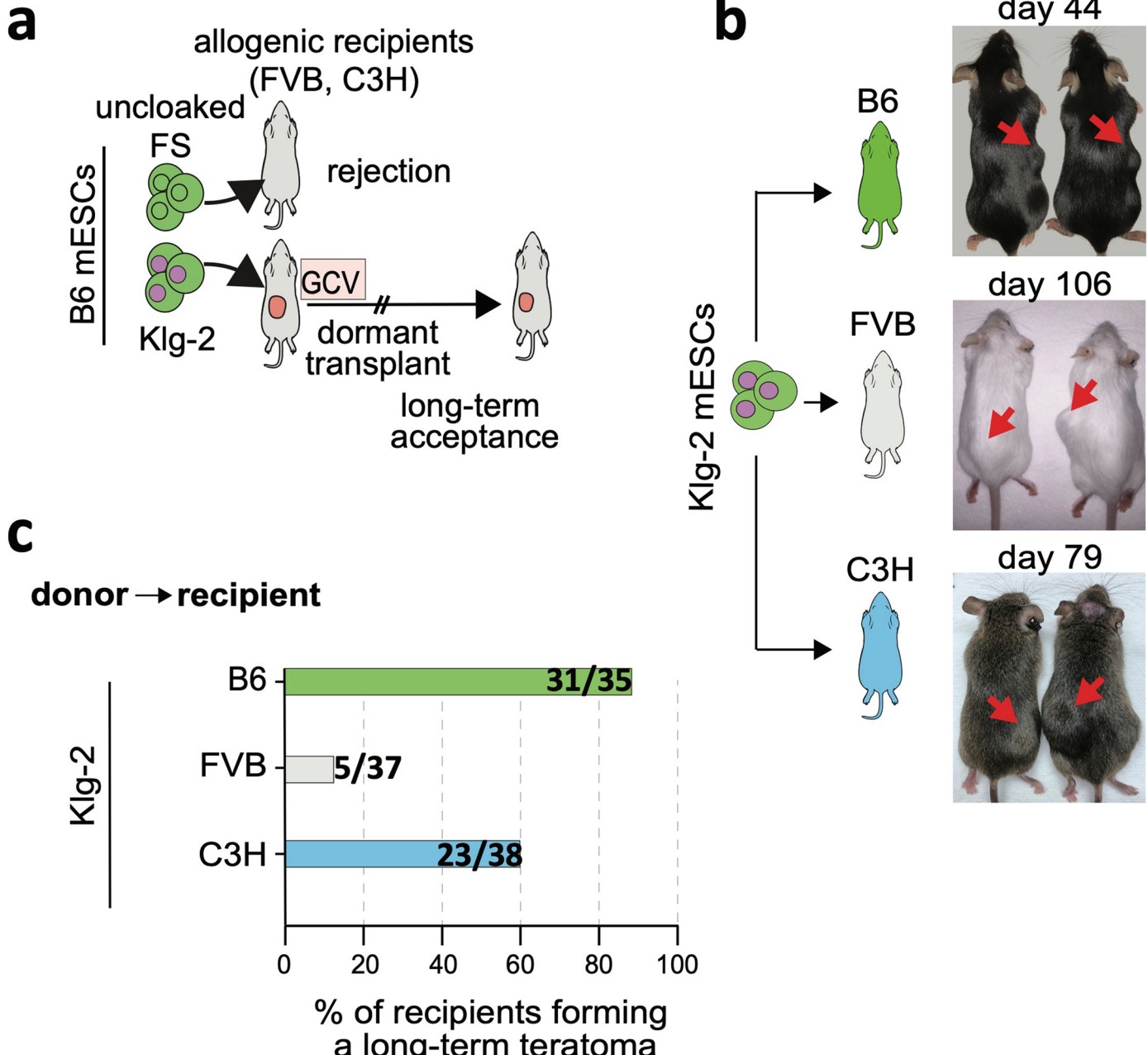

**Extended Data Fig. 2 | Long-term acceptance of dormant cloaked Klg-2 mESC transplant in allogeneic recipients. a.** Schema of long-term allogeneic acceptance assay. Clone 43 (in Fig. 1c, bordered in red) was renamed as Klg-2. **b.** Pictures of two representative Klg-2 derived, dormant transplant in isogeneic and allogeneic recipients (red arrows show subcutaneous transplants). **c.** Percent of isogenic and allogeneic recipients that formed dormant transplant from Klg-2 mESCs. Number in the horizontal bar indicates the total number of mice that transplants divided by the total number of mice injected. Observation period ranged from 2 to 6 months. Once formed, no instances of Klg-2 transplant-clearance occurred in any recipient. Data for all mice, including observation period, shown in Supplementary Table 3.

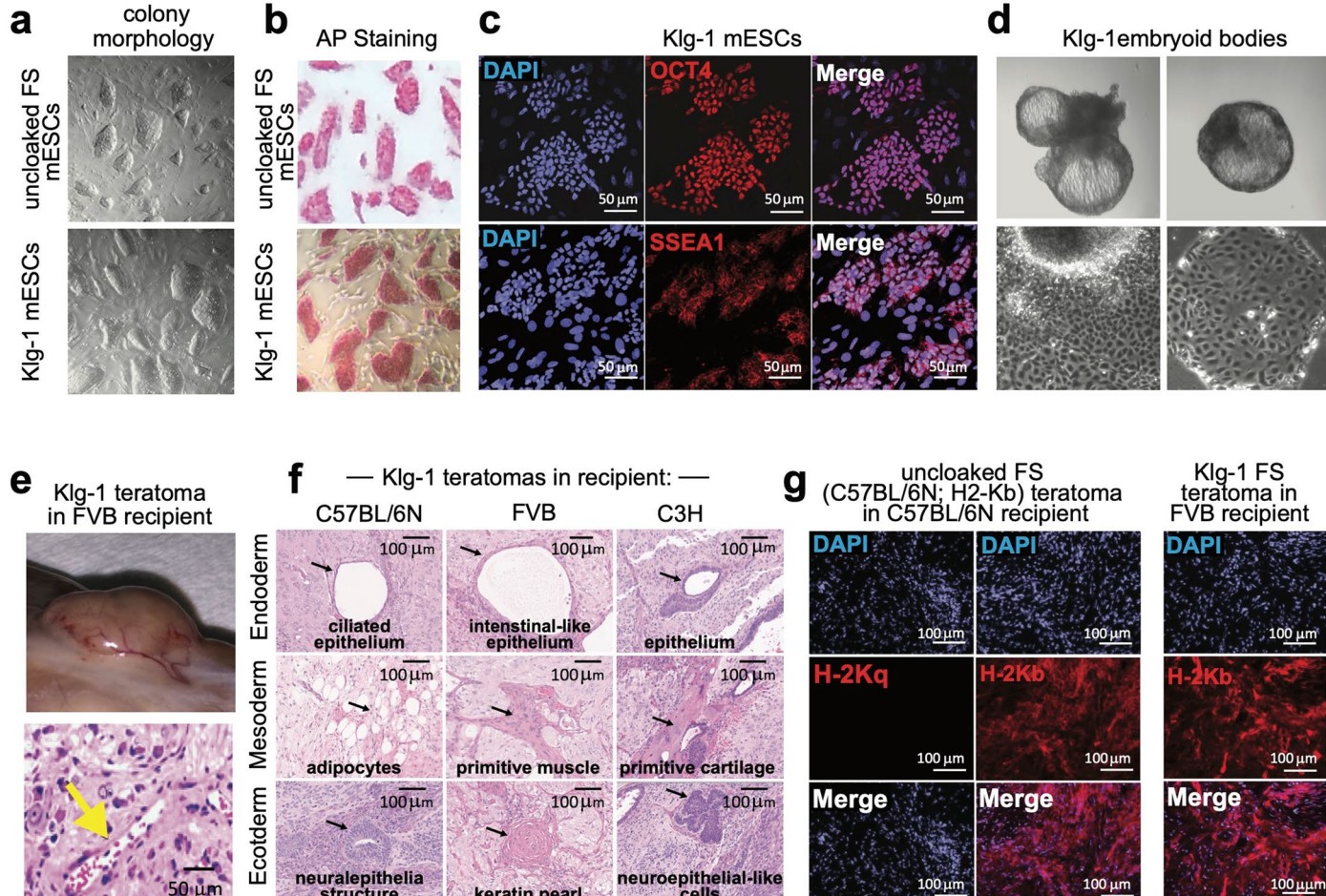

**Extended Data Fig. 3 | Characterization of uncloaked FS and Klg-1 mESCs.**
**a.** Representative brightfield images of the two cell lines showing typical ES cell morphology, **b.** Undifferentiated cells are Alkaline phosphatase (AP) positive, and **c.** express OCT4 and SSEA1 pluripotency markers shown by IHC staining. **d.** Klg-1 embryoid bodies and cell outgrowth. **e.** Picture of Klg-1-derived dormant transplant accepted in an allogeneic FVB recipient showing robust neovascularization of the tissue (top); H&E staining of tissue section showing a representative vessel that contains red blood cells (bottom). **f.** H&E staining of sections showing all three germ layers present in the long-term-accepted, Klg-1-derived dormant transplants in allogeneic recipients. Black arrows point to lineage indicated by text in each image. **g.** IHC showing similar MHC class I expression in dormant transplants derived from both isogeneic uncloaked FS and allogeneic Klg-1 mESCs. B6 mice express the b allele of the H2-K MHC class genes. Staining for the q allele serves as a negative control.

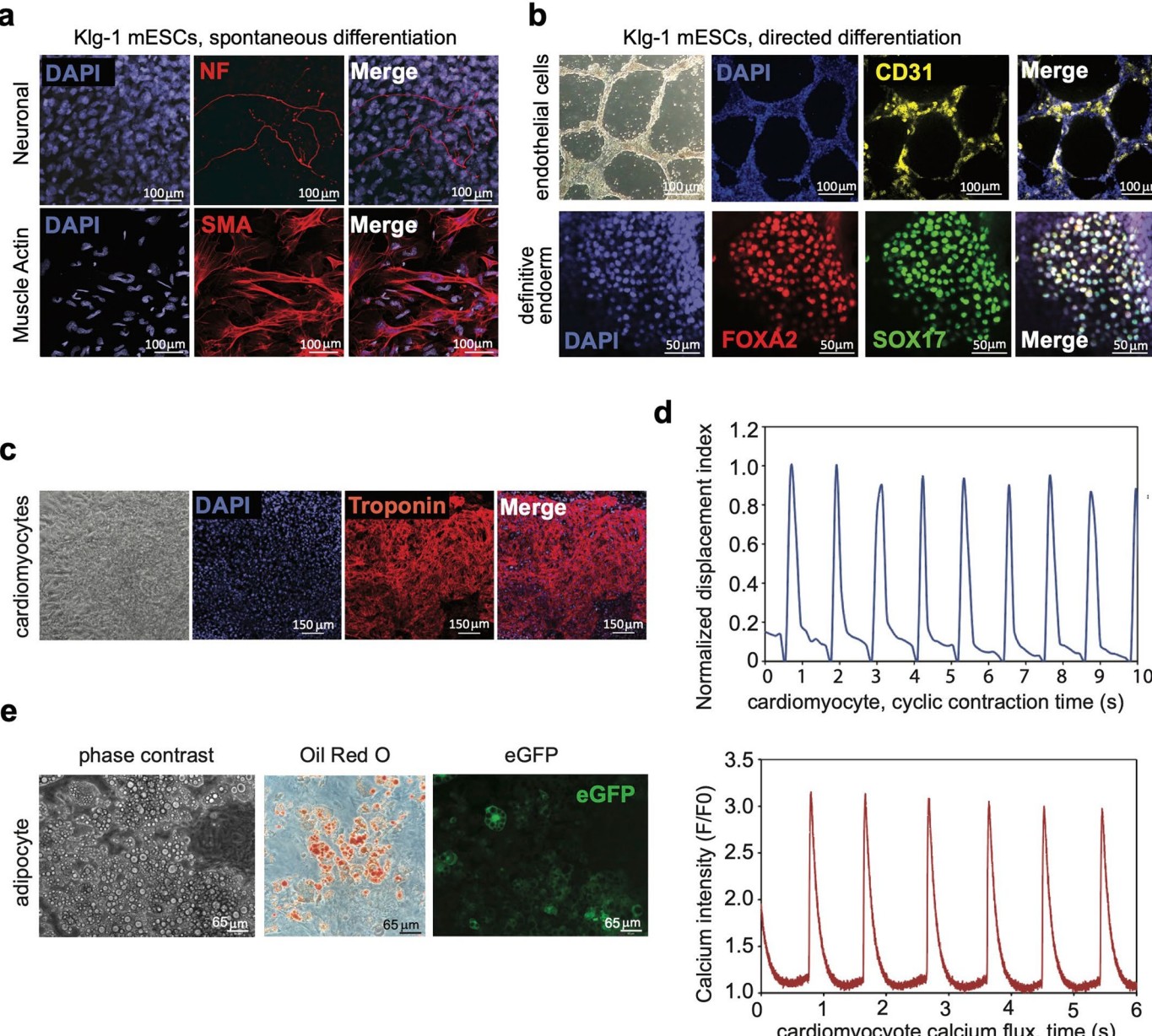

**Extended Data Fig. 4 | Differentiation of Klg-1 FS mESCs into distinct lineages and cell types. a.** IHC showing presence of cells expressing ectoderm (Neurofilament, NF) and mesoderm (Smooth Muscle Actin, SMA), early embryonic lineage markers, in Klg-1 embryoid bodies. **b**. CD31+ endothelial-like cells with vascular tube-like formations (top) and FOXA2 + , SOX17+ definitive endoderm derived from Klg-1 FS mESCs using directed protocols. **c**. Troponin+ cells after cardiomyocyte differentiation of Klg-1mESCs, and **d.** cyclic contraction (top graph) measured as a function of normalized cell displacement and calcium flux (bottom graph) after loading with Fluo-4 AM 15 days after cardiomyocyte differentiation. **e**. Adipocyte differentiation showing phase contrast, Oil Red O staining, and eGFP fluorescence. The eGFP reporter is expressed by the *Pdl1* transgene in Klg-1 FS mESCs shown in Extended Data Fig. 1b).

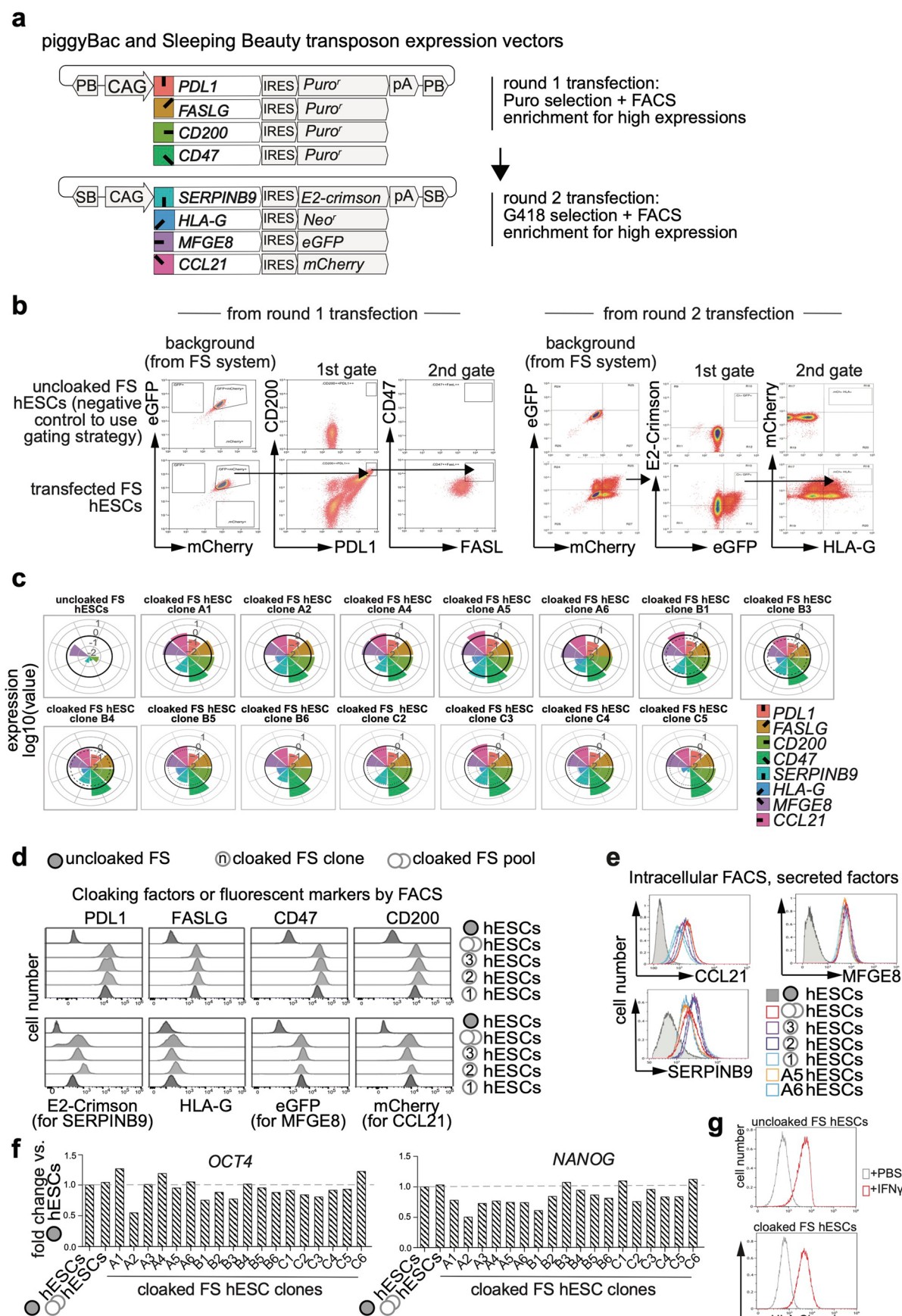

**Extended Data Fig. 5 | See next page for caption.**

**Extended Data Fig. 5 | Generation and characterization of hESC lines expressing immunomodulatory transgenes. a**. Structure of the piggyBac or Sleeping Beauty transposon plasmids and approach used to overexpress the eight human immunomodulatory genes *(CCL21, PDL1, FASLG, SERPINB9, HLA-G, CD47, CD200, and MFGE8)* in FS hESCs. The NCBI Refseq protein ID encoded by each gene CDS is listed in Supplementary Table 1. Uncloaked FS hESCs were co-transfected with piggyBac vectors encoding PDL1, FASLG, CD200, and CD47, followed by puromycin selection and FACS sorting on high expressors using antibodies against each factor. Then, this bulk population was transfected with Sleeping Beauty vectors encoding SERPINB9, HLA-G, MFGE8, and CCL21, followed by selection with G418, and FACS sorting into bulk and clonal lines using the fluorophores encoded on the expression vectors. Fluorophores were only linked to those immunomodulatory factors that are secreted or intracellular (MFGE8, CCL21 and SERPINB9). **b**. FACS plots showing protein expression levels and gating strategy used to isolate cloaked hESCs with expression of all eight immunomodulatory factors. Untransfected FS hESC stained with the same antibodies (which express intermediate levels of mCherry from the FS-containing cassette) were used as FACS gating controls. **c**. RT-qPCR showing expression levels of all eight immunomodulatory factors on clonal, cloaked hESCs lines after both rounds of transfection and FACS sorting. Concentric circles represent the $\log_{10}$ scale. **d**. FACS plots showing immunomodulatory protein levels on selected cloaked hESC clones using antibodies or the fluorophore marker linked to SERPINB9, CCL21, and MFGE8. **e**. Intracellular antibody staining against secreted factors SERPINB9, CCL21, and MFGE8 on selected cloaked hESC clones. Untransfected FS hESCs shown in gray. **f**. qPCR of *OCT4* and *NANOG* expression levels. **g**. Upregulation of HLA class I (using antibody recognizing -A, -B, and -C) 36 hours after *in vitro* stimulation with 100 ng/mL IFNg.

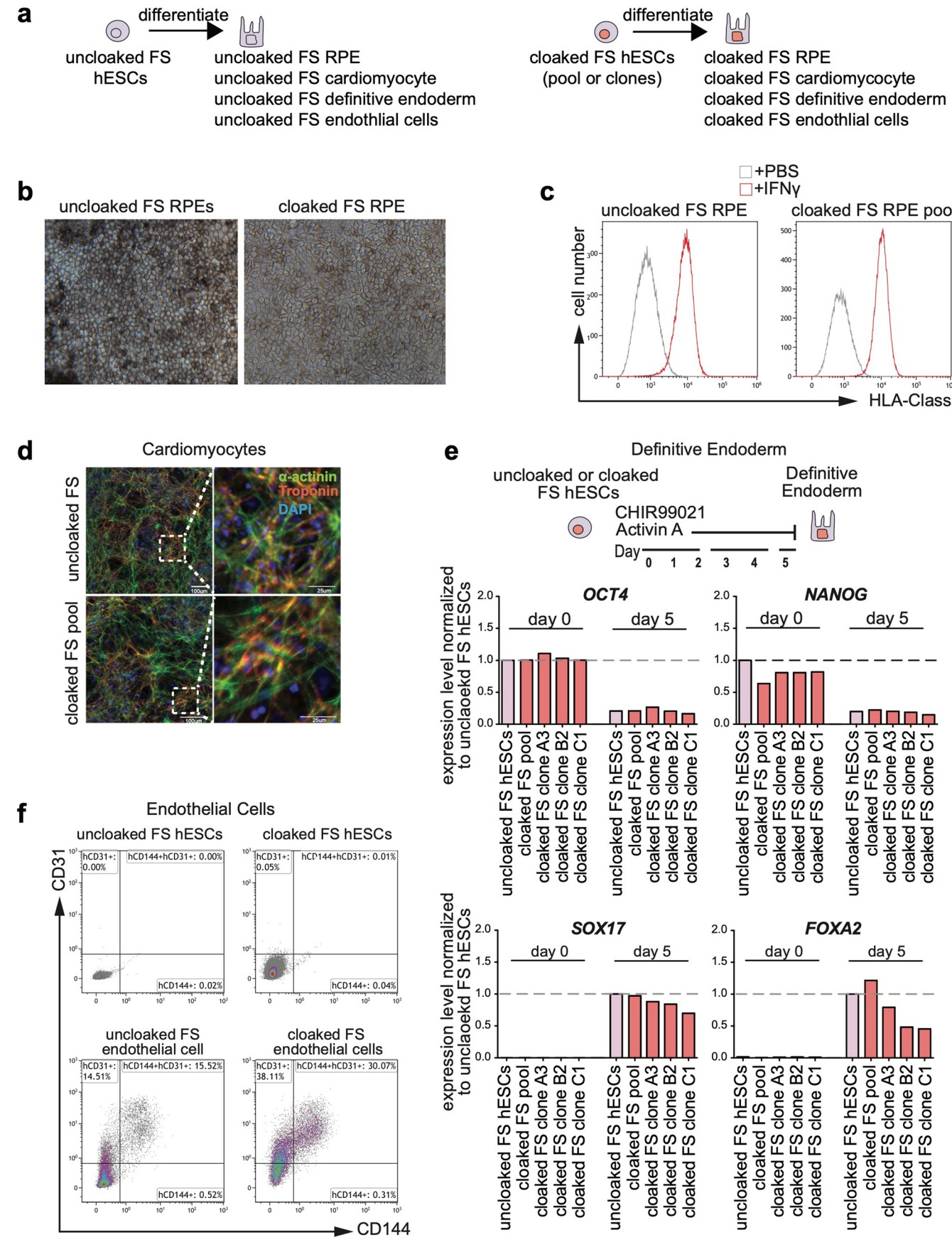

**Extended Data Fig. 6 | See next page for caption.**

**Extended Data Fig. 6 | Differentiation of uncloaked FS and cloaked FS hESCs into cell types with clinical relevance. a**. Uncloaked FS and cloaked hESCs were differentiated into Retinal Pigmented Epithelial (RPE), cardiomyocyte, definitive endoderm, and endothelial cells. **b**. Phase contrast microscopy shows *in vitro* differentiated pigmented RPE with characteristic morphology. **c. U**pregulation of HLA class I (using antibody recognizing -A, -B, and -C) on uncloaked FS and cloaked RPEs 36 hours after *in vitro* stimulation with 100 ng/mL IFNg. **d**. IHC showing alpha-actinin and troponin expression after cardiomyocyte differentiation. **e**. *OCT4 and NANOG versus SOX17* and *FOXA2 reciprocal* expression levels measured by qPCR after differentiation into definitive endoderm. *OCT4* and *NANOG* normalized cells to uncloaked FS hESCs at day 0. *SOX17* and *FOXA2* normalized to cells differentiated from uncloaked FS hESCs at day 5. **f**. CD144 and CD31 expression on uncloaked FS and cloaked FS hESCs before differentiation (top row), and after differentiation into endothelial cells (bottom row).

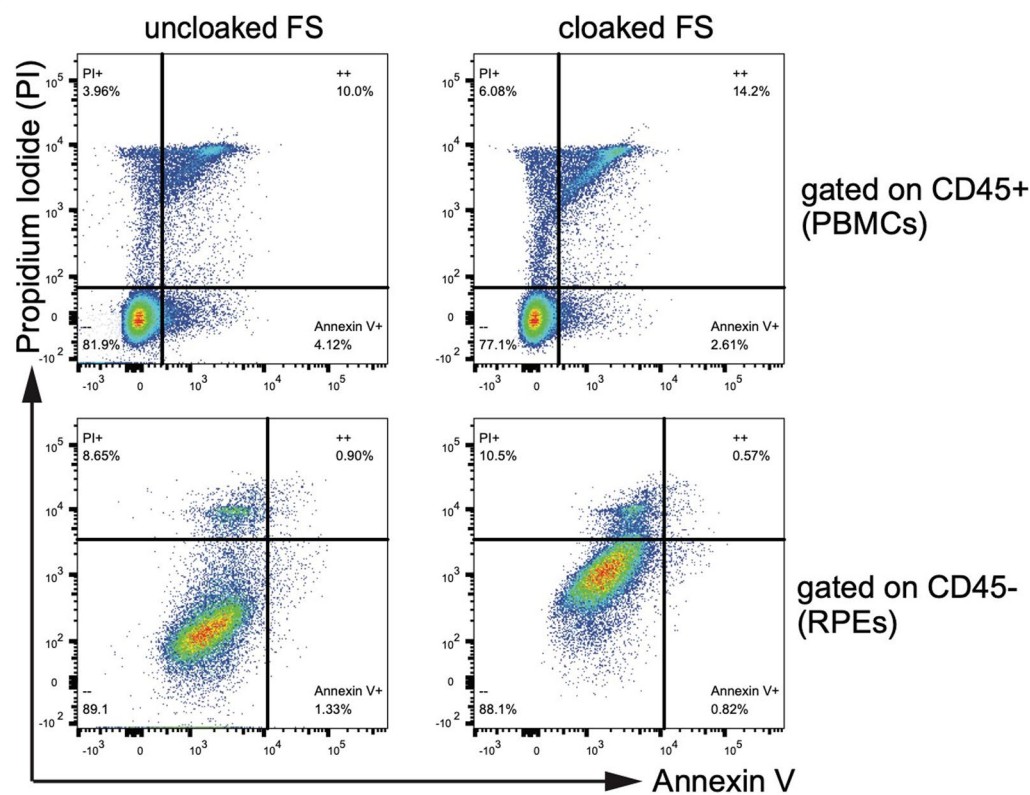

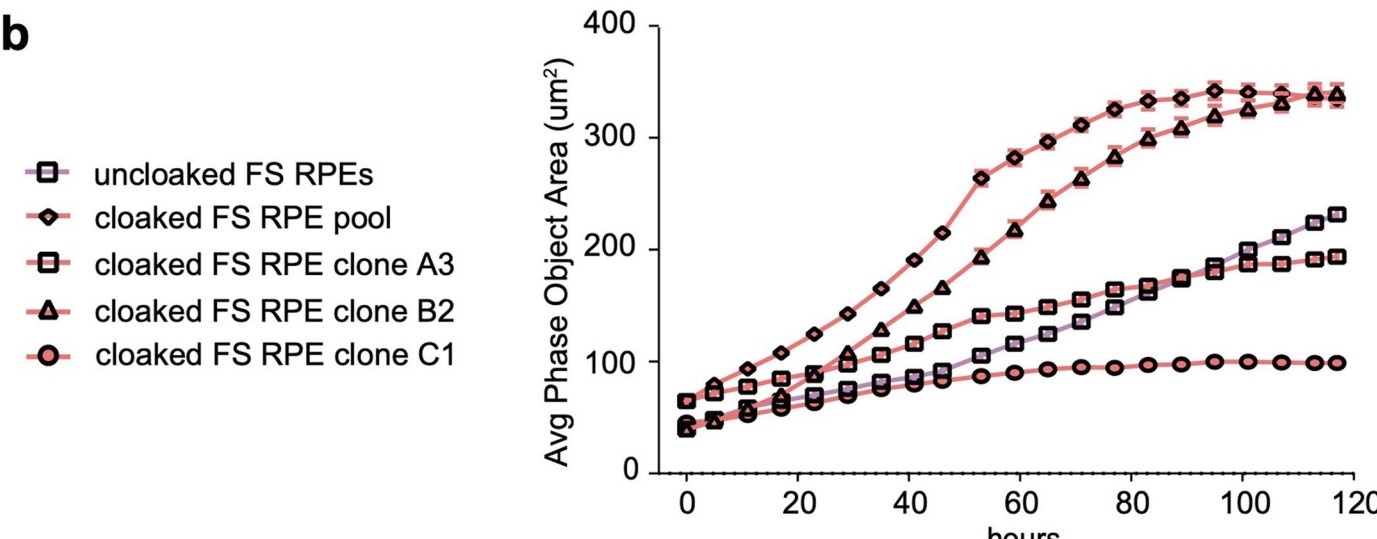

**Extended Data Fig. 7 | *In vitro* co-culture of human PBMCs and uncloaked FS or cloaked FS human RPEs.** RPEs were co-cultured with hPBMCs for 5 days. **a**. Annexin V and propidium iodide (PI) staining on the CD45+ and CD45- compartment of hPBMC. **b**. Phase object area as measured by the phase contrast microscopy (IncuCyte system). Images taken every 6 hours (mean ± SEM, n = 6 independent wells for each group, average of 5 fields / well).

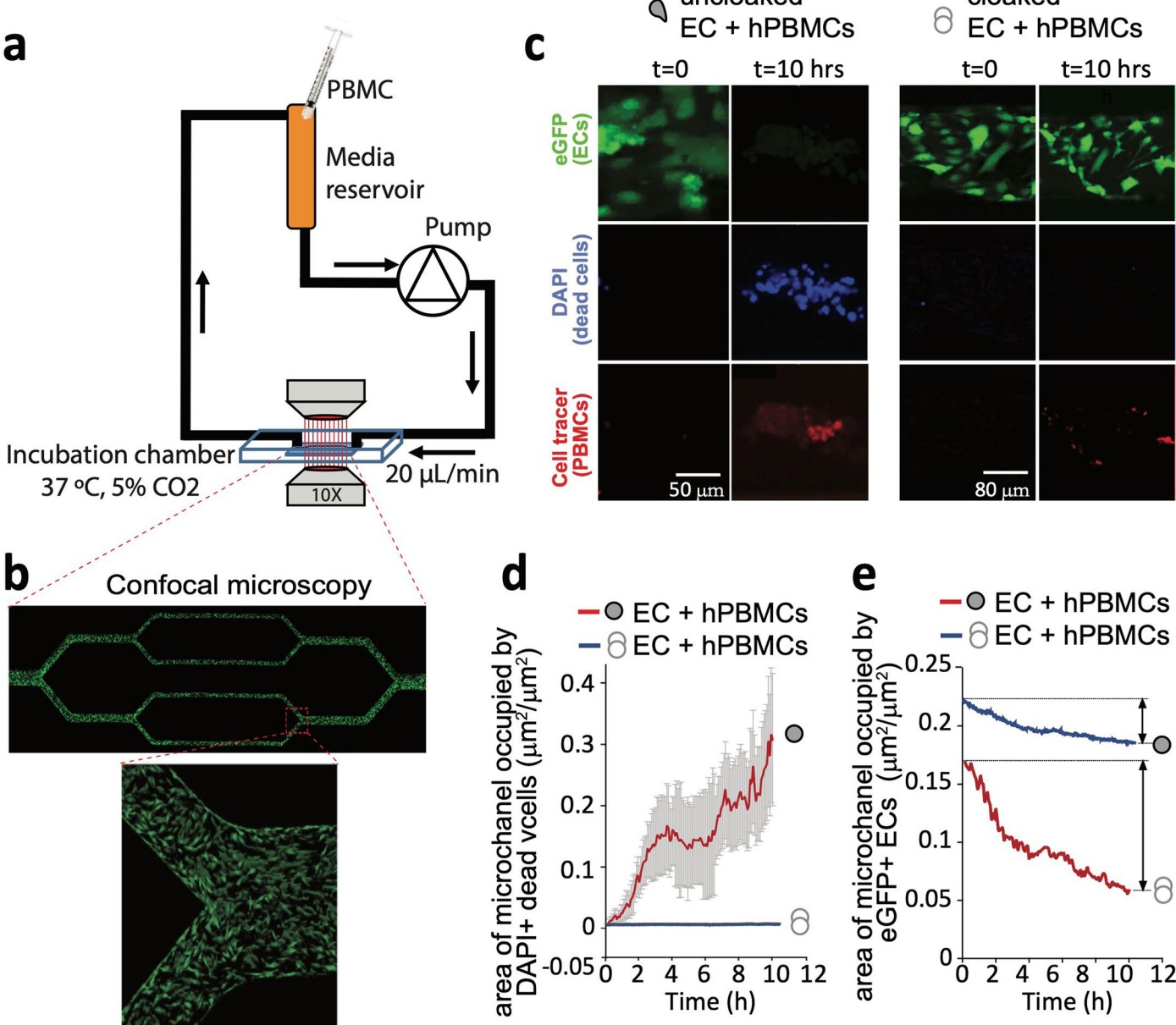

**Extended Data Fig. 8 | Microfluidic device to study the interactions between cloaked differentiated cells and hPBMCs. a**. Scheme showing the media reservoir, circulating system, and inlet for injecting PBMCs and cell viability reagents. **b**. Microfluidic chip showing eGFP+ endothelial cells lining the interior channels. **c**. Confocal microscopy live cell imaging showing the EC-lined microchannel of the microfluidic device after loading cell tracer-labelled hPBMCs into circulation at t = 0 and t = 10 hours. **d**. Normalized area of the microchannel occupied by DAPI+ dead cells. **e**. eGFP+ ECs 10 hours after injection of hPBMCs. Measurements were taken in at least 3 separate areas containing the microchannels of the device using ImageJ to measure the fluorescence signals on specific frames extracted from the live-cell imaging.

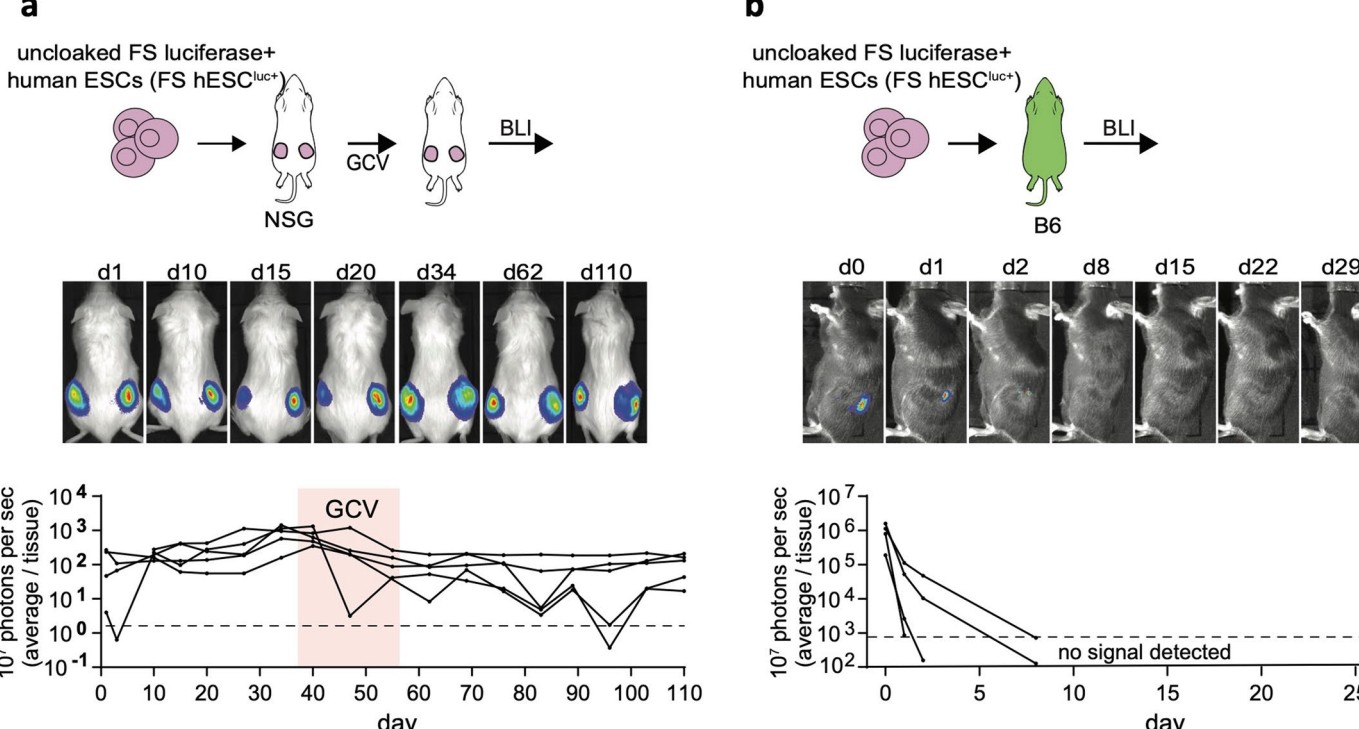

**Extended Data Fig. 9 | Luciferase imaging FS hESCs transplant in NSG and B6 mice.** 10-12 million luciferase-expressing uncloaked FS hESCs (FS hESCs[luc+]) were injected into **a**. NSG (n = 5) or **b**. B6 (n = 4) recipients and monitored by BLI imaging. Black dotted line shows the detection limit. Images show one representative mouse per group. NSG recipients were treated with GCV to halt growth of human cell transplant. BLI images were taken approximately 15 min after IP injection with luciferin, with 10 second exposure time.

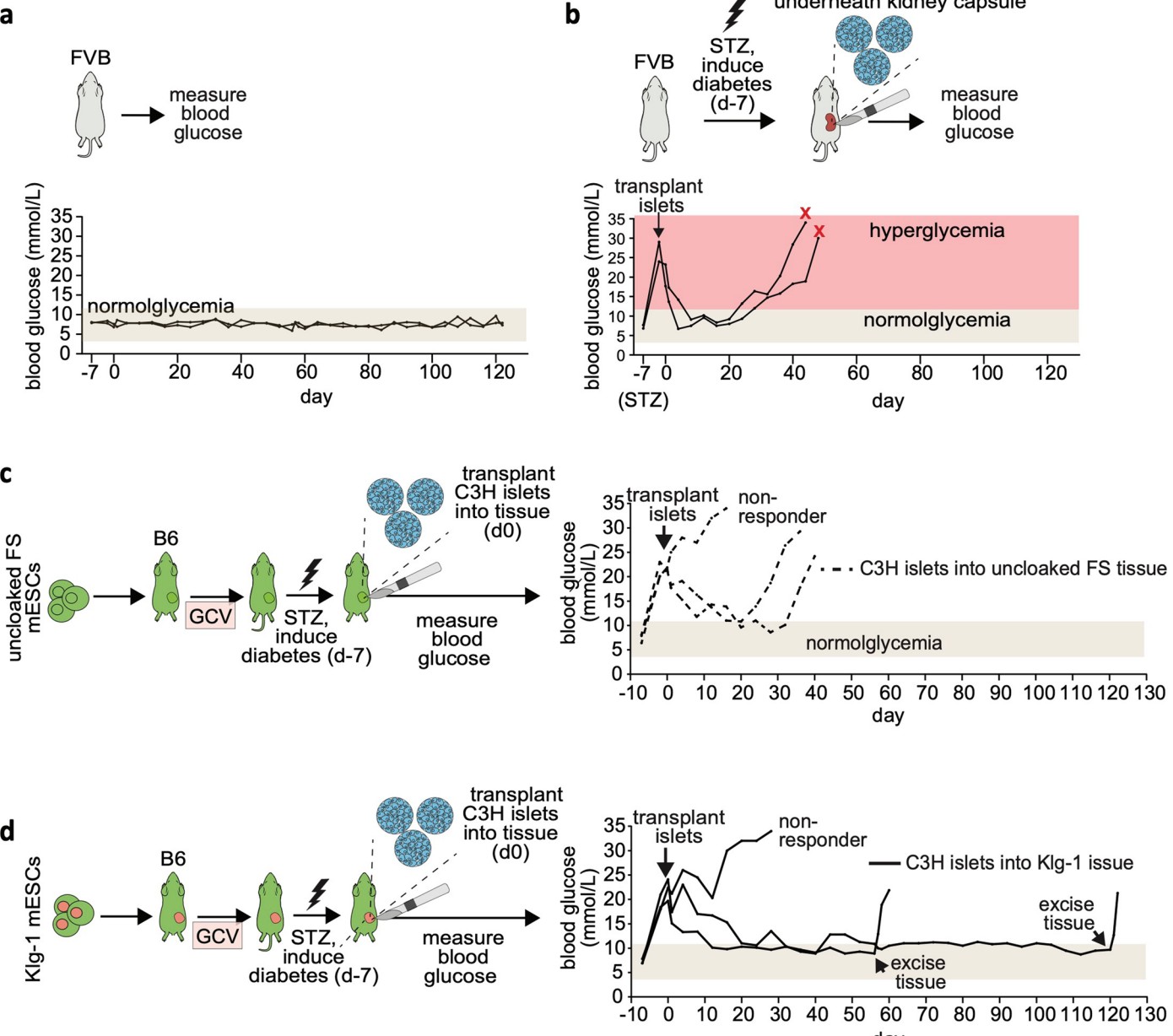

**Extended Data Fig. 10 | Klg-1 - derived dormant transplant forms immune-privileged tissue that protects allogeneic islets in a mouse model of diabetes.** **a**. Normoglycemic range in wild type FVB/NJ mice is between 4-11 mmol/L (brown highlighted region). **b**. FVB/NJ recipients were treated with STZ (d-7), a beta-cell cytotoxic agent, leading to the rapid development of hyperglycemia (red highlighted region). Seven days later, islets isolated from allogeneic C3H/HeJ donor mice were harvested and transplanted (d0) underneath the kidney capsule of STZ-treated FVB/NJ recipients. Normoglycemia was only restored short term, indicating the rejection of the allogeneic islets. **c**. Dormant transplants derived from uncloaked FS mESCs in B6 mice. Recipients were then treated with STZ to induce hyperglycemia, and 7 days later (d0), islets from allogeneic C3H mice were grafted into the tissue that rapidly returned to stable normoglycemia (n = 3 recipients). The hyperglycemia, however, returned after a few weeks, indicating the rejection of the allogeneic islets. **d**. Dormant transplants derived from Klg-1 cell line in B6 mice. Then STZ-induced hyperglycemia was treated with allogeneic C3H islets grafted into the transplants (n = 3 recipients). The animals then rapidly returned to stable normoglycemia that lasted until the transplant surgically removed.

# Reporting Summary

## Statistics

For all statistical analyses, confirm that the following items are present in the figure legend, table legend, main text, or Methods section.

| n/a | Confirmed | |
|---|---|---|
| ☐ | ☒ | The exact sample size (*n*) for each experimental group/condition, given as a discrete number and unit of measurement |
| ☐ | ☒ | A statement on whether measurements were taken from distinct samples or whether the same sample was measured repeatedly |
| ☐ | ☒ | The statistical test(s) used AND whether they are one- or two-sided<br>*Only common tests should be described solely by name; describe more complex techniques in the Methods section.* |
| ☐ | ☒ | A description of all covariates tested |
| ☒ | ☐ | A description of any assumptions or corrections, such as tests of normality and adjustment for multiple comparisons |
| ☒ | ☐ | A full description of the statistical parameters including central tendency (e.g. means) or other basic estimates (e.g. regression coefficient) AND variation (e.g. standard deviation) or associated estimates of uncertainty (e.g. confidence intervals) |
| ☐ | ☒ | For null hypothesis testing, the test statistic (e.g. *F*, *t*, *r*) with confidence intervals, effect sizes, degrees of freedom and *P* value noted<br>*Give P values as exact values whenever suitable.* |
| ☒ | ☐ | For Bayesian analysis, information on the choice of priors and Markov chain Monte Carlo settings |
| ☒ | ☐ | For hierarchical and complex designs, identification of the appropriate level for tests and full reporting of outcomes |
| ☒ | ☐ | Estimates of effect sizes (e.g. Cohen's *d*, Pearson's *r*), indicating how they were calculated |

*Our web collection on statistics for biologists contains articles on many of the points above.*

## Software and code

Policy information about availability of computer code

| Data collection | Zen 2011 (black edition), for the confocal images taken for immunostaining<br>OpenLab 5.5.2, for bright-field images<br>Bio-Rad CFX Manager 3.1, for qPCR data collection<br>PerkenELmer Living Image V4.5, for Bioluminescent Imaging (BLI) collection<br>Summit v6.2, for FACS (Beckman Coulter, MoFlow AStrios EQ cell sorter);<br>Gallios software, for flow cytometry (Beckman Coulter, GALLIOS flow cytometer) |
|---|---|
| Data analysis | Bio-Rad CFX Manager 3.1, for qPCR analysis<br>GraphPad Prism software v10, for all data processing.<br>Kaluza 1.5 for and FlowJo v10, for flow-cytometry analysis<br>Cytobank software, for Cytof data analysis<br>PerkenELmer Living Image V4.5, for bioluminescence imaging (BLI) analysis<br>Star tool, for Splinkerette PCR analysis<br>Integrative Genomic Viewer (IGV), for Splinkeretee PCR analysis<br>Timmomatic for total RNA Seq, and Splinkerette for PCR analysis<br>FASQC, for total RNA Seq (https://www.bioinformatics.babraham.ac.uk/projects/fastqc)<br>Kalliso, for total RNA Seq analysis<br>DESeq2, for total RNA Seq analysis<br>R, for total RNA seq (http://www.r-project.org) |

For manuscripts utilizing custom algorithms or software that are central to the research but not yet described in published literature, software must be made available to editors and reviewers. We strongly encourage code deposition in a community repository (e.g. GitHub). See the Nature Portfolio guidelines for submitting code & software for further information.

## Data

Policy information about availability of data

All manuscripts must include a data availability statement. This statement should provide the following information, where applicable:

- Accession codes, unique identifiers, or web links for publicly available datasets
- A description of any restrictions on data availability
- For clinical datasets or third party data, please ensure that the statement adheres to our policy

> The data for the teratoma-growth experiments (Figs. 2 and 4, and Extended Data Fig. 9) and for the blood-glucose experiments (Extended Data Fig. 10) are provided as Source Data. The raw and analysed datasets generated during the study are available for research purposes from the corresponding author on reasonable request.

## Research involving human participants, their data, or biological material

Policy information about studies with human participants or human data. See also policy information about sex, gender (identity/presentation), and sexual orientation and race, ethnicity and racism.

| | |
|---|---|
| Reporting on sex and gender | The study did not involve human participants. |
| Reporting on race, ethnicity, or other socially relevant groupings | — |
| Population characteristics | — |
| Recruitment | — |
| Ethics oversight | — |

Note that full information on the approval of the study protocol must also be provided in the manuscript.

# Field-specific reporting

Please select the one below that is the best fit for your research. If you are not sure, read the appropriate sections before making your selection.

☒ Life sciences          ☐ Behavioural & social sciences          ☐ Ecological, evolutionary & environmental sciences

For a reference copy of the document with all sections, see nature.com/documents/nr-reporting-summary-flat.pdf

# Life sciences study design

All studies must disclose on these points even when the disclosure is negative.

| | |
|---|---|
| Sample size | The number of samples used in different experiments was determined in accordance with similar studies in the field. |
| Data exclusions | No data were intentionally excluded. |
| Replication | Replication numbers and attempts were carried out as deemed necessary, and are reported in the paper. |
| Randomization | Samples were randomized when possible. |
| Blinding | Blinding was either not deemed necessary, owing to the detection method, or not possible owing to the nature of the experiment (such as growth of visible subcutaneous teratomas). |

# Reporting for specific materials, systems and methods

We require information from authors about some types of materials, experimental systems and methods used in many studies. Here, indicate whether each material, system or method listed is relevant to your study. If you are not sure if a list item applies to your research, read the appropriate section before selecting a response.

## Materials & experimental systems

| n/a | Involved in the study |
|-----|------------------------|
| ☐ | ☒ Antibodies |
| ☐ | ☒ Eukaryotic cell lines |
| ☒ | ☐ Palaeontology and archaeology |
| ☐ | ☒ Animals and other organisms |
| ☒ | ☐ Clinical data |
| ☒ | ☐ Dual use research of concern |
| ☒ | ☐ Plants |

## Methods

| n/a | Involved in the study |
|-----|------------------------|
| ☒ | ☐ ChIP-seq |
| ☐ | ☒ Flow cytometry |
| ☒ | ☐ MRI-based neuroimaging |

## Antibodies

| | |
|---|---|
| Antibodies used | Anti-mouse<br>anti-mouse CD3ε (BD Biosciences 553058)<br>anti-mouse CD28 (BD Biosciences 553295)<br>anti-mouse PD-L1 (alexa488, Novus NBP1-43262AF488)<br>anti-mouse CD47, (BV421, BD Biosciences 740055)<br>anti-mouse CD200 (alexa647, BD Biosciences 565544)<br>anti-mouse FASL (Novus NBP1-97519)<br>anti-mouse H2-M3 (BD Biosciences 551769)<br>anti-mouse SERPINB9 (HycultBiotech HP8035)<br>anti-mouse CCL21 (R&D AF457)<br>anti-mouse MFGE8 (Biolegend 518603)<br>anti-mouse CD31 (Biolegend 102417)<br>anti-mouse Neurofilament (Fisher Scientific MS359R7)<br>anti-mouse SSEA1 (Sigma MAB4301)<br>anti-mouse OCT4 (Santa Cruz sc-8628)<br>anti-mouse Smooth Muscle Actin (Abcam ab5694)<br>anti-mouse FOXA2 (Abcam ab108422)<br>anti-mouse SOX17 (R&D AF1924)<br>anti-mouse Troponin (cTNT, abcam, ab8295)<br>anti-mouse H-2Kb (Biolegend 116511)<br>anti-mouse H-2Kq (Biolegend 115106)<br><br>Anti-human<br>anti-human PDL1 (APC, Biolegend 329708)<br>anti-human FALSG (PE, Biolegend 306406)<br>anti-human CD200 (APC/Cy7, Biolegend 329220)<br>anti-human CD47 (PerCP/Cy5, Biolegend 323109)<br>anti-human SERPINB9 (ThermoFisher MA5-17648)<br>anti-human CCL21 (R&D AF366-SP)<br>anti-human MFGE8 (R&D IC27671A)<br>anti-human CD45 (APC, BD Biosciences 560973)<br>anti-human CD3 (Pe-Cy5, BD Biosciences 561006)<br>anti-human HLA-A,B,C (APC, Biolegend 311409)<br>anti-human CD31 (Pe-Cy7, Biolegend 303118)<br>anti-human CD144 (PE, Biolegend 138009)<br>anti-human CD16/CD32 Fc blocker (BD Biosciences 553141)<br>anti-human Human TruStain FxC Fc blocker (Biolegend 422301)<br><br>Secondaries and Others<br>anti mouse IgG (alexa488, ThermoFisher A10680)<br>anti goat IgG (alexa488 ThermoFisher A27012)<br>anti-Rat Fc (alexa568 secondary antibody (ThermoFisher A21112).<br>anti-hamster (Dylight 488 Biolegend 405503)<br>anti-rat IgG (DyLight 549 Novus NBP1-72975)<br>anti-mouse (alexa647 ThermoFisher 21244)<br>Annexin V (BV421, Biolegend 640923)<br>PI (Biolegend, 421301) |
| Validation | Antibody performance and validations are available from the vendor and/or manufacturer.<br><br>Experiments using antibodies included relevant negative biological controls, such as cells without transgene expression or undifferentiated cells. Antibodies relying on secondary antibodies also included secondary only-controls (no primary antibody). |

# Eukaryotic cell lines

Policy information about cell lines and Sex and Gender in Research

| | |
|---|---|
| Cell line source(s) | Mouse C67BL6/N C2 embryonic stem cells were derived in the Nagy lab and in The Centre for Phenogenomics. Human H1 embryonic stem cells were imported from WiCell. |
| Authentication | The cell lines were not authenticated before use. |
| Mycoplasma contamination | All cell lines and cultures were routinely tested for mycoplasma, and only negative cultures were used for experiments. |
| Commonly misidentified lines (See ICLAC register) | No commonly misidentified cell lines were used. |

# Animals and other research organisms

Policy information about studies involving animals; ARRIVE guidelines recommended for reporting animal research, and Sex and Gender in Research

| | |
|---|---|
| Laboratory animals | C57BL/6N (Strain 005304), C3H/HeJ (Strain 000659), FVB/NJ (Strain 001800), BALB/cJ (Strain 000651), and NSG mice (Stock 005557) were purchased from the Jackson Laboratory. CD1 (Stock 022) mice were purchased from Charles River. |
| Wild animals | The study did not involve wild animals. |
| Reporting on sex | Both male and female recipients were used in the study. |
| Field-collected samples | The study did not involve samples collected from the field. |
| Ethics oversight | All mouse housing, husbandy, and procedures were performed in compliance with the Animals for Research Act of Ontario and the Guidelines of the Canadian Council on Animal Care. The animal protocols used in this study were approved by the Toronto Center for Phenogenomics (TCP) Animal Care Committee. |

Note that full information on the approval of the study protocol must also be provided in the manuscript.

# Flow Cytometry

## Plots

Confirm that:

☒ The axis labels state the marker and fluorochrome used (e.g. CD4-FITC).

☒ The axis scales are clearly visible. Include numbers along axes only for bottom left plot of group (a 'group' is an analysis of identical markers).

☒ All plots are contour plots with outliers or pseudocolor plots.

☐ A numerical value for number of cells or percentage (with statistics) is provided.

## Methodology

| | |
|---|---|
| Sample preparation | To generate Klg-1, Klg-2, and cloaked human lines, transfected mouse or human ESCs were collected at 60–80% confluency, washed with FACS buffer (PBS + 1% BSA and 0.5% EDTA), stained with antibodies, resuspended in FACS buffer + 25mM HEPES, filtered into Falcon tubes with cell-strainer lids, and then FACS-sorted on a MoFlow Astrios EQ sorter. <br><br> Cells collected for Flow cytometry analysis only (no sorting) were prepared in an identical fashion, except the final suspension buffer did not contain 25mM HEPES. Cells were run on a GALLIOS flow cytometer. |
| Instrument | For cell-sorting experiments (FACS), a MoFlo Astrios EQ cell sorter (Beckman Coulter, Miami, FL, USA) was used, equipped with 305-nm, 405-nm, 488-nm, 561-nm and 640-nm lasers. <br><br> For analysis, a GALLIOS flow cytometer, Beckman Coulter, equipped with 405-nm, 488-nm, 561-nm and 640-nm lasers. |
| Software | FACS software: Summit v6.2 (Beckman Coulter); Analysis software (for Gallios data): Kaluza v1.5 (Beckman Coulter) and FlowJo V10 |
| Cell population abundance | All lines that were FACs-sorted were analysed by post-sort analysis and compared to controls to ensure that the desired population was sorted. Extended Data Fig. 5b shows representative gating and cell-abundance graphs for the desired cell population. |
| Gating strategy | All samples were first gated for live single cells using side-scatter pulse height and area along with a viability dye. Cells |

Gating strategy | without marker or factor expression, as well as single-color samples of the same cell type as the experimental samples, were used for negative controls to set gating and instrument compensation.

☒ Tick this box to confirm that a figure exemplifying the gating strategy is provided in the Supplementary Information.

