## [Peer Review File · Nature Biomedical Engineering]

Immune-privileged tissues formed from immunologically cloaked mouse embryonic stem cells survive long-term in allogeneic hosts

Corresponding author: Andras Nagy

Editorial note

This document includes relevant written communications between the manuscript's corresponding author and the editor and reviewers of the manuscript during peer review. It includes decision letters relaying any editorial points and peer-review reports, and the authors' replies to these (under 'Rebuttal' headings). The editorial decisions are signed by the manuscript's handling editor, yet the editorial team and ultimately the journal's Chief Editor share responsibility for all decisions.

Any relevant documents attached to the decision letters are referred to as **Appendix #**, and can be found appended to this document. Any information deemed confidential has been redacted or removed. Earlier versions of the manuscript are not published, yet the originally submitted version may be available as a preprint. Because of editorial edits and changes during peer review, the published title of the paper and the title mentioned in below correspondence may differ.

This manuscript was originally peer-reviewed at *Nature Biotechnology* before being transferred to *Nature Biomedical Engineering*.

Correspondence

Mon 26 Jun 2023

Decision on Article nBME-23-0930-T

Dear Dr Nagy,

Thank you again for your revised manuscript, "Engineering cells that form immune-privileged tissues and can survive in allogeneic, immune competent hosts", and for your patience with this round of peer review.

As noted in previous e-mail correspondence, two of the original reviewers recruited by *Nature Biotechnology* were unavailable for re-review or declined to re-review, and I recruited two new reviewers, whose reports are included at the end of this message (alongside the report from the original Reviewer #3, which I had already forwarded to you).

In their reports, you will see that Reviewer #3 offers a number of really useful points of advice for improving the focus and interpretability of the findings and claims, and that the new reviewers have a number of technical points that I am hoping you will be able to satisfactorily address. Regarding the experiments with the microfluidic device, in view of the comments by Reviewer #3, you may want to place this evidence in the Supplementary Information. As for point #5 from Reviewer #5, from an editorial standpoint we won't require evidence with humanized mice. Yet I hope you can satisfy point #6 by Reviewer #4 regarding the assessment of cloaked teratomas after GCV treatment.

When you are ready to resubmit your manuscript, please upload the revised files, a point-by-point rebuttal to the comments from all reviewers, the reporting summary, and a cover letter that explains the main improvements included in the revision and responds to any points highlighted in this decision.Please follow the following recommendations:

- * Clearly highlight any amendments to the text and figures to help the reviewers and editors find and understand the changes (yet keep in mind that excessive marking can hinder readability).
- * If you and your co-authors disagree with a criticism, provide the arguments to the reviewer (optionally, indicate the relevant points in the cover letter).
- * If a criticism or suggestion is not addressed, please indicate so in the rebuttal to the reviewer comments and explain the reason(s).
- * Consider including responses to any criticisms raised by more than one reviewer at the beginning of the rebuttal, in a section addressed to all reviewers.
- * The rebuttal should include the reviewer comments in point-by-point format (please note that we provide all reviewers will the reports as they appear at the end of this message).
- * Provide the rebuttal to the reviewer comments and the cover letter as separate files.

We hope that you will be able to resubmit the manuscript within 15 weeks from the receipt of this message. If this is the case, you will be protected against potential scooping. Otherwise, we will be happy to consider a revised manuscript as long as the significance of the work is not compromised by work published elsewhere or accepted for publication at *Nature Biomedical Engineering*.

We look forward to receive a further revised version of the work. Please do not hesitate to contact me should you have any questions.

Best wishes,

Pep

Pep Pàmies
Chief Editor, Nature Biomedical Engineering

Reviewer #3 (Report for the authors (Required)):

This manuscript, by Harding and colleagues, documents a monumental effort to develop a cellular system capable of evading the immune and inflammatory injury expected to arise in a foreign host, that is, rejection. Many others have pursued the same objective by matching histocompatibility alleles or eliminating histocompatibility antigens, by implantation of grafts in “privileged” sites, by induction of tolerance, by physical encapsulation among many others, and initial success was reported (usually in mice), but ultimately none of the countless approaches tested over the past century reliably enabled permanent engraftment of foreign tissue in fully immune competent humans or large animals. The approach described by Harding et al. however differs in several important respects. First, the approach is completely rational, which is to say it is multifaceted, countering cells critical to two or three pathways thought to underlie rejection of cell and tissue allografts. Second, the countering of cellular elements (antigen presenting cells, macrophages, T and NK cells) is mediated locally, in the graft, not systemically. Third, and most importantly, the protection conferred by the authors’ approach not only averts rejection of engineered cells but also extends to neighboring un-engineered cells and tissues.

The approach of Harding et al. consists of stable transfection of embryonic stem cells (ES cells) with constructs encoding eight proteins and an inducible thymidine kinase construct to suppress untoward proliferation. The specific approach might or might not prove directly applicable in clinical settings. One is loath to assume that ES cells engineered with multiple inserts would be embraced by regulatory agencies. However, there is no obvious reason why the approach could not be adapted to other, more acceptable cell

types. Regardless, of whether the approach applied exactly as described in this manuscript can meet regulatory hurdles, the findings suggest a direction to achieving effective and unintrusive alloengraftment and it takes little imagination to envision future applications. Whether Harding et al. have finally found a way to reliably avert rejection of allogeneic cells and tissues by unmodified (un-immunosuppressed) recipients or whether this approach like others before it will prove less effective than a first glimpse in mice suggests remains to be seen. The only way to address that question is to communicate the findings and put further inquiry into the hands of others.

The manuscript does have some limitations and flaws. These are relatively minor and should not lead to addition of more data but should motivate revision of some rhetoric in the manuscript.

Specific comments:

1. The size and vast scope of the manuscript will discourage many from thoroughly examining the content. A greater “concern” however is that the most important finding, protection of un-engineered cells and tissues presented by transplanted engineered cells does not appear until the last section of the Results (e.g., depicted in Figure 6C and in “extended data” Figure 10). This concern should not lead to reorganization of results but rather to revision of the Abstract and Introduction as described below.
2. The authors should revise the Abstract and Introduction to clearly state the challenge and specifically alert readers to the most novel findings and challenges overcome. For example, the abstract claims: “What is needed are strategies to “cloak” therapeutic cells.” That claim is imprecise and misleading. There already exist plenty of such strategies. Hundreds, perhaps thousands of engineers currently work on encapsulation techniques for allogeneic cells and tissues (not just for cells) and some of these techniques work as well in mice as the approach of Harding et al. Similarly, overuse of the term “cloak” implies the authors know the mechanism, but the mechanism is not proved and not even clear (see below). What is clear is that the transduced cells are not rejected by allogeneic recipients and more importantly appear to avert rejection of neighboring cells and tissues (isolated islets are tissues not cells). Readers informed about this advance in the Abstract and the Introduction should prepare readers as well.
3. The Introduction begins with a poorly worded assertion: “...cell products sourced from a foreign donor will be immune rejected unless patients are treated with toxic and potentially dangerous immunosuppressive drug,” as if to suggest the manuscript will report a solution that is demonstrably non-toxic and not dangerous. The authors did not test the toxicity of the engineered cells (or products thereof), especially whether the cells exert immunosuppressive effects on mice. That engrafted mice reject “uncloaked” tissues does not address that concern since individuals with significant levels of immunodeficiency reject allografts and since the modified cells might act in part by secretion of biologically active substances. This concern is NOT a demand for more experiments, but rather a suggestion the authors thoughtfully revise assertions made in the introduction (the claim also potentially insults readers working in encapsulation or isolating proteins or other products from foreign cells) and include a relatively non-toxic impact on immunity among the potential mechanisms.
4. Neither the abstract nor the introduction prepare readers for the most novel aspect of the report – the apparent protection the engineered cells confer on un-manipulated neighboring cells and tissues. None of the “experiments of nature” listed by the authors such as the fetus or parasite, and none of the other therapeutic approaches listed truly manifest this property. The fetus has a completely separate circulation and grafts placed adjacent to placenta are rejected; likewise, parasites evade immunity by various means but confer no protection on neighboring tissues. Put in another way, the absence of rejection of engineered ES cells is interesting but achievable by other methods (the authors cite a few of many reported in mice); protection of neighboring cells however distinguishes the results in this manuscript from all prior work. This impact warrants specific mention and emphasis in the abstract and the potential significance should be clearly set up in the Introduction.
5. The authors should/must make every effort to clearly indicate the number of experiments, conditions and/or animals used. Sometimes the number is clearly mentioned but too often the reader must search methods, figures and legends and sometimes no specific number is given. Particularly irksome was failure to mention the numbers of transplants, such as islet transplants.
6. The experiment examining interaction between human PBMCs and engineered endothelial like cells in a microfluidic model has dubious relevance to application of the technology. The kinetics of injury/detachment reported in the manuscript take place in a few hours whereas alloimmune injury prevented by cloaking in cell

and tissue grafts prevents is never evident sooner than days and, in this manuscript, not until weeks have elapsed. Although the mechanism of this in vitro effect was not elucidated (not should it be), it most likely reflects the action of proteases and other enzymes released from granules of phagocytes. The effect observed in the flow model is real, but the interpretation (actually, the lack of interpretation) might give the false impression that it represents destruction of an allotransplant, but it does not. In fact, the initial engraftment of cell and tissue grafts (both engineered and non-engineered allogeneic cells shown in earlier figures in the mouse) is facilitated by the secreted products that likely compromise attachment of EC in the flow model. Thus, differences between engineered and non-engineered cells in this model, while real, have a narrow and limited significance distinct from the thrust of the manuscript. Another concern is that survival of cell and tissue grafts (until rejection begins) depends on the in-growth of capillaries that derive from the recipient; hence both capillaries and PBMC are from the same source in biological settings. I think the model and results shown (Figure 5) contribute nothing to a resplendently documented and illustrated report. If the results are retained, the description and interpretation should be narrowed.

7. The manuscript contains a number of careless expressions and/or inaccurate claims besides several mentioned above. For example, in the abstract the authors claim cells were engineered “to autonomously prevent immune rejection in allogeneic recipients.” Putting aside the split infinitive, what does “autonomously” mean? After a typographical error in the next sentence the authors next claim to have “induced formation of a dormant, artificial tissue.” What is that? and would not one want to avoid rejection in autologous recipients? After omission of a comma in an enumerated list of genes in the next sentence, the authors claim to produce “a dormant, artificial tissue,” and then an “immune privileged artificial tissue.” What is meant by “dormant” and by “artificial?” The authors later claim that the “integrated FailSafe™ kill switch gives total control over the proliferation,” but that is not shown. What is shown is that ES cells will generate tumors in which growth is limited to a size that compromises well-being of mice. “Total control” is not shown or even tested. The extent of control is impressive, exaggeration undermines credibility. I have marked many errors throughout the manuscript, which can be improved by copy editing.

Reviewer #4 (Report for the authors (Required)):

Harding et al described an innovative and interesting approach to induce immune tolerance to grafts by introducing into ES cells eight genes thought to be involved in immune evasion of pathogens. Furthermore, by generating teratomas from ES cells transfected with the eight genes and the HSV-TK gene, and by creating an immunological sanctuary under the mouse skin in the presence of GCV, the authors were able to avoid immunological rejection of xenografts and allografts. The methodology presented by the authors is novel and the demonstrated effects are indeed interesting. However, there are some concerns regarding the lack of mechanistic analysis to support the observed experimental findings.

Major Comments:

1, We find it remarkable that even in the presence of remaining MHC, complete avoidance of allogeneic responses can be achieved solely through gene modification. However, the necessity of each individual element has not been adequately demonstrated. Are all eight elements essential?

2, It would be beneficial for the authors to compare this approach using MHC edited PSCs to highlight the advantages of their eight gene modification strategy and address the potential issues they perceive. If the authors state that ES cells modified with the eight genes pose a lower risk of disease persistence upon viral infection or tumorigenesis compared to MHC-deficient ES cell-derived cells, it is worth to know whether the eight genes prevent rejection in the presence of MHC-antigen presentation or not. In other word, the authors are asked to demonstrate that antigen-presenting cells derived from ES cells modified with the eight genes can be eliminated by CD8 T cells in MHC-restricted and antigen specific manner.

3, The methodology of selectively eliminating cells in the cell cycle using Cdk1-TK to create an immune tolerant container for immunogenic cells is innovative. However, there is a lack of information regarding the histopathological characteristics and cell biological mechanisms of this structure, resulting in a predominance of phenomenology. Therefore, the discussion regarding the clinical advantages remains too much speculative.

Minor Questions:

- 1, Could the introduction of genes coding secretion proteins have any detrimental effects in situations that do not require immunological tolerance on another cells?
- 2, In Figure 2b, why is the engraftment of Klg-1 teratomas more difficult in B6 mouse rather than in the other strain?
- 3, Why do the engraftment frequencies of Klg-1 teratomas so differ among mouse strains, as shown in Figure 2e? Were the results expected?
- 4, The explanation of the results in Figure 4J is insufficient. If apoptosis through FASL is considered, it would be necessary to conduct co-culture experiments with individual T cell subsets and evaluate apoptotic sensitivity of these T cell subsets.
- 5, The specific subset of PBMCs that shows increased or decreased reactivity in Figure 5 is not indicated. To confirm the experimental setting whether NK cell tolerance is induced, it is necessary to provide HLA information of the experiment, in addition, the experimental observation should be confirmed with multiple donors.
- 6, To demonstrate whether cloaked teratomas after GCV treatment can avoid humoral immune responses and protect cells inside of the container is important. So, I would recommend conducting an experiment where non-cloaked teratomas are initially immunized to induce anti-MHC antibodies in FVB mice and then transplanting cloaked ES to make immune-tolerant teratomas with GCV treatment. Furthermore, it would be advisable to confirm whether non-cloaked cells injected into the cloaked immune-tolerant structures can survive avoiding antibody-mediated attack.

Reviewer #5 (Report for the authors (Required)):

The authors prepared universal mESCs and hESCs using transposon-based overexpression of eight immunomodulatory transgenes in this manuscript. The selection of eight immunomodulatory transgenes is original work in this manuscript, although universal or hypoinmunogenic pluripotent stem cells have been reported by over expression of immunomodulatory transgenes such as PDL1 and CD47 (Q. Ye et al., Generation of universal and hypoinmunogenic human pluripotent stem cells, *Cell Proliferation*, 53 (2020) e12946). This method is a kind of gene therapy, which is extremely difficult to be accepted and approved by FDA, although some immunotherapy has been approved currently. There is a report to generate universal hiPSCs without using gene transduction (TC Sung et al., Transient characteristics of universal cells on human-induced pluripotent stem cells and their differentiated cells derived from foetal stem cells with mixed donor sources, *Cell Proliferation*, 54(3) (2021) e12995). The authors should write the merit of gene transduction method of this study together citing the above literature. The evaluation of immune privilege of the author's cells are reasonable in this study. However, when they prepare human ESCs or hiPSCs by overexpression of immunomodulatory transgenes, it is more difficult to study the immune-privilege evaluation. The challenging to generate universal human pluripotent stem cells and their universal and differentiated cells are important, the reviewer recommend the publication of this manuscript in *Nature Biomed Engineering* after revision of the following comments.

Specific comments:

1. On Line 70, "which remove the major source of antigen mismatch between donor and recipients5-9." There is an excellent review, which explain this issue. The authors may cite the following article; "Q. Ye et al., Generation of universal and hypoinmunogenic human pluripotent stem cells, *Cell Proliferation*, 53 (2020) e12946".
2. The authors should write the merit of gene transduction method of this study.
3. The authors used piggyBac and Sleeping Beauty transposon (cloaking) vectors in which transcription was driven by a CAG promoter. Why they did not select to use CRISPR/Cas-9 genome editing method?
4. On L-235; "These vectors were then transfected into a FS H1 human ESC (hESC) line" should be revised as "These vectors were then transduced into a FS H1 human ESC (hESC) line".

5. Immune-privilege study of FS hESCluc+ was performed using NSG mice. Most of researchers are using immune-privilage experiments using humanized mice. It is necessary to show the immune-privilege results using humanized mice.

Wed 02 Aug 2023

Decision on Article NBME-23-0930A

Dear Dr Nagy,

Thank you for your revised manuscript, "Engineering cells that form immune-privileged tissues and can survive in allogeneic, immune competent hosts". Having consulted with the reviewers (whose comments you will find at the end of this message), I am pleased to write that we shall be happy to publish the manuscript in *Nature Biomedical Engineering*.

We will be performing detailed checks on your manuscript, and in due course will send you a checklist detailing our editorial and formatting requirements. You will need to follow these instructions before you upload the final manuscript files. In the meantime, please do consider the useful comments and suggestions from Reviewers #3 and #4.

Best wishes,

Pep

Pep Pàmies
Chief Editor, Nature Biomedical Engineering

Reviewer #3 (Report for the authors (Required)):

The authors effectively incorporated my suggestions and pertinent suggestions of other reviewers in a revised manuscript. Below I list a few minor suggestions for consideration if further revision of the manuscript is requested. These points could be addressed at the authors' discretion in proof.

1. In the abstract – "...abrogated activation of human hematopoietic cells" is imprecise and incomplete. The expressed genes appear to avert destruction of foreign grafts by preventing activation or eliminating immune responder cells and by protecting foreign cells from immune and inflammatory injury.---The transgenes are not acting on human hematopoietic cells.
2. The extended critique of iPSC-based therapeutics might be correct but it is tangential, at best. If the manuscript were to be further revised, the authors should consider eliminating most of the first paragraph (all but the first sentence) of the Introduction to avoid unnecessarily alienating tens of thousands of people endeavoring to advance iPSC-based therapies and immunosuppressive drugs. The merit of the authors' technology speaks for itself and is not made more attractive by vague criticism of other technologies, especially since the authors have not systematically compared the merits and toxicities of other technologies with their own.
3. "...universal" cell lines to derive therapeutic products for the treatment of many MHC-mismatched patients"---this phrase makes no sense and the implicit claim regarding universal cell lines is imprecise and easily attacked. The authors' approach prevents rejection commendably well and establishes what appear to be immune privileged sites. The approach does not support or advance use of universal cell lines or even explain what is meant by that term. Is a cell line deemed "universal" because it is accepted by all recipients or is it universal because it can be differentiated into any mature tissue? The authors infer the former but mention as such is redundant. Also potentially redundant and unclear is the expression: "treatment for MHC mismatched patients." What are MHC-mismatched patients? The authors probably mean that the grafts are not matched with recipients but one usually reserves use of MHC matching and mismatching to circumstances in which there is a real donor and recipient. If the manuscript is revised the authors to delete both claims.

I should also mention a few points of departure from comments of other reviewers.

4. One reviewer asks whether all modifications are necessary and suggests requirements must be tested. The question is important but premature. Once this set of modifications is reported, I suspect the authors and others will undertake testing in various applications and it might well be found that different combinations or levels of expression are needed for protection of different tissues from rejection. One might well find that different levels of expression are needed to establish an immune privileged site that to prevent rejection of transduce cells.

5. One reviewer suggests the authors should compare their approach to results obtained using MHC-deficient ES-derived cells. I disagree. Efforts to promote engraftment by targeting or blocking one or more MHC encoded proteins have been pursued to little benefit for decades. Besides concerns about infection, which certainly are valid, MHC deficient cells are subject to injury by NK cells and other cells and products of MHC and related genes that are expressed in aberrant form are quite immunogenic. However, the reviewer appears to have raised the point because the authors unwisely compared their approach to MHC targeting.

6. One reviewer discusses anti-MHC antibody responses to engineered cells and recommends further experiments to test the impact of such antibodies. Although anti-MHC antibodies might be assayed as one measure and perhaps the most sensitive measure of immunogenicity, I disagree about the imperative for such testing. The preponderance of work over the last 80 years exploring the impact of antibodies on transplants indicates anti-MHC antibodies important for the outcome of organ transplants have little or no detrimental impact on cell and tissue transplants. The observations presented in the manuscript are fully consistent with that experience – if anti-MHC antibodies are made, they evidently cause little or no damage. As for the merit of testing, I might find the results interesting but not illuminating without much more information than most can imagine.

7. One reviewer suggests the authors show immune privilege in humanized mice. I disagree. Humanized mice have many valid uses, but the mice are aberrant in many respects and might or might not faithfully model immune privilege. More importantly, the human T cells generated and surviving in humanized mice are far less competent than murine T cells in wild type mice.

Reviewer #4 (Report for the authors (Required)):

I will provide comments on the authors' responses and revisions to each of my points.

Major comments

#1 I understand that this research is based on the hypothesis that in a long-term xenotransplantation model, multiple complementary immune evasion mechanisms would be necessary to completely avoid immune rejection. I also understand the caution to avoid suggesting that all eight factors are essential and instead demonstrate that these eight factors are sufficient to achieve the objective. I accept that the analysis of the impact of each factor on immune evasion is beyond the scope of this paper.

#2 I understand that in past revisions, experiments and discussions related to this topic were conducted but were removed due to editorial reasons. I also understand the assertion that careful experiments, such as investigating antigen-specific viral clearance, should be left for future characterization studies. I believe that this assertion could be included in the Discussion section.

#3 I appreciate the authors' approach of not overly emphasizing clinical applications in this study and instead focusing on conducting more detailed research and experiments to strengthen the scientific foundation and provide a more solid basis for the potential clinical benefits of the approach. Readers will quickly recognize that this research includes valuable insights that evoke clinical advantages. Therefore, it is important to include cautious statements that accurately convey the current understanding.

Minor comments

#1-6 I completely agree with the authors' thoughtful and careful responses.

Reviewer #5 (Report for the authors (Required)):

The authors revise the manuscript appropriately. No further comments.

Rebuttal 1

Response from authors:

We are extremely grateful for the reviewer feedback, especially for the helpful ideas on how to re-frame the story in the abstract and introduction to better prepare the reader what to expect from the results.

We think, and exactly as suggested, this has helped us clarify what is important and new about our findings in the context of the field. It also brought additional focus to the paper and which experiments or data are best kept for future studies.

Note, any reference to Figure numbers refer to the *updated* arrangement, where main Figure 5 has been moved to the extended data section. Now there are Main Figure 1-5 and Extended data 1-11.

Additionally, any changes in the new upload are highlighted in light gray, such as the highly revised abstract and introduction.

Point-by-point responses from Authors

Reviewer #3:

This manuscript, by Harding and colleagues, documents a monumental effort to develop a cellular system capable of evading the immune and inflammatory injury expected to arise in a foreign host, that is, rejection. Many others have pursued the same objective by matching histocompatibility alleles or eliminating histocompatibility antigens, by implantation of grafts in “privileged” sites, by induction of tolerance, by physical encapsulation among many others, and initial success was reported (usually in mice), but ultimately none of the countless approaches tested over the past century reliably enabled permanent engraftment of foreign tissue in fully immune competent humans or large animals. The approach described by Harding et al. however differs in several important respects. First, the approach is completely rational, which is to say it is multifaceted, countering cells critical to two or three pathways thought to underlie rejection of cell and tissue allografts. Second, the countering of cellular elements (antigen presenting cells, macrophages, T and NK cells) is mediated locally, in the graft, not systemically. Third, and most importantly, the protection conferred by the authors’ approach not only averts rejection of engineered cells but also extends to neighboring un-engineered cells and tissues.

The approach of Harding et al. consists of stable transfection of embryonic stem cells (ES cells) with constructs encoding eight proteins and an inducible thymidine kinase construct to suppress untoward proliferation. The specific approach might or might not prove directly applicable in clinical settings. One is loath to assume that ES cells engineered with multiple inserts would be embraced by regulatory agencies. However, there is no obvious reason why the approach could not be adapted to other, more acceptable cell types. Regardless, of whether the approach applied exactly as described in this manuscript can meet regulatory hurdles, the findings suggest a direction to achieving effective and unintrusive alloengraftment and it takes little imagination to envision future applications. Whether Harding et al. have finally found a way to reliably avert rejection of allogeneic cells and tissues by unmodified (un-immunosuppressed) recipients or whether this approach like others before it will prove

less effective than a first glimpse in mice suggests remains to be seen. The only way to address that question is to communicate the findings and put further inquiry into the hands of others.

The manuscript does have some limitations and flaws. These are relatively minor and should not lead to addition of more data but should motivate revision of some rhetoric in the manuscript.

Specific comments:

1. The size and vast scope of the manuscript will discourage many from thoroughly examining the content. A greater “concern” however is that the most important finding, protection of un-engineered cells and tissues presented by transplanted engineered cells does not appear until the last section of the Results (e.g., depicted in Figure 6C and in “extended data” Figure 10). This concern should not lead to reorganization of results but rather to revision of the Abstract and Introduction as described below.

Again, we thank the reviewer for this and related comments 1-4, which encourages us to highlight and better prep the reader for our results demonstrating that immune privileged tissue can protect neighboring, unmodified cells and tissues. Especially since we agree that these data separate our work from others in the field working on encapsulation or engineering hypoimmunogenic cells only.

We have completely revised the abstract and introduction with the goal of removing extraneous information to highlight what is important about the finding.

2. The authors should revise the Abstract and Introduction to clearly state the challenge and specifically alert readers to the most novel findings and challenges overcome. For example, the abstract claims: “What is needed are strategies to “cloak” therapeutic cells.” That claim is imprecise and misleading. There already exist plenty of such strategies. Hundreds, perhaps thousands of engineers currently work on encapsulation techniques for allogeneic cells and tissues (not just for cells) and some of these techniques work as well in mice as the approach of Harding et al. Similarly, overuse of the term “cloak” implies the authors know the mechanism, but the mechanism is not proved and not even clear (see below). What is clear is that the transduced cells are not rejected by allogeneic recipients and more importantly appear to avert rejection of neighboring cells and tissues (isolated islets are tissues not cells). Readers informed about this advance in the Abstract and the Introduction should prepare readers as well.

Please see our highly revised introduction and abstract which we hope addresses these important concerns.

3. The Introduction begins with a poorly worded assertion: “...cell products sourced from a foreign donor will be immune rejected unless patients are treated with toxic and potentially dangerous immunosuppressive drug,” as if to suggest the manuscript will report a solution that is demonstrably non-toxic and not dangerous. The authors did not test the toxicity of the engineered cells (or products thereof), especially whether the cells exert immunosuppressive effects on mice. That engrafted mice reject “uncloaked” tissues does not address that concern since individuals with significant levels of immunodeficiency reject allografts and since the modified cells might act in part by secretion of biologically active substances. This concern is NOT a demand for more experiments, but rather a suggestion the authors thoughtfully revise assertions made in the introduction (the claim also potentially

insults readers working in encapsulation or isolating proteins or other products from foreign cells) and include a relatively non-toxic impact on immunity among the potential mechanisms.

The reviewer is that we did not specifically test the potential cytotoxicity of our “cloaked” cells and tissue on t recipient, other than that engrafted mice could reject uncloaked cells subsequently injected at a distal side.

Among the changes in our highly revised introduction, we removed the phrase “toxic and potentially dangerous immunosuppressive drug,” to avoid misleading readers about the experiments were done.

We also strongly agree on the importance of acknowledging the work of others in the field, including encapsulation, many examples of which show promising results, especially in mice. We have also added references on encapsulation studies (Lines 400 and 402). Still, as far as we understand, we are the first to demonstrate that an engineered cell line can generate long-term accepted, allogeneic tissues in several MHC-mismatched, immunocompetent strains. We believe that, even within the field working towards engineering hypoimmunogenicity, this is significant advance.

4. Neither the abstract nor the introduction prepare readers for the most novel aspect of the report – the apparent protection the engineered cells confer on un-manipulated neighboring cells and tissues. None of the “experiments of nature” listed by the authors such as the fetus or parasite, and none of the other therapeutic approaches listed truly manifest this property. The fetus has a completely separate circulation and grafts placed adjacent to placenta are rejected; likewise, parasites evade immunity by various means but confer no protection on neighboring tissues. Put in another way, the absence of rejection of engineered ES cells is interesting but achievable by other methods (the authors cite a few of many reported in mice); protection of neighboring cells however distinguishes the results in this manuscript from all prior work. This impact warrants specific mention and emphasis in the abstract and the potential significance should be clearly set up in the Introduction.

The novelty of the immune-privileged tissue has now been better emphasized in the revised introduction and abstract.

We used naturally-evolved mechanisms of escape, such as transmissible cancers and the fetus, as general motivation for our approach to “cloaking”. However, we completely agree with the reviewer that our own experiments, which generated artificial immune privileged tissues, are novel even compared to these natural mechanisms.

5. The authors should/must make every effort to clearly indicate the number of experiments, conditions and/or animals used. Sometimes the number is clearly mentioned but too often the reader must search methods, figures and legends and sometimes no specific number is given. Particularly irksome was failure to mention the numbers of transplants, such as islet transplants.

We have gone through to make the number of animals (especially in transplant experiments) and experiment more visible. These changes are highlighted in grey throughout the Main text, figures, legends, etc.

Allogeneic transplants (Data in Fig. 2). Main text lines 123-124,137, 141-142, 147 and Fig. 2 legend lines 902-903, 904, 910

RNA seq experiments. Main text lines 167-169 (with wording clarified as contained a typographical error previously), Main Fig. 3 legend line 923, Suppl. table legend 4 lines 1135-1136

Immune privileged / Islet transplants; Main text lines 278-279, 294, 301, 306, and extended data figure 11 legend lines 1116, 1119. (No changes to main figure 5 legend as already indicates transplant number in line 976-977).

6. The experiment examining interaction between human PBMCs and engineered endothelial like cells in a microfluidic model has dubious relevance to application of the technology. The kinetics of injury/detachment reported in the manuscript take place in a few hours whereas alloimmune injury prevented by cloaking in cell and tissue grafts prevents is never evident sooner than days and, in this manuscript, not until weeks have elapsed. Although the mechanism of this in vitro effect was not elucidated (not should it be), it most likely reflects the action of proteases and other enzymes released from granules of phagocytes. The effect observed in the flow model is real, but the interpretation (actually, the lack of interpretation) might give the false impression that it represents destruction of an allotransplant, but it does not. In fact, the initial engraftment of cell and tissue grafts (both engineered and non-engineered allogeneic cells shown in earlier figures in the mouse) is facilitated by the secreted products that likely compromise attachment of EC in the flow model. Thus, differences between engineered and non-engineered cells in this model, while real, have a narrow and limited significance distinct from the thrust of the manuscript. Another concern is that survival of cell and tissue grafts (until rejection begins) depends on the in-growth of capillaries that derive from the recipient; hence both capillaries and PBMC are from the same source in biological settings. I think the model and results shown (Figure 5) contribute nothing to a resplendently documented and illustrated report. If the results are retained, the description and interpretation should be narrowed.

We take the reviewer's point that this main figure detracts from the other more relevant data sets and figures.

We have moved it to the supplemental.

7. The manuscript contains a number of careless expressions and/or inaccurate claims besides several mentioned above. For example, in the abstract the authors claim cells were engineered "to autonomously prevent immune rejection in allogeneic recipients." Putting aside the split infinitive, what does "autonomously" mean? After a typographical error in the next sentence the authors next claim to have "induced formation of a dormant, artificial tissue." What is that? and would not one want to avoid rejection in autologous recipients? After omission of a comma in an enumerated list of genes in the next sentence, the authors claim to produce "a dormant, artificial tissue," and then an "immune privileged artificial tissue." What is meant by "dormant" and by "artificial?" The authors later claim that the "integrated FailSafe™ kill switch gives total control over the proliferation," but that is not shown. What is shown is that ES cells will generate tumors in which growth is limited to a size that compromises well-being of mice. "Total control" is not shown or even tested. The extent of control is impressive, exaggeration undermines credibility. I have marked many errors throughout the manuscript, which can be improved by copy editing.

We carefully reviewed the manuscript to clean up imprecise language and done more copy-editing to correct typos, etc. We have tried to make things more precise and concrete, taking the reviewer's concerns about the original abstract and introduction seriously. Hopefully, our completely revised versions will address many of the issues regarding any imprecise or unclear framing of the work.

Reviewer #4

Harding et al described an innovative and interesting approach to induce immune tolerance to grafts by introducing into ES cells eight genes thought to be involved in immune evasion of pathogens. Furthermore, by generating teratomas from ES cells transfected with the eight genes and the HSV-TK gene, and by creating an immunological sanctuary under the mouse skin in the presence of GCV, the authors were able to avoid immunological rejection of xenografts and allografts. The methodology presented by the authors is novel and the demonstrated effects are indeed interesting. However, there are some concerns regarding the lack of mechanistic analysis to support the observed experimental findings.

Major Comments:

1, We find it remarkable that even in the presence of remaining MHC, complete avoidance of allogeneic responses can be achieved solely through gene modification. However, the necessity of each individual element has not been adequately demonstrated. Are all eight elements essential?

We certainly appreciate the question of whether all 8 transgenes are essential. We selected these transgenes based on their known function and mechanisms of action; i.e., interactions with dendritic cells, macrophages, T-cells, and NK-cells, involving antigen recognition, presentation, initiation of adaptive immunity, cell clearance, and cell-mediated cytotoxicity. Given the complexity of the mammalian immune system, we hypothesized from the onset of our work that it would take many, mutually-reinforcing immune transgenes to completely block immune rejection in a long-term allotransplant model using fully immune competent recipients. We think this is a major lesson coming from naturally evolved immune-escaper; it will not be simple, nor is it likely that just one or two immunomodulatory factors will do the job.

The combined expression of each factor defines a point in a kind of eight-dimensional space, and there may be many "rejection-resistant" points among this space with varying expression levels of the factors. We also expect that other combinations, with more or less factors, may also do the same.

In our own 8-factor system, we can only wish to have the workforce, funding, and time to tackle the challenge of defining the exact contribution and necessity of each transgene in such a large multi-transgene web. It would be a massive combinatorial undertaking. We tried to be careful in the manuscript to avoid suggesting that every factor was required. Instead, we proposed that our 8 factor approach was *sufficient* as an engineering approach, and additionally, that our methodology for transgene expression, utilizing transposons coupled with enrichment and clonal screening, was important. Our data suggest that all else being equal, high expression levels may be important.

2, It would be beneficial for the authors to compare this approach using MHC edited PSCs to highlight the advantages of their eight gene modification strategy and address the potential issues they perceive. If the authors state that ES cells modified with the eight genes pose a lower risk of disease persistence upon viral infection or tumorigenesis compared to MHC-deficient ES cell-derived cells, it is worth to know whether the eight genes prevent rejection in the presence of MHC-antigen presentation or not. In other word, the authors are asked to demonstrate that antigen-presenting cells derived from ES cells modified with the eight genes can be eliminated by CD8 T cells in MHC-restricted and antigen specific manner.

In an earlier version of our manuscript, we speculated in the discussion that overexpressing immune transgenes only – without deleting MHC genes as many others are doing - could leave cloaked cells with the ability to present antigens in the case of infection. We framed this as one benefit to our approach.

It is plausible that even in a local immune suppressive environment, certain infections could activate immune responses against cells that had previously been tolerated. Infection may be a path to clearance in some immune-suppressive environments ¹. It also provides a rational for Oncolytic Virus therapies ², which suggest that tolerogenic states can be achieved and then later broken by certain triggers.

We speculated the same could be true of our cloaked cells. However, based on previous feedback on the manuscript, we removed these discussion points since it was not the focus of the work, and we did not test them. We also did not want to mislead readers into thinking that we had tested these hypotheses.

We hope the reviewer will agree that since these points were removed from the manuscript and are not the focus of our proof-of-principle studies, these thoughtful experiments investigating antigen-specific viral clearance of cloaked cells are best left to future characterizations, which we are currently planning.

Separately, we believe an advantage of a transgene-only approach is the potential to make expression of the factors inducible, such as with the Tet-ON system. This would allow expression to be lowered or abolished if needed. While we did not generate these plasmids, we have left this point in the discussion.

3, The methodology of selectively eliminating cells in the cell cycle using Cdk1-TK to create an immune tolerant container for immunogenic cells is innovative. However, there is a lack of information regarding the histopathological characteristics and cell biological mechanisms of this structure, resulting in a predominance of phenomenology. Therefore, the discussion regarding the clinical advantages remains too much speculative.

Teratomas are intrinsically and highly heterogenous tissues. They contain many terminally differentiated cells, but also undifferentiated ones. When they form in the mouse, they continue to proliferate and expand until the mouse needs to be euthanized (see Main Fig. 2d top graph for unconstrained teratoma growth). The efficacy of the FailSafe™ system was demonstrated by it's ability to completely halt the growth of even these highly proliferative tissues *in vivo* (original publication; ³). We referred to these as “dormant” or “stabilized” teratomas.

These are highly vascularized, and easily-accessible subcutaneous tissue. This makes them experimentally easy and practical to engraft cells (needle injection, Main Fig. 5a,b,c) and tissues (small surgical excision, Extended Data Fig. 11c,d) inside of. The persistence of engrafted cells can also monitored by BLI. Furthermore, since it is a well-encapsulated tissue, the entire thing can be easily

removed, which allowed us to demonstrate that the source of a therapeutic effect was the graft (see Extended Data Fig. 11)

We acknowledge, however, that even dormant teratomas, due to their inherent heterogeneity, are not ideal for use in clinical applications. We have used them primarily as an experimental model to demonstrate the feasibility of creating an immunoprivileged environment using cloaked cells.

In a clinical setting, the goal would be to develop a more controlled and homogenous tissue or organ, which would provide a safe and effective way to protect transplanted cells and tissues from immune attack.

While the development of such a system may be a future challenge, we believe our studies provide a critical first step in this direction and open new avenues for further research and development.

We appreciate the reviewer's constructive criticism and have revised our manuscript to more clearly communicate these points.

Minor Questions:

1, Could the introduction of genes coding secretion proteins have any detrimental effects in situations that do not require immunological tolerance on another cells?

Our goal was to make cells that resist rejection without compromising the systemic immunity of the host. In our search for candidate factors, we focused only on local acting-factors. This includes transmembrane (PDL1, CD200, CD47, FASL, HLAG/H2-M3), cytosolic (SERPINB9), as well as two secreted factors, CCL21, and MFGE8. CCL21 highly and endogenously expressed the lymphatics and lymph nodes. As a chemokine, its function is generally linked to its point source expression, as shown by the lymph nodes, where cells are affected as they get closer. MFGE8 can bind to local phosphatidylserine and influence the response to apoptotic cells during opsonization.

We assume that in principle, secreted factors have a higher chance to compromise the systemic environment than transmembrane or cytosolic factors. Yet many secreted factors, like chemokines, do not necessarily do so. It will certainly depend on the factor.

Our data demonstrate that allogeneic recipients which have an existing, long-term teratoma derived from cloaked cells (Main Fig. 5a) – and which expresses all 8 factors including the two secreted (Main Fig. 3a) – can still reject unmodified, allogeneic cells. These data show that the host's immunity still remains intact enough to carry out innate and/or adaptive immune rejection at sites distal from the cloaked transplant.

2, In Figure 2b, why is the engraftment of Klg-1 teratomas more difficult in B6 mouse rather than in the other strain?

From our experience, we have noticed that some of the bioluminescent signal is always absorbed by the dark coat of the B6 mice. Consequently, even when the number of subcutaneous, luciferase-positive cells is similar and the bioluminescent signal strength is equivalent, the detector registers a lower signal in B6 mice compared to a strain like FVB which has white fur.

3, Why do the engraftment frequencies of Klg-1 teratomas so differ among mouse strains, as shown in Figure 2e? Were the results expected?

We appreciate this concern, although this was expected. There are known, strain-dependent differences based on variations in vascularization, differentiation signals at the implant site, and immune system activity. However, we are currently lacking in extensive data sets - or the embryonic stem cell (ESC) lines - from isogenic transplants using FVB, C3H, BALB/C, and CD-1 cells and recipients. This prevents us from delving deeper into this question. The key is that, using our conditions, we never see uncloaked donor ESCs forming teratomas in immune competent, allogeneic recipients.

4, The explanation of the results in Figure 4J is insufficient. If apoptosis through FASL is considered, it would be necessary to conduct co-culture experiments with individual T cell subsets and evaluate apoptotic sensitivity of these T cell subsets.

We agree with the reviewer that our interpretation of these results was somewhat speculative, and we did not specifically test in a co-culture assays if the cell death was mediated through FASL.

To avoid mislead our readers about the conducted experiments, we have removed the underlined lines from the results:

...and a depletion of central T and effector memory T cell subsets (Fig. 4j). Since memory T cells express FAS receptors⁴, it's possible their depletion results from FAS activation by FASL which is highly expressed on cloaked RPEs as a transgene (Fig. 4d).

5, The specific subset of PBMCs that shows increased or decreased reactivity in Figure 5 is not indicated. To confirm the experimental setting whether NK cell tolerance is induced, it is necessary to provide HLA information of the experiment, in addition, the experimental observation should be confirmed with multiple donors.

In main (now) Extended Data Fig. 9, we show the interactions of PBMCs flowing through a microfluidic device in which the interior walls (lumen) was coated with endothelial cells (ECs) derived from either cloaked or uncloaked iPSCs. Our data showed that the cloaked ECs, unlike uncloaked ECs, resisted clearance from the chamber by allogeneic, circulating PBMCs and increase the apoptosis of the PBMCs themselves.

In response to the valuable feedback from Reviewer #3, we have moved Main Fig. 5 into the Extended Data section. This came from the concern that these experiments do not significantly add to the overall findings, and distract from the manuscript's more novel findings, especially the work demonstrating how cloaked tissue can protect unmodified cells.

Since we have shifted the prominence of these experiments and figure, we hope the reviewer will agree that additional experiments would not substantially enhance the manuscript.

6, To demonstrate whether cloaked teratomas after GCV treatment can avoid humoral immune responses and protect cells inside of the container is important. So, I would recommend conducting an experiment where non-cloaked teratomas are initially immunized to induce anti-MHC antibodies in FVB mice and then transplanting cloaked ES to make immune-tolerant teratomas with GCV treatment. Furthermore, it would be advisable to confirm whether non-cloaked cells injected into the cloaked immune-tolerant structures can survive avoiding antibody-mediated attack.

We thank the reviewer comment, and certainly understand the interest in specific mechanisms of rejection, including Antibody-mediated rejection (AMR). There are many other pathways that support rejection of allogeneic cells and tissues, including cytotoxic NK and T-cells, myeloid populations, and other innate subsets. The pathways may vary in their time scales depending on whether cells or organs are being transplanted.

After transplantation of cells (as opposed to tissue with existing vasculature) AMR likely begins over the medium-term (weeks) after the initiation of adaptive T- and B-cell response. Chronic AMR that builds and is active over the long-term (years) is a serious issue in all aspects of transplant biology^{5,6}.

It is this reason that we chose an *in vivo* model system that allowed us to test long-term survival in a completely immune component background. Our data shows the survival of cloaked Klg-1 mESCs for almost 200 days in immunocompetent, allogeneic immunocompetent recipients (Main Fig 2d, and Supplemental Table 3). Once an allogeneic teratoma is formed and stabilized, we have never observed any instances where it was later cleared or rejected. In other words, all teratomas originating from iPSCs appear to be fully accepted in the long-term. Furthermore, based on Fig. 5c, we found that xenogeneic cells transplanted into cloaked teratomas also survive long-term, with two of these recipients even kept for 240 days to measure xenogeneic cell survival.

Many humanized models of allotransplantation are either unable of, or very poor at, recapitulating these time-scales. Often in these models, and in many other studies, 50 days is claimed to reflect long-term acceptance. We have been more cautious, as we are aware of the long-term horizons of chronic mechanisms such as AMR. As our cells are accepted in the long-term, we can conclude are not being rejected by AMR (or other potential mechanisms).

In this manuscript and work, we did not extensively explore the empirical mechanisms of this immune evasion *in vivo*, beyond our characterization of co-culture experiments shown in Figure 4. Based on previous feedback, we deliberately kept this manuscript focused on the proof-of-principle for our engineering approach to immune escape, and for creating immune-privileged tissue that can host unmodified cells and tissues. The feedback from reviewer #3 also support this approach and current framing.

We trust that the reviewer will agree that, based on these reasons, an in-depth study of these mechanistic aspects would be more appropriately addressed in future research, which we are currently planning.

Reviewer #5 (Report for the authors (Required)):

The authors prepared universal mESCs and hESCs using transposon-based overexpression of eight immunomodulatory transgenes in this manuscript. The selection of eight immunomodulatory transgenes is original work in this manuscript, although universal or hypoimmunogenic pluripotent stem cells have been reported by over expression of immunomodulatory transgenes such as PDL1 and CD47 (Q. Ye et al., Generation of universal and hypoimmunogenic human pluripotent stem cells, *Cell Proliferation*, 53 (2020) e12946). This method is a kind of gene therapy, which is extremely difficult to be accepted and approved by FDA, although some immunotherapy has been approved currently. There is a report to generate universal hiPSCs without using gene transduction (TC Sung et al., Transient characteristics of universal cells on human-induced pluripotent stem cells and their differentiated cells derived from foetal stem cells with mixed donor sources, *Cell Proliferation*, 54(3) (2021) e12995). The authors should write the merit of gene transduction method of this study together citing the above literature. The evaluation of immune privilege of the author's cells are reasonable in this study. However, when they prepare human ESCs or hiPSCs by overexpression of immunomodulatory transgenes, it is more difficult to study the immune-privilege evaluation. The challenging to generate universal human pluripotent stem cells and their universal and differentiated cells are important, the reviewer recommend the publication of this manuscript in *Nature Biomed Engineering* after revision of the following comments.

Specific comments:

1. On Line 70, "which remove the major source of antigen mismatch between donor and recipients5-9." There is an excellent review, which explain this issue. The authors may cite the following article; "Q. Ye et al., Generation of universal and hypoimmunogenic human pluripotent stem cells, *Cell Proliferation*, 53 (2020) e12946".

This is a very helpful review. We have added the reference to the introduction (2nd paragraph).

2. The authors should write the merit of gene transduction method of this study and
3. The authors used piggyBac and Sleeping Beauty transposon (cloaking) vectors in which transcription was driven by a CAG promoter. Why they did not select to use CRISPR/Cas-9 genome editing method?

Indeed, these are critical questions (with our response merging both #2 and #3). For our proof-of-principle studies, we used transposons because they are efficient and result in multicopy and multisite insertion of the transgenes. Our cloaked Klg-1 line had approximately 50 transposon vector

integrations within the genome (shown in supplemental Table 4), indicating that each of the 8 cloaking factors had roughly 5-6 copies on average at different genomic sites. This allows for extremely high transgene-expressing clones which we could enrich for and isolate using drug selection and Flow cytometry sorting (Extended Data Fig 1 for mouse cells, Extended Data Fig 5 for human). This also allows for the generation of clones with unique expression “signatures”.

We propose in our discussion that this approach could be a vital step in developing cloaked lines (See Fig 1). Our data imply that it is crucial to have high and reliable (non-silencing) expression of the cloaking factors.

However, since transposon are integrated randomly, it becomes more challenging to replicate the engineering process precisely from one line to another. Targeted integration, using nucleases such as CRISPR/Cas9, which introduce a one (hemizygous) or two (homozygous) copies of the transgene(s) at a defined locus may be ideal due to regulatory considerations.

Nevertheless, identifying defined sites where a single (or double) copy integration leads to reliable and high expression is far from trivial. Numerous groups have spent decades trying to achieve this. Even many well-known safe harbor sites, such as AAVS1, can lead to silencing issues in various cell types.

In pursuit of a clinically relevant genome editing strategy, we are currently combining these transgenes into several, multicistronic vectors and integrating them into defined genomic sites with CRISPR to ensure predictable and high expression. This work is still in progress and will take more time. Although a clinically-relevant genome editing strategy was not necessary for our proof-of-principle studies, we agree with the reviewer that it is a part of the long-term goal of engineering cells for clinical use.

4. On L-235; “These vectors were then transfected into a FS H1 human ESC (hESC) line” should be revised as “These vectors were then transduced into a FS H1 human ESC (hESC) line”.

In our experience and interactions with other groups, transduction has consistently denoted the introduction of transgenes (or any foreign DNA) using viral vectors, such as Lenti. Transfection, , on the other hand, typically refers to the use of chemical reagents (lipofectamine 3000, Jetprime, Fugene) or even electroporation (Neon, nucleofection, etc), for inserting plasmids into the cells.

All of our experiments relied on simple chemical reagents to introduce the transgene-containing transposon vectors (plasmid) into the cell to integrate into the genome, along with the plasmids encoding the transient, non-integrating transposase enzymes (sleeping beauty SB100x or piggyBac hyPBBase). Accordingly, we believe ‘transfection’ is the most accurate term to describe these experiments.

5. Immune-privilege study of FS hESCluc+ was performed using NSG mice. Most of researchers are using immune-privilage experiments using humanized mice. It is necessary to show the immune-privilege results using humanized mice.

We appreciate this suggestion, and acknowledge the significant role that humanized mouse models have in allotransplantation studies.

While we recognize the utility of these models, we find they inadequately simulate many of the long term and chronic rejection pathways that prove most challenging to circumvent. Please refer to our response to reviewer #4, point 6 for additional details, particularly to our concerns regarding antibody-mediated rejection (AMR), which can mound and persist for years. We argue that the known problem of chronic AMR, highlights the inadequacy of these models for understanding and testing many key components of long-term immune rejection.

The use of fully immune-competent recipients in our study allowed us to test these long-term pathways in mammalian recipients. Consequently, we are confident in saying that our engineered cells are indeed long-term accepted (as long as we keep the recipients alive), unlike other studies that might only demonstrate survival for 30-50 days, depending on the model.

However, we recognize there are important differences between the mouse and human immune system. It is for this reason that we also generated human cells (please see new Fig 4) which overexpress the functional orthologues of the cloaking transgenes tested in mouse PSCs.

We examined these in a series of *in vitro* experiments (Fig. 4). However, we did not test them in any humanized mouse models due to our aforementioned concerns. We believe the next informative set of experiments will come from immunocompetent, non-human primate models, in which we currently have several ongoing studies. However, these studies, which are extremely expensive, are were not required for these proof-of-principle investigations.

References

1. Russell, S.J. & Barber, G.N. Oncolytic Viruses as Antigen-Agnostic Cancer Vaccines. *Cancer Cell* **33**, 599-605 (2018).
2. Santos Apolonio, J. et al. Oncolytic virus therapy in cancer: A current review. *World J Virol* **10**, 229-255 (2021).
3. Liang, Q. et al. Linking a cell-division gene and a suicide gene to define and improve cell therapy safety. *Nature* **563**, 701-704 (2018).
4. Ramaswamy, M. et al. Specific elimination of effector memory CD4+ T cells due to enhanced Fas signaling complex formation and association with lipid raft microdomains. *Cell Death Differ* **18**, 712-720 (2011).
5. Mayer, K.A., Doberer, K., Eskandary, F., Halloran, P.F. & Bohmig, G.A. New concepts in chronic antibody-mediated kidney allograft rejection: prevention and treatment. *Curr Opin Organ Transplant* **26**, 97-105 (2021).
6. Garces, J.C. et al. Antibody-Mediated Rejection: A Review. *Ochsner J* **17**, 46-55 (2017).

Rebuttal 2

Point-by-point responses from Authors to final comments from Reviewers.

Reviewer #3:

1. In the abstract – “...abrogated activation of human hematopoietic cells” is imprecise and incomplete. The expressed genes appear to avert destruction of foreign grafts by preventing activation or eliminating immune responder cells and by protecting foreign cells from immune and inflammatory injury.---The transgenes are not acting on human hematopoietic cells.

Please note this phrase in the abstract refers to our generation of human ESCs with human immunomodulatory transgenes (Figure 4, Extended Data 4-7). For this we used the human functional orthologues of the mouse transgenes which were used to generate all our mouse data (Fig 1-3).

Fig4 shows co-culture of the human ESCs expressing the immune transgenes with human hematopoietic cells, and the (expected) reduction in a number of inflammatory related markers; IFNg, TNFa, etc, which we described as “...abrogated activation...”.

2. The extended critique of iPSC-based therapeutics might be correct but it is tangential, at best. If the manuscript were to be further revised, the authors should consider eliminating most of the first paragraph (all but the first sentence) of the Introduction to avoid unnecessarily alienating tens of thousands of people endeavoring to advance iPSC-based therapies and immunosuppressive drugs. The merit of the authors’ technology speaks for itself and is not made more attractive by vague criticism of other technologies, especially since the authors have not systematically compared the merits and toxicities of other technologies with their own.

We very much appreciate this critique. We also do not wish to alienate others working towards the same end goal - successful cell-based therapies. We’ve re-wrote the first paragraph of the introduction to remove the critical tone and reframe these “other” approaches as other promising avenues.

3. “...universal” cell lines to derive therapeutic products for the treatment of many MHC-mismatched patients”---this phrase makes no sense and the implicit claim regarding universal cell lines is imprecise and easily attacked. The authors’ approach prevents rejection commendably well and establishes what appear to be immune privileged sites. The approach does not support or advance use of universal cell lines or even explain what is meant by that term. Is a cell line deemed “universal” because it is accepted by all recipients or is it universal because it can be differentiated into any mature tissue? The authors infer the former but mention as such is redundant. Also potentially redundant and unclear is the expression: “treatment for MHC mismatched patients.” What are MHC-mismatched patients? The authors probably mean that the grafts are not matched with recipients but one usually reserves use of MHC matching and mismatching to circumstances in which there is a real donor and recipient. If the manuscript is revised the authors to delete both claims.

These points are well taken. Especially the critique that the term “universal” is imprecise unless specifically defined as “pluripotency” or “escaping allogeneic immune response”. The reviewer is right that we’ve used “Universal” to refer to immune escape (PMID 32763181). Nevertheless we changed the language in this final paragraph of the introduction to be more precise.

4. One reviewer asks whether all modifications are necessary and suggests requirements must be tested. The question is important but premature. Once this set of modifications is reported, I suspect the authors and others will undertake testing in various applications and it might well be found that different combinations or levels of expression are needed for protection of different tissues from rejection. One might well find that different levels of expression are needed to establish an immune privileged site that to prevent rejection of transduce cells.

We completely agree with all points and the importance of the question. We plan to explore this in future studies.

5. One reviewer suggests the authors should compare their approach to results obtained using MHC-deficient ES-derived cells. I disagree. Efforts to promote engraftment by targeting or blocking one or more MHC encoded proteins have been pursued to little benefit for decades. Besides concerns about infection, which certainly are valid, MHC deficient cells are subject to injury by NK cells and other cells and products of MHC and related genes that are expressed in aberrant form are quite immunogenic. However, the reviewer appears to have raised the point because the authors unwisely compared their approach to MHC targeting.

We appreciate the comment, and the note about the implications of MHC KO and infection. We note that many groups are building hypoimmunogenic platforms around MHC knock-out, with or without additional immune transgenes. Our comparison is primarily to highlight the distinction of our approach, which is based solely on transgene insertion.

6. One reviewer discusses anti-MHC antibody responses to engineered cells and recommends further experiments to test the impact of such antibodies. Although anti-MHC antibodies might be assayed as one measure and perhaps the most sensitive measure of immunogenicity, I disagree about the imperative for such testing. The preponderance of work over the last 80 years exploring the impact of antibodies on transplants indicates anti-MHC antibodies important for the outcome of organ transplants have little or no detrimental impact on cell and tissue transplants. The observations presented in the manuscript are fully consistent with that experience – if anti-MHC antibodies are made, they evidently cause little or no damage. As for the merit of testing, I might find the results interesting but not illuminating without much more information than most can imagine.

We completely agree. In the manuscript we did not explore the precise mechanisms of *in vivo* immune escape in our mouse experiments. There may be an absence of anti-graft antibodies, or even if they are generated, they are not causing clearance of the graft. It is extremely interesting either way, but for later studies.

7. One reviewer suggests the authors show immune privilege in humanized mice. I disagree. Humanized mice have many valid uses, but the mice are aberrant in many respects and might or might not faithfully model immune privilege. More importantly, the human T cells generated and surviving in humanized mice are far less competent than murine T cells in wild type mice.

Completely agree. Even with the promising advances coming from next generation humanized mouse models, they are deficient in many respects especially for long-term studies.

Reviewer #4:

#1 I understand that this research is based on the hypothesis that in a long-term xenotransplantation model, multiple complementary immune evasion mechanisms would be necessary to completely avoid immune rejection. I also understand the caution to avoid suggesting that all eight factors are essential and instead demonstrate that these eight factors are sufficient to achieve the objective. I accept that the analysis of the impact of each factor on immune evasion is beyond the scope of this paper.

Thank you. No comment or change needed.

#2 I understand that in past revisions, experiments and discussions related to this topic were conducted but were removed due to editorial reasons. I also understand the assertion that careful experiments, such as investigating antigen-specific viral clearance, should be left for future characterization studies. I believe that this assertion could be included in the Discussion section.

We completely agree that it will be important to understand how any engineering related to immune evasion will affect the cell's ability to respond to infection. Can they present antigen, will it affect clearance, etc? We have added two short sentences in the first paragraph of the discussion.

#3 I appreciate the authors' approach of not overly emphasizing clinical applications in this study and instead focusing on conducting more detailed research and experiments to strengthen the scientific foundation and provide a more solid basis for the potential clinical benefits of the approach. Readers will quickly recognize that this research includes valuable insights that evoke clinical advantages. Therefore, it is important to include cautious statements that accurately convey the current understanding.

Thank you, no change needed.

Reviewer #5 (Report for the authors (Required)):

The authors revise the manuscript appropriately. No further comments.

Thank you, no change needed.